# Generalizing Beyond Suboptimality: Offline Reinforcement Learning Learns Effective Scheduling through Random Solutions

## Abstract

The Job Shop Scheduling Problem (JSP) and Flexible Job Shop Scheduling Problem (FJSP) are combinatorial optimization problems with wide-ranging applications in industrial operations. In recent years, many online reinforcement learning (RL) approaches have been proposed to learn constructive heuristics for JSP and FJSP. Although effective, these online RL methods require millions of interactions with simulated environments, and their random policy initialization leads to poor sample efficiency. To address these limitations, we introduce Conservative Discrete Quantile Actor-Critic (CDQAC), a novel offline RL algorithm that learns effective scheduling policies directly from datasets, eliminating the need for training in a simulated environment, while still being able to improve upon suboptimal training data. CDQAC couples a quantile-based critic with a delayed policy update, estimating the return distribution of each machine–operation pair rather than selecting pairs outright. Our extensive experiments demonstrate CDQAC's remarkable ability to learn from diverse data sources. CDQAC consistently outperforms the original data-generating heuristics and surpasses state-of-the-art offline and online RL baselines. In addition, CDQAC is highly sample efficient, requiring only 10–20 training instances to learn high-quality policies. Notably, CDQAC performs best when trained on datasets generated by a random heuristic, leveraging their wider distribution over the state space, to surpass policies trained on datasets generated by significantly stronger heuristics.

## 1 Introduction

The Job Shop Scheduling Problem (JSP) and Flexible Job Shop Scheduling Problem (FJSP) are fundamental challenges in manufacturing and industrial operations (Bhatt & Chauhan, 2015), where the goal is to optimally schedule *jobs* on available *machines* to minimize objectives such as total completion time (makespan). Exact methods such as Constraint Programming (CP) (Da Col & Teppan, 2022) and Mathematical Programming (Fan & Su, 2022) guarantee optimality but face scalability issues for large-sized instances. Therefore, in practice, heuristic methods such as Genetic Algorithms (GA) (Bhatt & Chauhan, 2015) and Priority Dispatching Rules (PDRs) (Veronique Sels & Vanhoucke, 2012) are preferred, as they can find acceptable solutions in reasonable time.

Recently, deep reinforcement learning (RL) has shown promise for learning priority dispatching rules (PDRs). Approaches such as Learning-to-Dispatch (L2D) (Zhang et al., 2020) learn policies that generalize from small to larger instances and solve new cases orders of magnitude faster than exact solvers or evolutionary algorithms. However, most RL methods train policies from scratch via trial-and-error in simulators. Due to random initialization, they typically require millions of interactions to converge, leading to severe sample inefficiency (Mai et al., 2022). At the same time, a wide range of heuristics, such as PDR and GA, are commonly used for JSP, FJSP, and related scheduling problems. This widespread use should allow for the collection of training data. However, because these heuristics do not guarantee optimality, this training data is inherently suboptimal.

Offline RL emerges as an alternative approach to learning effective dispatching policies by training directly on datasets generated by suboptimal heuristics. Instead of simply imitating observed actions, offline RL methods learn their estimated value and leverage this to generalize policies that can surpass the heuristics that generated the dataset (Levine et al., 2020; Kumar et al., 2022). Recently, Remmerden et al. (2025) proposed the first offline RL method, called Offline-LD, to solve

JSP, and showed that it can learn good scheduling policies with a small training dataset of only 100 instances, outperforming several methods, including the online RL method L2D, behavioral cloning, and heuristics. Despite its promising performance, Offline-LD relies on high-quality solutions generated by a Constraint Programming (CP) solver for training. However, generating training data using the CP solver is computationally expensive and intractable for large problem instances.

We propose **Conservative Discrete Quantile Actor-Critic** (CDQAC), a novel offline RL method, that learns effective scheduling policies from **low-quality data** generated by a wide range of heuristics. CDQAC learns an approximated representation of the value of each action from which it can generalize a new policy that can outperform the heuristic that generated the data. CDQAC achieves this through a quantile-based critic, with a novel dueling architecture. This critic provides value estimates that guide the actor, while a delayed policy update prevents the propagation of early noisy critic predictions, ensuring stable joint learning of the policy and value function.

Our work offers the following contributions: (1) We propose CDQAC, a novel offline RL method, which can effectively learn a scheduling policy from a wide variety of datasets of various quality. (2) We show that CDQAC significantly outperforms all other baselines, including Offline-LD, heuristics used to generate training sets, and online RL baselines on JSP and FJSP benchmark instances. (3) CDQAC is highly sample efficient, requiring only 10-20 instances to learn good policies, significantly less than online RL approaches, which require up to 1000 instances. (4) CDQAC achieves the highest performance when trained on a dataset generated by random heuristics, contradicting previous findings in offline RL research, which generally show that the combination of higher-quality and slightly lower-quality training examples results in better performance (Schweighofer et al., 2022; Kumar et al., 2022).

## 2 RELATED WORK

**Learning-based methods for Scheduling Problems.** Most prior work on scheduling has focused on JSP. Early work showed that online reinforcement learning (RL) with graph neural networks (GNN) can learn effective scheduling policies (Zhang et al., 2020; Park et al., 2021; Smit et al., 2025), later improved through curriculum (Iklassov et al., 2023) and imitation learning (Tassel et al., 2023). Recent approaches learn improve heuristics via RL (Zhang et al., 2024a;b), while self-supervised methods outperform RL at the cost of longer training (Corsini et al., 2024; Pirnay & Grimm, 2024). None of these methods can learn a policy for the Flexible Job Shop scheduling problem (FJSP), due to the increased complexity of selecting both an operation and a machine. Song et al. (2023) introduced a heterogeneous GNN for FJSP, which learns the relation between machines and operations, and Wang et al. (2023) proposed a dual attention architecture to capture this relation, whereby both methods can also function for JSP (Reijnen et al., 2023). However, all of these methods for JSP and FJSP remain sample-inefficient, requiring extensive interactions with simulated environments to learn well-performing scheduling policies. In contrast, we focus on an offline RL approach that can learn directly from diverse and potentially suboptimal datasets, thereby eliminating the need for simulator-based training.

**Offline Reinforcement Learning.** Most offline RL work focus on continuous action spaces (An et al., 2021; Kostrikov et al., 2021), with limited exploration in discrete domains. Transformer-based sequence models show promise (Chen et al., 2021; Janner et al., 2021) but assume fixed state/action sizes, incompatible with FJSP/JSP instance-dependent state/action sizes. Conservative Q-learning (CQL) (Kumar et al., 2020) has shown promise for discrete action spaces (Kumar et al., 2023) and prevents overestimation of OOD actions through regularization of Q-values. Offline-LD (Remmerden et al., 2025) first demonstrated offline RL's potential for JSP using (near-)optimal constraint programming solutions, surpassing online and imitation methods, especially with noisy data. However, Offline-LD focused solely on JSP and need (near-)optimal data for training. We extend this to FJSP, focusing on learning from diverse suboptimal examples. Consequently, we build upon CQL, well-suited for such data, by introducing novel algorithmic and architectural components tailored for effective scheduling in FJSP and JSP. This distinguishes our setting from imitation learning (IL), also known as behavioral cloning (BC), which learns to imitate the policy that generated optimal or near-optimal solutions (Luo et al., 2023; Drakulic et al., 2023; Lee & Kim, 2025).

## 3 PRELIMINARIES

**JSP & FJSP.** We formulate the Job Shop Scheduling (JSP) and Flexible Job Shop Scheduling Problem (FJSP) as follows. Given a set of $n$ jobs, represented as $\mathcal{J}$, and a set of $m$ machines, represented as $\mathcal{M}$, each job $J_i \in \mathcal{J}$ has $n_i$ operations. These operations $\mathcal{O}_i = \{O_{i,1}, O_{i,2}, ..., O_{i,n_i}\}$ must be processed in order, forming a precedence constraint. In JSP, each operation $O_{i,j}$ can only be processed by a single machine, whereas in FJSP, $O_{i,j}$ can be processed on any machine in its set of compatible available machines $\mathcal{M}_{i,j} \subseteq \mathcal{M}$. Each machine $M_k \in \mathcal{M}_{i,j}$ has a specific processing time for an operation $O_{i,j}$ denoted as $p_{i,j}^k$, where $p_{i,j}^k > 0$. The objective is to minimize the makespan, defined as the completion of the last operation $C_{\max} = \max_{O_{i,j} \in \mathcal{O}} C(O_{i,j})$, where $C(O_{i,j})$ represents the completion time of operation $O_{i,j}$.

**Offline Reinforcement Learning.** We formalize FJSP and JSP as a Markov Decision Process (MDP) denoted as $\mathbf{M}_{\text{MDP}} = \langle \mathcal{S}, \mathcal{A}(s_t), P, R, \gamma \rangle$. A state $s_t \in \mathcal{S}$ represents the progress of the current schedule in the timestep $t$, and includes all operations $O_{i,j} \in \mathcal{O}_t$ that are available to be scheduled on machines $M_k \in \mathcal{M}_t$, whereby $\mathcal{M}_t$ only contains machines that are free at timestep $t$. The action space $a_t \in \mathcal{A}(s_t)$ corresponds to all available machine-operation pairs $(O_{i,j}, M_k)$ at $t$. $P$ is the transition function and determines the next state $s_{t+1}$ on the selected machine-operation pair $(O_{i,j}, M_k)$, whereby unavailable pairs, due to $M_k$ being selected, being removed and new available pairs added. The reward $r_t$ is the negative increase in the (partial) makespan resulting from action $a_t$: $r_t = \max_{O_{i,j} \in \mathcal{O}} C(O_{i,j}, s_t) - \max_{O_{i,j} \in \mathcal{O}} C(O_{i,j}, s_{t+1})$ $\gamma$ is the discount factor that determines the importance of future rewards. We set $\gamma = 1$. In offline RL, a policy $\pi(a|s)$ is learned through a static dataset $D = \{(s, a, r(s,a), s')_i\}$, where $s'$ is the next state. $D$ is generated through one or more behavioral policies $\pi_\beta$.

## 4 CONSERVATIVE DISCRETE QUANTILE ACTOR-CRITIC FOR SCHEDULING

Our goal is to learn a scheduling policy $\pi_\psi$ from a static dataset $D$ that surpasses the behavioral policies $\pi_\beta$ that generated it. $\pi_\beta$ may be any (possibly non-Markovian) heuristic, such as PDRs, genetic algorithms, or random schedulers (Kumar et al., 2022). To outperform $\pi_\beta$, the learner must estimate accurate state–action values $Q_\theta(s, a)$ in $D$ and "stitch" high-value segments into a better policy. This differs from Behavioral Cloning, which learns to imitate the actions of $\pi_\beta$. Because $\pi_\psi$ is updated solely via $Q_\theta$, the critic must (1) model the *return distribution* for state–action pairs observed in $D$ and (2) remain *conservative* on out-of-distribution (OOD) actions to avoid overestimation under distributional shift, while still enabling improvement beyond the data. For this purpose, we propose **Conservative Discrete Quantile Actor-Critic** (CDQAC), an offline RL approach for JSP and FJSP. CDQAC introduces a novel offline RL approach for scheduling, that integrates a **quantile critic** (Dabney et al., 2018) with a delayed policy update, enabling the learning of a scheduling policy from a dataset $D$ composed of suboptimal examples, while still discovering policies that outperform those contained in $D$.

**Quantile Critic.** To learn an accurate representation of the value of all scheduling actions in a dataset $D$, we utilize a *distributional* approach for our critic. In a distributional approach, we want to approximate the random return $Z^\pi = \sum_{t=0}^\infty \gamma^t r(s_t, a_t)$, rather than approximating the expectation as $Q^\pi(s, a) = \mathbb{E}[Z^\pi(s, a)]$, and it has shown to learn more accurate representations than standard DQN (Bellemare et al., 2017; Dabney et al., 2018). To approximate $Z^\pi$, we use a **quantile critic**, who approximates the return by learning a set of $N$ quantiles. These quantiles are estimated for specific fractions $\tau_n = \frac{2n-1}{2N}, n \in [1, ..., N]$, which represent the target cumulative probabilities for which the quantile values are estimated, formulated as:

$$Z_{\theta_i}(s, a) = \frac{1}{N} \sum_{j=1}^{N} \delta\big(\theta_i^j(s, a)\big), \tag{1}$$

with $\theta_i^j$ predicting the $j$-th of $N$ quantiles and $\delta$ the Dirac delta. We update the quantile critic through a distributional Bellman update (Bellemare et al., 2017) given as:

$$\mathcal{T}Z(s, a) = r(s, a) + \gamma Z_{\hat{\theta}}(s', a'), \quad s' \sim D, \quad a' \sim \pi_\psi(\cdot \mid s'), \tag{2}$$

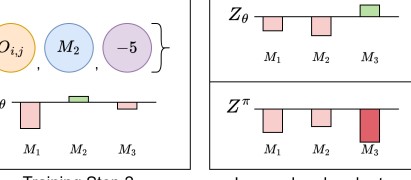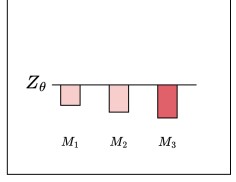

Figure 1: Illustrative example of overestimating OOD actions. In training steps 1 and 2 examples are shown of negative outcomes of pairing operation $O_{i,j}$ with either machine $M_1$, with a reward of $-3$, or $M_2$, with a reward of $-5$, learning that $M_3$ results in the best outcome, since the combination $(O_{i,j}, M_3)$ does not exist in the dataset. The real return $Z_\pi$ shows that $M_3$ results in the worst outcome. CQL ensures OOD actions are not overestimated, in comparison to actions in the dataset.

whereby $\hat{\theta}$ represents the target network. The action $a'$ for the target state $s'$ is carried out by the current policy $\pi_\psi$, ensuring that the learned value distribution reflects the expected return under the policy $\pi_\psi$. We use the distributional Bellman update from Eq. 2 to calculate the temporal difference (TD) loss for our critic, which is as follows:

$$\mathcal{L}_{TD}(\theta) = \mathbb{E}_{s,a,s'\sim D, a'\sim\pi_\psi(\cdot|s)}\left[\rho_\tau^H(\mathcal{T}Z_{\hat{\theta}}(s', a') - Z_\theta(s, a))\right], \qquad (3)$$

where $\rho_\tau^H$ is the asymmetric quantile Huber loss proposed in (Dabney et al., 2018), which updates $\theta$ for all quantile fractions $\tau$. The target network is updated through a Polyak update, whereby $\hat{\theta}$ is updated as a fraction $\rho$ of $\theta$. We can retrieve a scalar value from $Z_\theta$, by the mean over the quantiles $Q_\theta^Z(s, a) = \mathbb{E}[Z_\theta(s, a)]$.

**Conservative Q-Learning.** In the offline setting, CDQAC updates $Z_\theta$ using targets that can involve actions without support in the static dataset $D$. This support mismatch, i.e. *distributional shift* between the state–action distribution in $D$ and that induced by the learned policy, leads the critic to overestimate the values for out-of-distribution (OOD) actions, as illustrated in Fig. 1. This overestimation is not an issue for online RL, since it can explore these actions during training; however, offline RL cannot due to learning from a static dataset. To avoid this overestimation, we add Conservative Q-learning (CQL) (Kumar et al., 2020) to the loss of the critic. CQL penalizes overestimation of OOD actions, by introducing a regularization term used in combination with standard critic loss:

$$\mathcal{L}_Z(\theta) = \alpha_{\text{CQL}}\mathbb{E}_{s\sim D}\left[\log\sum_{a'\in\mathcal{A}(s)}\exp(Q_\theta^Z(s, a')) - \mathbb{E}_{a\sim D}[Q_\theta^Z(s, a)]\right] + \mathcal{L}_{TD}(\theta), \qquad (4)$$

where, $\alpha_{\text{CQL}}$ determines the strength of the penalty, and $\mathcal{L}_{TD}(\theta)$ is the loss in Eq. 3.

**Delayed Policy.** The approximated return $Z_\theta$ allows CDQAC to learn which scheduling action to perform and which not. This requires $Z_\theta$ to accurately model the real return $Z^\pi$, which it does not yet do at the start of training. $\pi_\psi$ will learn to maximize based on a noisy critic $Z_\theta$, who in turn will be updated based on noisy updates of $\pi_\psi$ (Eq. 2). To prevent this, we introduce a *delayed* policy update, where $\pi_\psi$ is updated every $\eta$ steps, based on prior work in online RL (Fujimoto et al., 2018). This allows $Z_\theta$ to receive more updates than $\pi_\psi$, improving the stability and accuracy of both $\pi_\psi$ and $Z_\theta$. We formalize the loss of $\pi_\psi$ as follows:

$$\mathcal{L}_\pi(\psi) = \mathbb{E}_{s\sim D, a\sim\pi_\psi(\cdot|s)}\left[-Q_\theta^Z(s, a) + \lambda\,\mathcal{H}\big[\pi_\psi(\cdot\mid s)\big]\right], \qquad (5)$$

where $\mathcal{H}[\pi_\psi(\cdot\mid s)]$ is an entropy bonus preventing $\pi_\psi$ from converging to a single action and its strength is determined by $\lambda$. To avoid overestimation in Q-learning-based actor-critic methods, we parameterize $Z_\theta$ with two heads ($Z_{\theta_1}, Z_{\theta_2}$) and calculate the target $Z_{\hat{\theta}}$ (Eq. 3) and $Q_\theta^Z$ in the policy update (Eq. 5) as the minimum value of both heads $Z_\theta = \min(Z_{\theta_1}, Z_{\theta_2})$ (Christodoulou, 2019; Zhou et al., 2024).

### 4.1 NETWORK ARCHITECTURE

To encode an FJSP or JSP instance, we use a dual attention network (DAN), adapted from DANIEL (Wang et al., 2023), for both the policy network $\pi_\psi$ and the quantile critic $Z_\theta$. Fig. 2

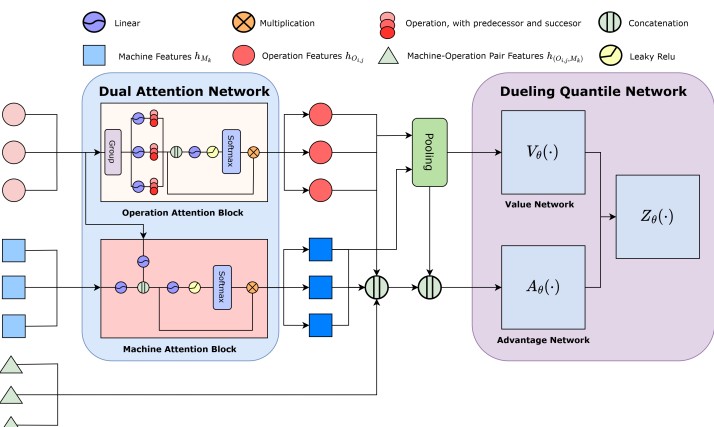

Figure 2: The network architecture. (Left) The Dual Attention Network (DAN) encodes the operations and machines. (Right) The Dueling Quantile Network uses these embeddings to learn the machine-operation pair, whereby it combines the Value $V_\theta$ and Advantage $A_\theta$ streams through Eq. 6.

shows our network architecture. DAN processes two parallel attention streams that take the relevant operations $O_{i,j} \in \mathcal{O}_t$ and machines $M_k \in \mathcal{M}_t$. DAN learns the complex relation between each machine-operation pair at timestep $t$ as input and embeds them as $h_{O_{i,j}}$ and $h_{M_k}$. A detailed explanation of DAN and the input features can be found in App. B.

From machine embeddings $h_{M_k}$ and operation embeddings $h_{O_{i,j}}$, we calculate a global embedding as $h_G = \left[\left(\frac{1}{|\mathcal{O}_t|}\sum_{O_{i,j}\in\mathcal{O}_t} h_{O_{i,j}}\right) \| \left(\frac{1}{|\mathcal{M}_t|}\sum_{M_k\in\mathcal{M}_t} h_{M_k}\right)\right]$, where $\|$ is a concatenation.

For the actor network, we use the global embeddings $h_G$, combined with the embeddings of the operation $h_{O_{i,j}}$ and machine $h_{M_k}$, and the specific features of the machine-operation pair $h_{(O_{i,j},M_k)}$, as input for the policy $\pi_\psi$. This allows $\pi_\psi$ to select a machine-operation pair, based on the embeddings of the machine-operation pair in relation to the global embedding.

**Dueling Quantile Network.** The quantile critic in CDQAC uses a novel dueling architecture based on prior work by Wang et al. (2016), which divides the state action value into two components: a value stream $V(s)$ and an advantage stream $A(s,a)$. The major benefit is that $V_\theta$ is updated at each training step, while $A_\theta$ is only updated for each individual machine-operation pair, allowing $V_\theta$ to learn a richer representation and more accurate $Z_\theta$. In Wang et al. (2016) approach $V_\theta$ and $A_\theta$ share the same input, we propose separate inputs where $V(s)$ only receives the global embedding $h_G$, whereas $A_\theta$ also receives the operation-, machine-, and pair-specific embeddings (Fig. 2). This allows $V_\theta$ to focus only on the state value, whereas $A_\theta$ can focus on each individual machine-operation pair $(O_{i,j}, M_k)$, resulting in the following formulation:

$$Z_\theta(h_{O_{i,j}}, h_{M_k}, h_{(O_{i,j},M_k)}, h_G) = V_\theta(h_G) + \Big(A_\theta(h_{O_{i,j}}, h_{M_k}, h_{(O_{i,j},M_k)}, h_G)$$
$$- \frac{1}{|\mathcal{A}(s_t)|}\sum_{(O',M')\in\mathcal{A}(t)} A_\theta(h_{O'}, h_{M'}, h_{(O',M')}, h_G)\Big), \quad (6)$$

where $\mathcal{A}(s_t)$ are all the available machine-operations pairs $(O', M')$ in state $s_t$ at timestep $t$. In Eq. 6, we subtract the average advantage stream from the advantage of an action. This is required since the value stream and the advantage stream are not uniquely identifiable (Wang et al., 2016).

## 5 EXPERIMENTS

**Generated & benchmark instances.** We use generated instances for training (500 instances) and evaluation and standard benchmarks. **FJSP:** train on sizes $\{10\times5, 15\times10, 20\times10\}$; each job has $\lfloor 0.8m \rfloor$–$\lfloor 1.2m \rfloor$ operations; processing times are integers in $[1, 99]$. Evaluate on 100 instances of sizes $\{10\times5, 15\times10, 20\times10, 30\times10, 40\times10\}$ and on Brandimarte (mk) (Brandimarte, 1993)

and Hurink (edata, rdata, vdata) (Hurink et al., 1994). **JSP:** train 500 instances at $10 \times 5$ and $15 \times 10$ following Taillard (1993); evaluate on Taillard (Taillard, 1993) and Demirkol (Demirkol et al., 1998). Benchmark details are in App. C.

**Training dataset generation.** Offline RL trains on a fixed dataset $D$. We collect trajectories using three kind of heuristics: (i) Priority Dispatching Rules (**PDR**)—for FJSP, 4 job-selection $\times$ 4 machine-selection rules (16 trajectories per instance); for JSP, 4 job rules (machines fixed). (ii) Genetic Algorithms **GA** (Reijnen et al., 2023)—use the entire final population (typically higher quality, lower diversity than PDRs). (iii) **Random**—uniformly sample feasible actions. We build four datasets: **PDR** (16 FJSP / 4 JSP trajectories), **GA** (200 trajectories per instance), **PDR–GA** (union), and **Random** (100 trajectories per instance). These datasets matches the setup used in offline RL work (Fu et al., 2020), where datasets with different qualities are used. From each trajectory, we extract the transitions with which CDQAC and offline RL baselines are trained. Duplicate trajectories are removed before training; heuristic details are in App. D.

**Metrics.** We report the *optimality gap*: Gap $= \frac{C_{\max}^{j} - C_{\mathrm{ub}}}{C_{\mathrm{ub}}} \times 100$, which measures the difference between $C_{\max}^{j}$, the makespan found by method $j$, and $C_{\mathrm{ub}}$, which is the optimal or best-known makespan for the given instance. For generated instances, we used solutions generated by OR tools (Perron et al., 2023), with a solving time limit of 30 minutes per instance, as reported in (Wang et al., 2023). For the benchmark instances, Taillard, Demirkol, Brandimarte, and Hurink, we used the best known solutions noted in the literature [1].

**Baselines.** We benchmark CDQAC against both offline and online RL approaches and strong heuristics. Each learning-based policy is evaluated in two modes: **greedy** (argmax) and **sampling** (100 solutions sampled; best kept), averaging over three different evaluations seeds (1,2, 3).

(1) **Offline**: We compare CDQAC with Offline-LD (Remmerden et al., 2025), originally developed for JSP, Behavioral Cloning (BC), and Implicit Q-Learning (IQL) (Kostrikov et al., 2021). All baselines are adapted to FJSP by using DAN (Wang et al., 2023) as the encoder. For Offline-LD, we implement both variants—maskable QRDQN (mQRDQN) and discrete maskable SAC (d-mSAC)—as introduced in (Remmerden et al., 2025). Each method is trained separately on the four training datasets (PDR, GA, PDR-GA, Random) and three instance sizes ($10 \times 5$, $15 \times 10$, $20 \times 10$), as relative offline RL performance can vary substantially across datasets. Full implementation details are provided in App. E.

(2) **Online RL**: For FJSP, we compare with FJSP-DRL (Song et al., 2023) and DANIEL (Wang et al., 2023), both using PPO and trained on 1,000 instances with 20 runs each, relying on the results reported in their papers. We also include Residual (Ho et al., 2024), which uses REINFORCE with a custom baseline. We retrain Residual under the same protocol as FJSP-DRL and DANIEL (1,000 generated instances, 20 runs each) for sizes $10 \times 5$, $15 \times 10$, and $20 \times 10$. All three online baselines use generated instances from the same distribution as CDQAC and share the same validation set. We also compare against a Genetic Algorithm (GA), the two best-performing dispatching rules (MOR-SPT, MOR-EST), and CP (30-minute limit) on the benchmark instances. For JSP, we compare with L2D (Zhang et al., 2020) trained on 10,000 instances (4 runs), Offline-LD (Remmerden et al., 2025) trained on 100 noisy-expert solutions, as well as DANIEL and Residual. We include DANIEL and Residual because our focus is on RL methods applicable to both JSP and FJSP. The JSP results for DANIEL come from Reijnen et al. (2023), and we retrain Residual. CDQAC, DANIEL, and Residual are all trained on JSP instances of size $10 \times 5$. Both DANIEL and Residual use online training on 1,000 instances with 20 runs each. For JSP, we also include MOR and MWKR, both dispatching rules, as well as MIP and CP, exact solvers with a 30 minute time limit.

**Training Setup.** We evaluate the stability of CDQAC by running all experiments with four different seeds (1, 2, 3, 4). Although this is standard practice in offline RL (Fu et al., 2020), online RL methods for FJSP (Song et al., 2023; Wang et al., 2023) typically report results from a single seed. Consequently, we present mean and standard deviation for our offline RL comparisons, but

---

[1]The best known solutions for both Taillard and Demirkol can be found at `https://optimizizer.com/jobshop.php` and for Hurink (edata, rdata, vdata) and Brandimarte at `https://scheduleopt.github.io/benchmarks/fjsplib`

Table 1: Average gap (%) on all FJSP evaluation sets. $\pi_\beta$ best performance of heuristics that generated dataset. **Bold** is best result of the method (row) for each training dataset (column).

| | | PDR | GA | PDR-GA | Random |
|---|---|---|---|---|---|
| Greedy | BC | $29.13 \pm 3.2$ | $13.91 \pm 0.6$ | $22.37 \pm 2.24$ | $21.85 \pm 2.51$ |
| | Offline-LD (mQRDQN) | $22.26 \pm 2.43$ | $30.85 \pm 3.57$ | $21.80 \pm 3.64$ | $21.49 \pm 2.62$ |
| | Offline-LD (d-mSAC) | $23.28 \pm 3.06$ | $21.02 \pm 2.13$ | $25.94 \pm 2.29$ | $16.91 \pm 1.89$ |
| | IQL | $19.93 \pm 1.83$ | $20.66 \pm 2.18$ | $19.24 \pm 2.34$ | $21.34 \pm 3.54$ |
| | **CDQAC (Ours)** | $\mathbf{12.34 \pm 1.72}$ | $\mathbf{13.06 \pm 2.10}$ | $\mathbf{11.31 \pm 1.33}$ | $\mathbf{10.68 \pm 0.51}$ |
| Sampling | BC | $10.71 \pm 0.99$ | $8.3 \pm 0.15$ | $9.49 \pm 0.56$ | $13.15 \pm 0.09$ |
| | Offline-LD (mQRDQN) | $13.64 \pm 0.20$ | $14.26 \pm 0.26$ | $13.68 \pm 0.17$ | $13.63 \pm 0.23$ |
| | Offline-LD (d-mSAC) | $11.61 \pm 1.32$ | $8.83 \pm 0.69$ | $11.69 \pm 1.23$ | $7.79 \pm 0.86$ |
| | IQL | $10.01 \pm 0.58$ | $9.19 \pm 0.56$ | $9.48 \pm 0.59$ | $10.79 \pm 0.74$ |
| | **CDQAC (Ours)** | $\mathbf{6.57 \pm 0.76}$ | $\mathbf{6.43 \pm 0.87}$ | $\mathbf{5.87 \pm 0.51}$ | $\mathbf{5.86 \pm 0.30}$ |
| | $\pi_\beta$ | 14.13 | 6.74 | 6.74 | 28.16 |

only single seed results (seed 1) when comparing with online methods. We conducted experiments on servers equipped with a NVIDIA A100 GPU, Intel Xeon CPU, and 360GB of RAM. Detailed descriptions of the hyperparameters and the network architecture can be found in App. F.

## 5.1 COMPARISON WITH OFFLINE RL

We first compare CDQAC with the offline RL baselines Offline-LD, Implicit Q-learning (IQL) and Behavioral Cloning (BC), all implemented with a DAN network (Wang et al., 2023). This allows us to evaluate whether novel aspects of CDQAC, such as the delayed policy and the dueling quantile critic, contributed to the performance compared to offline baselines[2]. All methods are trained across all datasets, as each dataset serves as a distinct benchmark in offline RL; prior work has shown that the relative performance between methods trained on the same dataset can vary significantly between different qualities of the dataset (Figueiredo Prudencio et al., 2024). Table 1 shows that CDQAC outperforms both versions of Offline-LD by a significant margin. Furthermore, CDQAC consistently outperforms all heuristics that generated the datasets (denoted with $\pi_\beta$). In contrast, the other offline RL baselines, Offline-LD and IQL, never outperformed GA, or even the PDR heuristics with greedy evaluation. The second highest performance was achieved with BC, when trained on GA (Greedy: $13.91 \pm 0.6$, Sampling: $8.3 \pm 0.15$); however, BC still performed significantly worse than CDQAC, even when trained on the same GA dataset (Greedy: $13.06 \pm 2.1$, Sampling: $6.43 \pm 0.87$), on which CDQAC performed the worst. Additional results of our offline RL comparison are in App. H.2.

Both Offline-LD (d-mSAC) and CDQAC achieve the best performance when trained on the *Random dataset*. Offline-LD (d-mSAC) achieves gaps of $16.91\%\pm1.89\%$, $7.79\%\pm0.86\%$, while CDQAC achieves even better performance with gaps of $10.68\%\pm0.51\%$, $5.86\%\pm0.30\%$ for greedy and sampling, respectively. These results contradict prior offline RL work (Schweighofer et al., 2022; Kumar et al., 2022) where noisy-expert datasets typically outperform random datasets. Both CDQAC and Offline-LD only learn the state-action value in a dataset, we hypothesize that for such approaches a diverse suboptimal dataset is preferred, over a high-quality, but less diverse dataset with offline RL in FJSP.

**Why random solutions outperform expert data?** To empirically evaluate the diversity of a dataset, we use **State-Action Coverage** (SACo) (Schweighofer et al., 2022), defined as $\text{SACo}(D) = \frac{u_{s,a}(D)}{u_{s,a}(D_{\text{ref}})}$ where $u_{s,a}(D)$ denotes the number of unique state–action pairs observed in dataset $D$. We take *PDR* as the reference dataset, that is, $D_{\text{ref}} = PDR$, so by definition $\text{SACo}(PDR) = 1$.

Table 2 shows that *Random* has substantially higher state–action coverage. This ranking largely mirrors the main results in Table 1. Previous theoretical work on offline RL (Jin et al., 2021; Kumar et al., 2022) shows that

Table 2: The **State-Action Coverage** (SACo) of the FJSP training datasets of each instance size. PDR is the reference dataset, and a higher SACo is better.

| Instance Size | PDR | GA | PDR-GA | Random |
|---|---|---|---|---|
| $10 \times 5$ | $1 \pm 0$ | $3.13 \pm 0.38$ | $4.13 \pm 0.38$ | $\mathbf{8.46 \pm 0.71}$ |
| $15 \times 10$ | $1 \pm 0$ | $2.59 \pm 0.46$ | $3.59 \pm 0.46$ | $\mathbf{6.93 \pm 0.29}$ |
| $20 \times 10$ | $1 \pm 0$ | $3.16 \pm 0.4$ | $4.16 \pm 0.4$ | $\mathbf{7.7 \pm 0.18}$ |
| Average | $1 \pm 0$ | $2.96 \pm 0.49$ | $3.96 \pm 0.49$ | $\mathbf{7.7 \pm 0.77}$ |

---

[2]A full ablation study of each component can be found in App H.1.

Table 3: Results FJSP benchmarks sets. CDQAC trained on Random dataset; all models on $10\times5$ or $15\times10$ instances. **Bold** indicates best performance.

| | Method | mk | | edata | | rdata | | vdata | |
|---|---|---|---|---|---|---|---|---|---|
| | | Gap(%) | Time(s) | Gap(%) | Time(s) | Gap(%) | Time(s) | Gap(%) | Time(s) |
| **Greedy** 10×5 | FJSP-DRL | 28.52 | 1.26 | 15.53 | 1.4 | 11.15 | 1.4 | 4.25 | 1.37 |
| | Residual | 25.53 | 0.68 | 15.97 | 0.5 | 11.78 | 0.63 | 2.8 | 0.8 |
| | DANIEL | 13.58 | 1.29 | 16.33 | 1.37 | 11.42 | 1.37 | 3.28 | 1.37 |
| | **CDQAC (Ours)** | 13.04 | 1.1 | **13.86** | 1.18 | **10.10** | 1.18 | **2.75** | 1.18 |
| 15×10 | FJSP-DRL | 26.77 | 1.25 | 15 | 1.4 | 11.14 | 1.4 | 4.02 | 1.37 |
| | Residual | 25.22 | 0.68 | 16.99 | 0.5 | 11.19 | 0.62 | 4.04 | 0.79 |
| | DANIEL | 12.97 | 1.3 | 14.41 | 1.38 | 12.07 | 1.36 | 3.75 | 1.37 |
| | **CDQAC (Ours)** | **12.64** | 1.08 | 14.74 | 1.15 | 10.47 | 1.14 | 3.13 | 1.14 |
| **Sampling** 10×5 | FJSP-DRL | 18.56 | 4.13 | 8.17 | 4.91 | 5.57 | 4.81 | 1.32 | 4.71 |
| | Residual | 21.65 | 65.01 | 13.61 | 49.84 | 7.42 | 60.75 | 1.76 | 80.37 |
| | DANIEL | 9.53 | 4.12 | 9.08 | 4.71 | **4.95** | 4.73 | 0.69 | 4.77 |
| | **CDQAC (Ours)** | 8.96 | 3.36 | 9.4 | 3.82 | 5.59 | 3.84 | **0.65** | 3.84 |
| 15×10 | FJSP-DRL | 19 | 4.13 | 8.69 | 4.87 | 5.95 | 4.82 | 1.34 | 4.72 |
| | Residual | 19.91 | 66.09 | 11.94 | 50.61 | 8.25 | 61.52 | 1.58 | 77.59 |
| | DANIEL | 8.95 | 4.08 | 8.72 | 4.7 | 5.49 | 4.73 | 0.72 | 4.75 |
| | **CDQAC (Ours)** | **7.94** | 3.22 | **7.77** | 3.66 | 5.08 | 3.68 | 0.69 | 3.72 |
| | MOR-SPT | 25.67 | 0.1 | 17.75 | 0.11 | 14.38 | 0.1 | 6.06 | 0.11 |
| | MOR-EST | 29.59 | 0.1 | 17.59 | 0.11 | 14.3 | 0.1 | 5.59 | 0.11 |
| | GA | 14.29 | 232.95 | 4.55 | 237.06 | 4.43 | 243.91 | 0.67 | 283.97 |
| | CP | 1.5 | 1447 | 0 | 900 | 0.11 | 1397 | 0 | 639 |

diverse datasets can be more optimal than narrow expert datasets, given that the RL problem has a horizon of $H \geq 40$, while FJSP has a minimum horizon of $H = 50$ for $10 \times 5$, and increasing with larger instance sizes. A larger $H$ means that *Random* has enough transitions to "stitch" together an optimal policy, since it increases the likelihood of them occurring in the training dataset. Jin et al. (2021) highlights this explanation with *intrinsic uncertainty*, referring to how uncertain an offline RL method is depending on the absence of state–action pairs from the optimal policy in dataset $D$. This means that a higher SACo increases the probability that state-action pairs, done by an optimal policy, are present in the dataset. For example, *GA* and *PDR* alone have an intrinsic uncertainty greater than that of the union of them, *PDR-GA*, and *Random*. Moreover, a wider coverage, both in state action pairs and in solution quality, enables CDQAC to confirm *pessimism*, the CQL regression, resulting in more accurate learning of the returns (Jin et al., 2021; Kumar et al., 2022).

## 5.2 COMPARISON WITH ONLINE RL ON FJSP BENCHMARKS

In this set of experiments, we examined the performance difference between CDQAC and online RL approaches for FJSP. Table 3 shows that CDQAC outperforms both the online RL approaches FJSP-DRL (Song et al., 2023), Residual (Ho et al., 2024), and DANIEL (Wang et al., 2023), on all benchmark sets, except for the sampling evaluation of Hurink rdata, where DANIEL marginally outperforms CDQAC (Gaps 4.95% vs 5.08%). Moreover, Table 3 indicates that CDQAC mitigates distributional shift, since the benchmark instances have a different distribution than the instances on which CDQAC is trained.

For generated instances, Table 4 shows that CDQAC performs similarly to DANIEL (Wang et al., 2023) on $10 \times 5$, and outperforms DANIEL on $15 \times 10$. This suggests that CDQAC achieves similar performance to online RL approaches on evaluation sets that mirror the online RL's training distribution, to which online RL methods often become highly specialized or overfit during training. CDQAC achieves these results with only 500 instances, compared to FJSP-DRL (Song et al., 2023) and DANIEL (Wang et al., 2023) 1000 instances. Furthermore, Table 5 shows that CDQAC is able to generalize better to larger instances than DANIEL. With CDQAC's greedy evaluation matching $30 \times 10$ and outperforming $40 \times 10$ DANIEL's sampling evaluation.

Interestingly, CDQAC, trained with the *Random* dataset, can outperform online RL approaches, contrast the conclusions of prior work on offline RL (Fu et al., 2020; Fujimoto et al., 2019; Kumar et al., 2023), where online RL typically dominates. Although Remmerden et al. (2025) showed that Offline-LD outperformed its online counterpart L2D (Zhang et al., 2020), this was only achieved through an expert dataset generated with CP. In comparison, CDQAC can outperform other baselines through training on a random dataset. We attribute the performance of CDQAC to two factors: (1) the ability of CDQAC to learn an accurate representation $Z_\theta$ of the state action values in the training dataset. (2) CDQAC is an off-policy Q-learning-based method, non-standard for JSP or FJSP.

Table 4: Results generated FJSP evaluation instances. CDQAC trained on Random dataset; training instances size is same as evaluation instance size. **Bold** indicates best performance per evaluation mode.

Table 5: Generalization to large FJSP instances. CDQAC trained on Random dataset; training size 10×5. **Bold** indicates best performance per evaluation mode.

| Method | 10 × 5 | | 15 × 10 | | 20 × 10 | |
|---|---|---|---|---|---|---|
| | Gap(%) | Time(s) | Gap(%) | Time(s) | Gap(%) | Time(s) |
| FJSP-DRL (Greedy) | 16.03 | 0.45 | 16.33 | 1.43 | 10.15 | 1.91 |
| Residual | 15.23 | 0.27 | 15.93 | 0.85 | 10.01 | 1.28 |
| DANIEL | **10.87** | 0.45 | 12.42 | 1.35 | **1.31** | 1.85 |
| **CDQAC (Ours)** | 11.56 | 0.39 | **11.1** | 1.16 | 4.34 | 1.56 |
| FJSP-DRL (Sampling) | 9.66 | 1.11 | 12.13 | 3.98 | 9.64 | 6.23 |
| Residual | 9.85 | 27.04 | 12.38 | 77.5 | 9.81 | 116.41 |
| DANIEL | **5.57** | 0.74 | 6.79 | 3.89 | **-1.03** | 6.35 |
| **CDQAC (Ours)** | 5.98 | 0.64 | **5.85** | 3.06 | 1.79 | 4.83 |
| MOR-SPT | 19.67 | 0.03 | 17.89 | 0.1 | 11.25 | 0.15 |
| MOR-EST | 19.66 | 0.03 | 19.98 | 0.1 | 12.08 | 0.14 |
| GA | 6.0 | 71.65 | 10.42 | 266.15 | 6.78 | 348.87 |

| Method | 30 × 10 | | 40 × 10 | |
|---|---|---|---|---|
| | Gap(%) | Time(s) | Gap(%) | Time(s) |
| FJSP-DRL (Greedy 10 × 5) | 14.61 | 2.86 | 14.21 | 3.82 |
| Residual | 13.16 | 2.11 | 12.82 | 3.1 |
| DANIEL | 5.1 | 2.78 | 3.65 | 3.77 |
| **CDQAC (Ours)** | **4.43** | 2.32 | **3.17** | 3.19 |
| FJSP-DRL (Sampling 10 × 5) | 12.36 | 12.79 | 12.26 | 24.54 |
| Residual | 12.94 | 213.89 | 12.85 | 319.69 |
| DANIEL | 4.43 | 12.37 | 3.77 | 22.58 |
| **CDQAC (Ours)** | **3.11** | 9.57 | **2.21** | 16.01 |
| MOR-SPT | 14.99 | 0.23 | 14.57 | 0.33 |
| MOR-EST | 15.88 | 0.22 | 15.17 | 0.32 |
| GA | 11.26 | 521.19 | 11.26 | 736.36 |

Table 6: Results JSP benchmarks. Average gap (%) is reported. CDQAC trained on Random dataset for 10 × 5. For DANIEL Wang et al. (2023), only Tailard was reported. **Bold** indicates best result.

| | | Greedy | | | | | | | Sampling | | | Exact | |
|---|---|---|---|---|---|---|---|---|---|---|---|---|---|
| | Instance Size | MWR | MOR | L2D | Offline-LD | DANIEL | Residual | **CDQAC (Ours)** | DANIEL | Residual | **CDQAC (Ours)** | MIP | CP |
| Tailard | 15 × 15 | 18.9 | 21.4 | 28.1 | 25.8 | 19.0 | 17.6 | **15.0** | 13.2 | 13.3 | **10.4** | 0.1 | 0.1 |
| | 20 × 15 | 23.0 | 23.6 | 32.7 | 30.2 | 22.1 | 21.2 | **17.7** | 17.4 | 16.1 | **13.2** | 3.2 | 0.2 |
| | 20 × 20 | 21.6 | 21.7 | 31.8 | 28.9 | 18.0 | 18.0 | **17.6** | 13.3 | 15.8 | **12.9** | 2.9 | 0.7 |
| | 30 × 15 | 24.3 | 23.2 | 30.2 | 29.2 | 21.7 | 20.1 | **19.1** | 17.2 | 18.0 | **14.9** | 10.7 | 2.1 |
| | 30 × 20 | 24.8 | 25.0 | 35.2 | 33.1 | 23.2 | 22.3 | **21.2** | 19.0 | 19.7 | **17.9** | 13.2 | 2.8 |
| | 50 × 15 | 16.5 | 17.3 | 21.0 | 20.6 | 14.8 | 15.6 | **13.0** | 12.7 | 13.2 | **9.9** | 12.2 | 3.0 |
| | 50 × 20 | 18.1 | 17.9 | 26.1 | 24.3 | 16.0 | 14.4 | **12.8** | 13.1 | 14.1 | **11.0** | 13.6 | 2.8 |
| | 100 × 20 | 8.3 | 9.1 | 13.3 | 12.7 | 7.3 | 6.5 | **5.3** | 5.9 | 6.5 | **3.6** | 11.0 | 3.9 |
| | Mean | 19.4 | 19.9 | 27.3 | 25.6 | 18.2 | 17.0 | **15.2** | 14.4 | 14.6 | **11.7** | 8.4 | 2.0 |
| Demirkol | 20 × 15 | 27.8 | 30.3 | 36.3 | 35.8 | – | 26.1 | **22.9** | – | 22.6 | **18.4** | 5.3 | 1.8 |
| | 20 × 20 | 26.8 | 26.9 | 34.4 | 32.8 | – | 21.5 | **20.3** | – | 18.9 | **16.5** | 4.7 | 1.9 |
| | 30 × 15 | 31.9 | 36.4 | 37.8 | 38.8 | – | 27.6 | **27.1** | – | 29.4 | **23.1** | 14.2 | 2.5 |
| | 30 × 20 | 31.9 | 33.7 | 38.0 | 36.0 | – | 29.9 | **27.9** | – | 28.3 | **23.4** | 16.7 | 4.4 |
| | 40 × 15 | 26.5 | 35.5 | 34.6 | 35.5 | – | 26.2 | **25.5** | – | 28.4 | **20.2** | 16.3 | 4.1 |
| | 40 × 20 | 32.0 | 35.9 | 39.2 | 38.5 | – | **27.7** | 27.9 | – | 30.9 | **24.1** | 22.5 | 4.6 |
| | 50 × 15 | 27.3 | 34.8 | 33.2 | 34.1 | – | 27.4 | **25.0** | – | 29.5 | **21.7** | 14.9 | 3.8 |
| | 50 × 20 | 29.9 | 36.5 | 37.7 | 38.9 | – | 30.0 | **28.6** | – | 32.8 | **25.1** | 22.5 | 4.8 |
| | Mean | 29.2 | 33.7 | 36.4 | 36.3 | – | 27.0 | **25.7** | – | 27.6 | **21.6** | 14.6 | 3.5 |

Since CDQAC is an off-policy method, it allows CDQAC to reuse all training examples, whereas PPO and REINFORCE will only use the most recent examples. Therefore, DANIEL, Residual and FJSP-DRL will focus more on exploiting the training distribution, while CDQAC is able to generalize better to different distributions, as seen in Tables 3 and 5, where CDQAC trained on 10 × 5 outperforms DANIEL, when also trained on 10 × 5, although DANIEL outperforms CDQAC on the same distribution instances 10 × 5 (Table 4). We have included a convergence analysis between DANIEL and CDQAC in App I.

Although direct training on large FJSP instances such as 20 × 10 presents additional challenges due to the size of the action space, with the action space growing to at most 200 actions. App. G shows this is related to training, since CDQAC is able to converge stabile for both 10 × 5 and 15 × 10 in all training datasets, but not for 20 × 10. Yet, this limitation does not affect the ability of CDQAC to generalize. In fact, when trained on smaller instances, CDQAC outperforms DANIEL on larger unseen instances (e.g. 30 × 10 and 40 × 10 in Table 5), suggesting that CDQAC is better able to generalize to larger unseen instances than DANIEL, and that CDQAC mitigates distributional shift to larger, unseen instance sizes. Moreover, these promising results highlight future research direction for addressing training challenges in large action spaces, such as factorized action spaces (Beeson et al., 2024) or stochastic Q-learning (Fourati et al., 2024).

## 5.3 COMPARISON ON JSP INSTANCES

In the JSP evaluation, we assess whether CDQAC attains performance on JSP comparable to its results on FJSP. To this end, we benchmark CDQAC against online RL methods that operate on both JSP and FJSP, namely, Residual (Ho et al., 2024) and DANIEL (Wang et al., 2023), with both

methods retrained on JSP. We additionally include Offline-LD (Remmerden et al., 2025), an offline RL baseline for JSP, as well as L2D (Zhang et al., 2020), which was part of the comparison in Remmerden et al. (2025). An extended set of JSP results is provided in App. H.5, where we also compare against JSP-specific learning-based approaches that do not function on FJSP.

Table 6 shows that CDQAC, trained solely on the Random dataset, surpasses all online RL baselines (Residual, DANIEL, and L2D) as well as the offline RL method Offline-LD. The gap relative to Offline-LD is particularly striking: despite Offline-LD being trained on expert demonstrations generated by CP, CDQAC, trained only on random data, achieves substantially better performance, highlighting the strength of CDQAC.

CDQAC also consistently outperforms both Residual and DANIEL. On the Taillard instances, CDQAC achieves gaps of 15.2% (greedy) and 11.7% (sampling), compared to 18.2% and 14.4% for DANIEL and 17.0% and 14.6% for Residual. On the Demirkol benchmark, CDQAC (25.7% greedy, 21.6% sampling) similarly improves over Residual (27.0% greedy, 27.6% sampling). These findings indicate that CDQAC is more effective for JSP than both online RL baselines.

Finally, Table 6 shows that CDQAC demonstrates favorable scaling on large Taillard instances. Its sampling evaluation outperforms MIP on $50 \times 15$ and $50 \times 20$, and exceeds both MIP and CP on $100 \times 20$. This suggests that CDQAC scales to larger JSP problem sizes more effectively than exact solvers.

### 5.4 PERFORMANCE WITH REDUCED TRAINING DATA

To test the sample efficiency of CDQAC, we evaluated CDQAC by reducing the number of instances in the Random training dataset. Fig. 3 shows that increasing the size of the dataset has only a marginal positive effect on performance. We noticed the greatest performance difference for $10 \times 5$ between 5 instances (greedy 11.8%) and 10 instances (greedy 10.5%), whereas other results show no significant difference. Importantly, even a small number of Random trajectories maintains high state–action diversity (Table 2). Furthermore, as an off-policy, bootstrapped method, CDQAC continually refines the target for each transition through the dueling quantile critic. Thus, each replayed transition provides a progressively more informative learning signal, enabling CDQAC to extract significantly more value from limited data. In conclusion, we see that CDQAC needs only a fraction of the original dataset (1%

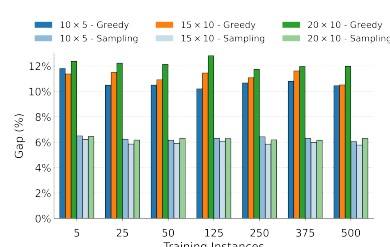

Figure 3: Results of reducing the number of instances in the Random dataset, evaluated on FJSP benchmarks Hurink and Brandimarte.

to 5%) to achieve performance similar to the full dataset, and significantly less than online RL approaches (Song et al., 2023; Wang et al., 2023), requiring up to a 1000 instances. We have included extended results in App. H.4.

## 6 CONCLUSION

This paper introduced **Conservative Discrete Quantile Actor-Critic**, a novel offline RL algorithm for JSP and FJSP. To our knowledge, CDQAC is the first offline RL for both JSP and FJSP that trains fully on suboptimal data, while being able to outperform strong online RL baselines, contradicting prior work in offline RL. CDQAC achieves this by learning an accurate representation of the returns of a possible scheduling action from a static dataset, enabling CDQAC to "stitch" together high-quality partial solutions to learn a new policy. CDQAC also generalized well from small to larger instance sizes.

Offline RL remains underexplored in scheduling and, more broadly, in combinatorial optimization problems. In this work, we demonstrate that offline RL can be highly competitive in learning effective heuristics for complex scheduling tasks. In future work, we plan to extend our approach to other combinatorial optimization problems. Future research could extend CDQAC to real-world scheduling, for which building a simulated environment is infeasible but has suboptimal training data generated by heuristics.

## REPRODUCIBILITY STATEMENT

All experimental settings, datasets, and evaluation are specified in the main text and in the appendices. We detail instance generation and benchmark details for FJSP and JSP (sizes, processing time ranges, and evaluation sets) in Sect. 5 and App. C–D, including how PDR/GA/Random datasets are generated. We note all the seeds used in our experiments in Sect. 5. Complete hyperparameters for CDQAC and baselines (optimizer, learning rates, quantile bins, CQL coefficient, policy update frequency, network sizes, batch size, and training steps) are given in App. F, Table 8. Hardware details (NVIDIA A100 GPU, Intel Xeon CPU, 360 GB RAM) and evaluation modes (greedy vs. sampling with 100 samples) are also specified. We will release our code for CDQAC (training and evaluation), datasets, and dataset generators for PDR/GA/Random on Github upon acceptance; hyperparameter and seed configurations match those in App. F. This should allow researchers to replicate our experiments and results.

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

## A  PSEUDOCODE

---

**Algorithm 1** Training Procedure of CDQAC

---

**Require:** Dataset $D$, batch size $B$, policy update frequency $\eta$, total training steps $T$, CQL coefficient $\alpha_{\text{CQL}}$, entropy coefficient $\lambda$, target update rate $\rho$, learning rates $\ell_\psi, \ell_\theta$

**Ensure:** Initialized policy network $\psi$, critic network $\theta$, target network $\hat{\theta} \leftarrow \theta$

1: **for** $t = 1$ to $T$ **do**
2:     Sample mini-batch $\{(s_i, a_i, r_i, s_i')\}_{i=1}^B \sim D$
3:     Compute target quantiles: $\mathcal{T}Z_i \leftarrow r_i + \gamma Z_{\hat{\theta}}(s_i', a_i')$ where $a_i' \sim \pi_\psi(\cdot \mid s_i')$
4:     Compute TD loss: $\mathcal{L}_{\text{TD}}(\theta) \leftarrow \frac{1}{B} \sum_{i=1}^B \sum_{j=1}^N \rho_{\tau_j}^H(\mathcal{T}Z_i - Z_\theta(s_i, a_i))$
5:     Compute conservative critic loss:

$$
\mathcal{L}_Z(\theta) \leftarrow \frac{1}{B} \sum_{i=1}^B \left[ \log \sum_{a' \in \mathcal{A}(s_i)} \exp(Q_\theta^Z(s_i, a')) - Q_\theta^Z(s_i, a_i) \right] + \mathcal{L}_{\text{TD}}(\theta)
$$

6:     Update critic: $\theta \leftarrow \theta + \ell_\theta \nabla_\theta \mathcal{L}_Z(\theta)$
7:     **if** $t \bmod \eta = 0$ **then**
8:         Compute policy loss:

$$
\mathcal{L}_\pi(\psi) \leftarrow \frac{1}{B} \sum_{i=1}^B \left[ \sum_{a \in \mathcal{A}(s_i)} -Q_\theta^Z(s_i, a)\pi_\psi(a \mid s_i) + \lambda\mathcal{H}[\pi_\psi(\cdot \mid s_i)] \right]
$$

9:         Update policy: $\psi \leftarrow \psi + \ell_\psi \nabla_\psi \mathcal{L}_\pi(\psi)$
10:     **end if**
11:     Update target network: $\hat{\theta} \leftarrow (1 - \rho)\hat{\theta} + \rho\theta$
12: **end for**

---

Algorithm 1 shows the training process of CDQAC. In it, we train CDQAC using a static dataset $D = (s, a, r, s')$ of scheduling transitions. At each training step, we sample a mini-batch of $B$ transitions from $D$. For each transition, we compute the target $\mathcal{T}Z = r + \gamma Z_{\hat{\theta}}(s', a')$ using the target network $\hat{\theta}$ and next actions $a' \sim \pi_\psi(\cdot \mid s')$ drawn from the current policy. The critic is optimized through a conservative quantile-based objective, combining the temporal difference (TD) loss $\mathcal{L}_{\text{TD}}$ (Eq. 3) with a CQL penalty that discourages overestimation of out-of-distribution actions (Eq. 4). The critic parameters $\theta$ are updated via gradient descent on the combined loss $\mathcal{L}_Z$.

To stabilize training, we employ a delayed policy update strategy: the actor $\pi_\psi$ is updated every $\eta$ steps by minimizing the Q-learning objective (Eq. 5), with the entropy bonus $\mathcal{H}[\pi_\psi(\cdot \mid s)]$. The policy update relies on the scalarized quantile values $Q_\theta^Z(s, a) = \mathbb{E}[Z_\theta(s, a)]$, where $Z_\theta$ is the minimum of two dueling quantile networks. Finally, the target network is updated using Polyak averaging: $\hat{\theta} \leftarrow (1 - \rho)\hat{\theta} + \rho\theta$.

## B  NETWORK ARCHITECTURE

The dual attention network (Wang et al., 2023) (DAN) is an attention-based network architecture for JSP and FJSP that encodes the operation features $h_{O_{i,j}}^{(L)}$, and machine features $h_{M_k}^{(L)}$, where $L$ presents the current layer input, so $L = 1$ is the input features. DAN is able to learn the complex relation between each operation $O_{i,j}$ and each compatible machine $M_k$, through separate *operation attention blocks* and *machine attention blocks* as seen in Fig. 2 in Sect. 4.1. In this section, we provide an overview of each attention block, and their interaction. Afterwards, we state the features used for the operations, machines and machine-operation pairs.

**Operation Attention Block.**  To capture the sequential nature of operations within jobs, the operation attention blocks attend each operation $O_{i,j}$ in the context of its predecessor $O_{i,j-1}$ and

successor $O_{i,j+1}$, if they exist. An attention coefficient is calculated between these operations:

$$a_{i,j,p} = \text{Softmax}\left(\text{LeakyReLU}\left(\mathbf{V}^T \left[\left(\mathbf{W}h_{O_{i,j}}^{(L)} \parallel \mathbf{W}h_{O_{i,p}}^{(L)}\right)\right]\right)\right), \tag{7}$$

where $\mathbf{W}$, and $\mathbf{V}$ are learned projections. The attention coefficient $a_{i,j,p}$, calculated in Eq. 7, is used to calculate the output of the operation attention block as follows:

$$h_{O_{i,j}}^{(L+1)} = \sigma\left(\sum_{p=j-1}^{j+1} a_{i,j,p}\mathbf{W}h_{O_{i,p}}^{(L)}\right), \tag{8}$$

where $\sigma$ is an activation function. The operation blocks in DAN (Wang et al., 2023) function similar to a GNN, in that information, one by one, is propagated through the operations.

**Machine Attention Block.** The machine attention block considers the relationship between two machines $M_y \in \mathcal{M}_t$ and $M_z \in \mathcal{M}_t$ in relation to the set of unscheduled operations $\hat{O}_{y,z}$ that can be processed by either $M_y$ or $M_z$. The embedding of the pooled operation is calculated as $h_{\hat{O}_{y,z}}^{(L)} = \frac{1}{|\hat{O}_{y,z}|}\sum_{O_{i,j} \in \hat{O}_{y,z} \cap \mathcal{O}_c} h_{O_{i,j}}^{(L)}$, where $\mathcal{O}_c$ represents the current operations available to schedule. The attention in this block is calculated through:

$$u_{y,z} = \text{Softmax}\left(\text{LeakyReLU}\left(\mathbf{X}\left[(\mathbf{Y}h_{M_y}^{(L)}) \parallel (\mathbf{Y}h_{M_z}^{(L)}) \parallel (\mathbf{Z}h_{\hat{O}_{y,z}}^{(L)})\right]\right)\right) \tag{9}$$

where $\mathbf{X}$, $\mathbf{Y}$, and $\mathbf{Z}$ are linear projections. Whenever two machines $M_y$ and $M_z$ do not share any operations in the current candidate set $\hat{O}_{y,z} \cap J_c = \emptyset$, we set the attention $u_{y,z}$ to zero. The output of the machine operation block is calculated as:

$$h_{M_k}^{(L+1)} = \sigma\left(\sum_{q \in \mathcal{N}_k} u_{k,q}\mathbf{Y}h_{M_q}^{(L)}\right), \tag{10}$$

where $\mathcal{N}_k$ is the set of machines, for which $M_k$ shares operations, including $M_k$ itself.

Lastly, DAN (Wang et al., 2023) uses a multihead attention approach, whereby each operation attention and machine attention block consist of $H$ heads. The results of the $H$ heads can be concatenated or averaged. Following the prior work of Wang et al. (2023), we concatenate the heads for each layer, except the last layer, which was averaged over the $H$ heads. We use ELU as our activation function for both operation and machine attention blocks.

### B.1 FEATURES

Table 7 shows the features used in our paper, based on the prior work of Wang et al. (2023). Both the machine features $M_k$ and the operation features $O_{i,j}$ are embedded using the DAN network. These embeddings, with the machine-operation pair $(O_{i,j}, M_k)$ features are used as input for the quantile critic and actor networks. In Table 7, we introduce the notation $\mathcal{O}_k$, which represents all operations $O_{i,j} \in \mathcal{O}_k$ that $M_k$ can process.

## C BENCHMARK INSTANCE SETS

As described in Sect.5, we evaluate our approach on generated instance sets as well as four established benchmark sets. For FJSP, we use the generated evaluation instances, the Brandimarte (mk) benchmark (Brandimarte, 1993) and the Hurink benchmark (Hurink et al., 1994), which includes the edata, rdata, and vdata subsets. For JSP, we evaluate on the Taillard (Taillard, 1993) and Demirkol (Demirkol et al., 1998) benchmarks. For each benchmark, we report the range of processing times, number of jobs, number of machines, and, specifically for FJSP, the number of machines available per operation.

Table 7: Features used by CDQAC, separated by operation $O_{i,j}$, machine $M_k$, and machine-operation pair $(O_{i,j}, M_k)$.

| Feature | Description |
|---|---|
| **Operation Features $O_{i,j}$** | |
| Min. proc. time | $\min_{M_k \in \mathcal{M}_{i,j}} p_{i,j}^k$ |
| Mean proc. time | $\frac{1}{|\mathcal{M}_{i,j}|} \sum_{M_k \in \mathcal{M}_{i,j}} p_{i,j}^k$ |
| Span proc. time | $\max_{M_k \in \mathcal{M}_{i,j}} p_{i,j}^k - \min_{M_k \in \mathcal{M}_{i,j}} p_{i,j}^k$ |
| Compatibility ratio | $\frac{|\mathcal{M}_{i,j}|}{|\mathcal{M}|}$ |
| Scheduled | 1 if scheduled, 0 otherwise |
| Estimated LB | Estimated lower bound completion time $C(O_{i,j})$ |
| Remaining ops $J_i$ | Number of unscheduled operations in $J_i$ |
| Remaining proc. time $J_i$ | Total proc. time of unscheduled operations in $J_i$ |
| Waiting time | Time since $O_{i,j}$ became available |
| Remaining proc. time | Remaining processing time (0 if not started) |
| **Machine Features $M_k$** | |
| Min. proc. time | $\min_{O_{i,j} \in \mathcal{O}_k} p_{i,j}^k$ |
| Mean proc. time | $\frac{1}{|\mathcal{O}_k|} \sum_{O_{i,j} \in \mathcal{O}_k} p_{i,j}^k$ |
| Total unscheduled ops | $|\mathcal{O}_k|$ |
| Schedulable ops at $t$ | # of ops schedulable at timestep $t$ |
| Free time | Time until $M_k$ becomes available |
| Waiting time | 0 if $M_k$ is working |
| Working status | 1 if working, 0 otherwise |
| Remaining proc. time | Time left on current task (0 if idle) |
| **Machine-Operation Pair $(O_{i,j}, M_k)$** | |
| Processing time | $p_{i,j}^k$ |
| Ratio to max of $O_{i,j}$ | $\frac{p_{i,j}^k}{\max_{M_k} p_{i,j}^k}$ |
| Ratio to max schedulable on $M_k$ | $\frac{p_{i,j}^k}{\max p_{i,j}^k \in \mathcal{O}_k(t)}$ |
| Ratio to global max | $\frac{p_{i,j}^k}{\max p_{i,j}^k \in \mathcal{O}}$ |
| Ratio to $M_k$'s unscheduled max | $\frac{p_{i,j}^k}{\max p_{i,j}^k \in \mathcal{O}_k}$ |
| Ratio to compatible max | $\frac{p_{i,j}^k}{\max p_{i,j}^k \in \mathcal{M}_{i,j}}$ |
| Ratio to $J_i$ workload | $\frac{p_{i,j}^k}{\sum p_{i,j} \in J_i}$ |
| Joint waiting time | Sum of $O_{i,j}$ and $M_k$ waiting times |

## C.1 FJSP

**Generated Evaluation Instances.** We generated 100 instances for each of the following sizes: $10 \times 5, 15 \times 10, 20 \times 10, 30 \times 10, 40 \times 10$, using the same generation procedure as for the training data (Sect. 5). Each operation is assigned between 1 and $|\mathcal{M}|$ available machines, selected uniformly at random.

**Brandimarte (mk) Benchmark.** The Brandimarte benchmark (Brandimarte, 1993) comprises 10 instances, each with 10 to 20 jobs and 4 to 15 machines. Processing times range from 1 to 19. The average number of machines available per operation ranges from 1.4 to 4.1, depending on the instance.

**Hurink Benchmark.** The Hurink benchmark (Hurink et al., 1994) consists of three subsets, edata, rdata, and vdata, each containing 40 instances. These subsets vary in degree of flexibility, with edata providing the lowest and vdata the highest average number of machines per operation. All instances include between 7 and 30 jobs and between 4 and 15 machines, with processing times between 5 and 99. The average number of machines available per operation is as follows:

- **edata:** Between 1.13 and 1.2.
- **rdata:** Between 1.88 and 2.06.
- **vdata:** Between 2.38 and 6.7.

## C.2 JSP

**Taillard Benchmark.** The Taillard benchmark (Taillard, 1993) contains 80 instances, ranging from $15 \times 15$ to $100 \times 20$. Processing times range between 1 and 99. These instances are similar to those used to train CDQAC.

**Demirkol Benchmark.** The Demirkol benchmark (Demirkol et al., 1998) includes 80 instances, with instance sizes ranging from $20 \times 15$ to $80 \times 20$. Processing times range from 1 to 200, twice the maximum value found in Taillard and CDQAC's training data.

# D    DETAILS OF DATASET GENERATION HEURISTICS

Our experimental setup in Sect. 5 stated that we used three types of heuristics to generate our training datasets, namely, priority dispatching rules (PDR), genetic algorithms (GA) and a random policy. We will now give a detailed explanation of each heuristic, and, in the case of GA, the hyperparameters.

## D.1    PRIORITY DISPATCHING RULES (PDR)

For the priority dispatching rules (PDR), we have separate rules for the selection of *jobs* and *machines* for FJSP. In our setup, first, a job $J_i \in \mathcal{J}$ is selected by the job selection rule. This job selection rule selects a job based on a specific rule, in which it is checked if there are still operations in $J_i$ to be scheduled. The machine selection rule selects the machine $M_k \in \mathcal{M}_{i,j}$ for operation $O_{i,j} \in J_i$, where $O_{i,j}$ is the current operation in $J_i$ that needs to be scheduled. For JSP, we only considered the job selection rules, since only one machine is ever available per operation. Furthermore, both the job and machine selection rules follow the MDP formulation, stated in Sect. 3, by which operation $O_{i,j}$ can only be scheduled on $M_k$, if it is free at timestep $t$. In the following, we give an overview of the job selection rules and the machine selection rules.

**Job selection rules.** We utilized four different job selection rules, namely, *Most Operations Remaining* (MOR), *Least Operations Remaining* (LOR), *Most Work Remaining* (MWR), and *Least Work Remaining* (LWR). Both MOR and LOR decide on the basis of the number of unscheduled operations in a job $J_i$. MOR selects the job with the most operations and LOR selects the job with the least operations to be scheduled. MWR and LWR focus on the remaining total processing times, a.k.a. the summation of processing times in a $J_i$, whereby we average the processing times of the available machines $M_k \in \mathcal{M}_{i,j}$. MWR selects the job with the highest total remaining processing times, whereas LWR selects the job with the least.

**Machine selection rules.** We considered four different machine selection rules, namely, *Shortest Processing Time* (SPT), *Longest Processing Time* (LPT), *Earliest Start Time* (EST), and *Latest Start Time* (LST). Both SPT and LPT select a machine $M_k \in \mathcal{M}_{i,j}$ for operation $O_{i,j}$ based on the processing time, with SPT selecting the machine with the shortest and LPT with the longest. EST and LST consider how long a machine $M_k$ is already free, with EST selecting the machine that is free the shortest, and LST the longest.

## D.2    GENETIC ALGORITHMS (GA)

For our genetic algorithm (GA), we used the implementation of Reijnen et al. (2023), whereby we introduced the constraint that $O_{i,j}$ can only be scheduled if machine $M_k$ is free at that time. This results in a more tight solution, with no gaps. Furthermore, we used a population size of 200, and ran the GA for 100 generations. The crossover probability was set at 0.7, and the mutation probability at 0.2.

## D.3    RANDOM POLICY

The random policy adheres to the MDP introduced in Sect. 3. This means that the random policy selects a random machine-operation pair based on those available at the time step $t$. The random policy can only select a machine-operation pair, if it can be scheduled at timestep $t$.

# E    DETAILS OF OFFLINE REINFORCEMENT LEARNING BASELINES

For our comparison of CDQAC to Offline-LD (Remmerden et al., 2025) in Sect. 5.1, we adapted both versions of it, namely, Offline-LD with a maskable Quantile Regression DQN (mQRDQN) and

with a discrete maskable Soft Actor-Critic (d-mSAC), using a dual attention network (Wang et al., 2023), such that both versions of Offline-LD used the same encoding as our introduced CDQAC approach. We provide a brief explanation of our implementations of each method, in which we state the hyperparameters used for each. If a hyperparameter is not stated, it is the same as CDQAC, as stated in App. F.

**Offline-LD (mQRDQN).** The mQRDQN version of Offline-LD is implemented identically as described by Remmerden et al. (2025). The hyperparameters are identical to CDQAC, whereby we set $\ell_\theta = 2 \times 10^{-4}$. In the original implemented of Offline-LD (mQRDQN) was not able to sample actions; therefore, for the sampling evaluation, we use Boltzmann sampling.

**Offline-LD (d-mSAC).** For d-mSAC version of Offline-LD, we implemented both the policy network and the Q network with a separate dual attention network (Wang et al., 2023) for each. We used the hyperparameters as with CDQAC, except for $\alpha_{\text{CQL}}$, which we set to $\alpha_{\text{CQL}} = 0.1$, and the target entropy of d-mSAC, which we set to 0.3. During initial testing, we found that this increased stability and performance with d-mSAC.

**Implicit Q-learning.** The main difference between Implicit Q-learning (IQL) (Kostrikov et al., 2021) and Offline-LD and CDQAC is that IQL constrains training by not using OOD actions, whereas Offline-LD and CDQAC regularize the Q-values of OOD actions during training to prevent overestimation. IQL consists of three networks, a policy, a value, and a Q network. Two hyperparameters of IQL are important to mention, namely $\beta_{\text{IQL}}$ and $\tau_{\text{IQL}}$. Firstly, $\beta_{\text{IQL}}$ controls how much the policy should learn to "exploit" the learned Q-values, or if it should stay close to the behavior found in the dataset, with $\beta_{\text{IQL}} = 0$, being equal to behavioral cloning. We decided, due to the suboptimality of our training datasets, to set $\beta_{\text{IQL}} = 15$. $\tau_{\text{IQL}}$ controls how much IQL should focus on positive examples, whereby $\tau_{\text{IQL}} = 0.5$ is equal to a SARSA update. Kostrikov et al. (2021) reported settings between 0.7 and 0.9 for $\tau_{\text{IQL}}$. We therefore tested 0.7, 0.8 and 0.9 to identify the ideal value and found that $\tau_{\text{IQL}} = 0.7$ result in the most stable updates. We set all learning rates at $2 \times 10^{-4}$, by which we also tested $2 \times 10^{-5}$; however, we found that this did not produce good results.

**Behavioral Cloning** Behavioral Cloning (BC) learns to imitate the behavioral policy $\pi_\beta$, which generated the training dataset. The BC loss is the cross-entropy loss between the predicted action for each state and the action found in the dataset. BC only trains a policy network and does not use a critic. All hyperparameters are the same as CDQAC (Table 8).

## F  HYPERPARAMETERS

In Table 8, we state the hyperparameters used in all our experiments. Furthermore, we used two layers of the DAN network, whereby we concatenated the output of each head for the first layer and averaged the heads for the second layer. Both the value stream $V_\theta$ and the advantage stream $A_\theta$, consist of three layers, each having 64 neurons. For each seed, we train for $200,000$ steps, with a batch size of 256. We normalize all features in the training dataset. We used ADAM (Kingma, 2014) optimizer.

## G  TRAINING PLOTS

Fig. 4, Fig. 5, and Fig 6 show the training plots for all the methods used in our FJSP evaluations (Tables 1, 10, 11, and 12). In the figure, we note that CDQAC converges in significantly fewer steps than the 200,000 training steps used. For example, for $10 \times 5$ CDQAC requires around 10,000 steps according to Fig. 4, and around 25,000 training steps for $15 \times 10$ as seen in Fig. 5.

Based on the training plots, we can determine that CDQAC achieves the most stable training with the highest average Makespan in each evaluation step. The only exception is with the Random dataset for $20 \times 10$ (Fig. 6d), where both Offline-LD (d-mSAC) and IQL are more stable and have a higher evaluation at the last training step. However, CDQAC for all other datasets and training datasets. For example, Offline-LD (d-mSAC) cannot learn a policy with the PDR-GA dataset for

Table 8: Hyperparameter settings CDQAC.

| Hyperparameter | Value |
|---|---|
| Policy Frequency Update $\eta$ | 4 |
| CQL Strength $\alpha_{\text{CQL}}$ | 0.05 |
| Number of quantile fractions $N$ | 64 |
| Learning rate quantile critic $\ell_\theta$ | $2 \times 10^{-4}$ |
| Learning rate policy $\ell_\psi$ | $2 \times 10^{-5}$ |
| Target Update Frequency $\rho$ | 0.005 |
| Entropy Coefficient $\lambda$ | 0.005 |
| Batch Size | 256 |
| Training Steps | 200,000 |
| Network Parameters | |
| Layers DAN network | 2 |
| Output Dimension DAN | (32, 8) |
| Number of Heads $H$ | 4 |
| Hidden Dimension Quantile Critic $Z_\theta$ | 64 |
| Hidden Layers Quantile Critic $Z_\theta$ | 2 |
| Hidden Dimension Policy $\pi_\psi$ | 64 |
| Hidden Layers Policy $\pi_\psi$ | 2 |

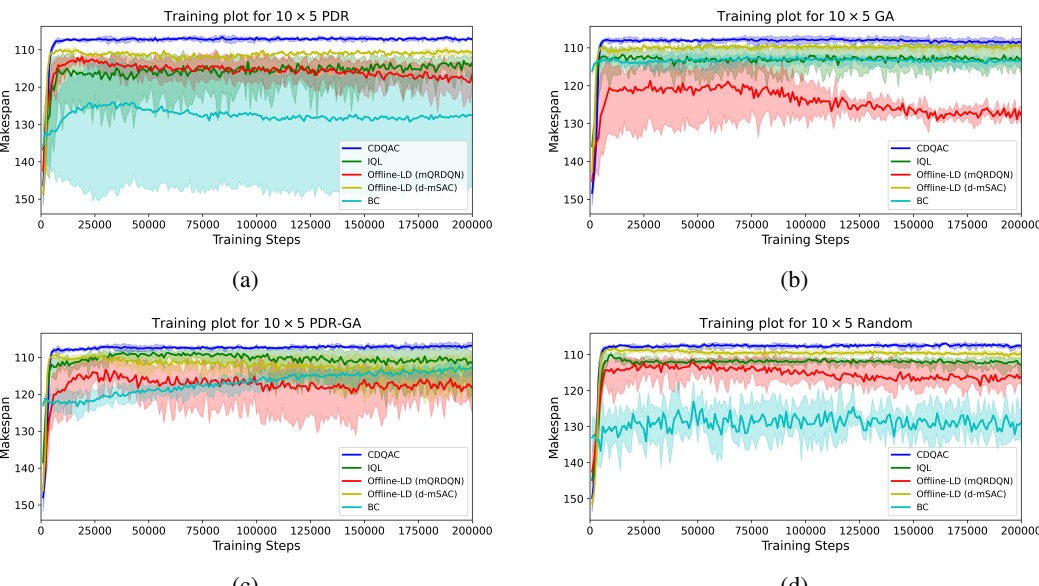

(a)  (b)

(c)  (d)

Figure 4: The training plots when trained of FJSP instances of size $10 \times 5$ for BC, CDQAC, IQL and Offline-LD, both mQRDQN and d-mSAC. Fig. 4a shows the training plots when trained on the PDR dataset, Fig. 4b with the GA dataset, Fig. 4c with the PDR-GA dataset, and Fig. 4d the Random dataset. The line is average makespan over four different seeds and the shaded area is minimal and maximal makespan of these seeds. We evaluate each method at every 1,000 steps of offline training.

$20 \times 10$ (Fig. 6c), and IQL with the PDR dataset for all training sizes. (Figs 4a, 5a and 6a). Lastly, we can notice for all training plots that CDQAC converges significantly faster than the other offline RL methods.

# H  ADDITIONAL RESULTS

## H.1  ABLATION STUDY

We conducted ablation studies to evaluate the contribution of two critical components of CDQAC: the use of a quantile critic with a dueling network architecture, and the impact of the delayed policy update frequency $\eta$. All experiments were performed on $10 \times 5$ instances using the Random dataset. We report results separately for generated instances (similar distribution as training data) and benchmark instances (Hurink and Brandimarte) to assess generalization.

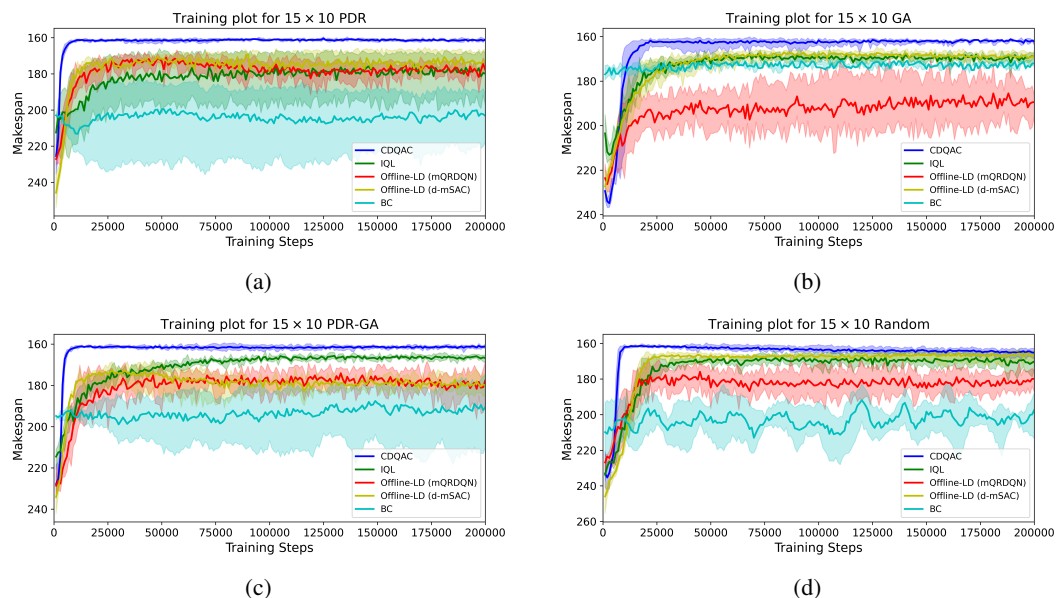

(a)          (b)

(c)          (d)

Figure 5: The training plots when trained of FJSP instances of size $15 \times 10$ for BC, CDQAC, IQL and Offline-LD, both mQRDQN and d-mSAC. Fig. 5a shows the training plots when trained on the PDR dataset, Fig. 5b with the GA dataset, Fig. 5c with the PDR-GA dataset, and Fig. 5d the Random dataset. The line is average makespan over four different seeds and the shaded area is minimal and maximal makespan of these seeds. We evaluate each method at every 1,000 steps of offline training.

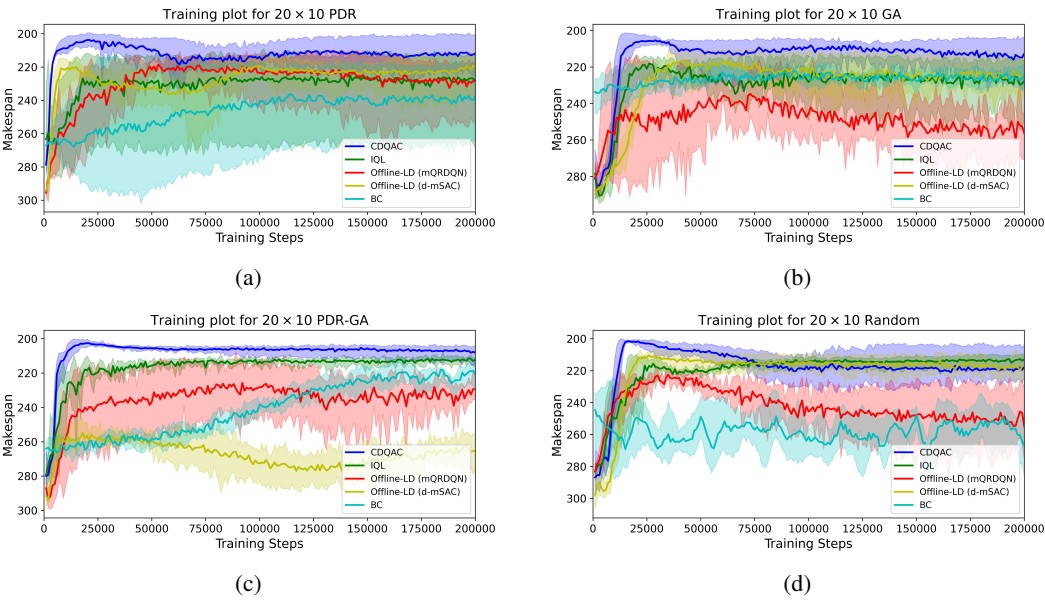

(a)          (b)

(c)          (d)

Figure 6: The training plots when trained of FJSP instances of size $20 \times 10$ for BC, CDQAC, IQL and Offline-LD, both mQRDQN and d-mSAC. Fig. 6a shows the training plots when trained on the PDR dataset, Fig. 6b with the GA dataset, Fig. 6c with the PDR-GA dataset, and Fig. 6d the Random dataset. The line is average makespan over four different seeds and the shaded area is minimal and maximal makespan of these seeds. We evaluate each method at every 1,000 steps of offline training.

**Critic Architecture.** In our ablation study for the critic, we tested both the effect of the quantile critic (yes or no quantile) compared to a critic that uses a standard DQN approach and our dueling network approach (yes or no dueling). This results in four different configurations: No Quantile

Table 9: Ablation study of the components of CDQAC, namely: the critic network architecture, the effect of the policy frequency update $\eta$, CQL regression $\alpha_{\text{CQL}}$, and the number of quantiles. CDQAC is trained on the Random dataset for instance size $10 \times 5$. The mean and standard deviation of the gap (%) are reported from four different seeds, separated for generated instances $10 \times 5$, and FJSP benchmarks (Brandimarte and Hurink). **Bold** indicates best result (lowest gap) for either the Greedy and Sampling (100 solutions) evaluation. (baseline) indicates the setup used in all other experiments. Avg $\Delta(\%)$ lists the average percentage difference in the gap of each variant relative to the baseline configuration.

| | Generated $10 \times 5$ (Gap %) | | Benchmarks (Gap %) | | Avg $\Delta(\%)$ |
|---|---|---|---|---|---|
| | Greedy | Sampling | Greedy | Sampling | |
| Critic Network Architecture | | | | | |
| No Quantile - No Dueling | $11.87 \pm 0.35$ | $5.98 \pm 0.22$ | $10.8 \pm 0.51$ | $6.31 \pm 0.17$ | 3.90 |
| No Quantile - Yes Dueling | $11.72 \pm 0.53$ | $6.05 \pm 0.11$ | $10.5 \pm 0.21$ | $6.24 \pm 0.14$ | 2.86 |
| Yes Quantile - No Dueling | $11.59 \pm 0.53$ | $5.99 \pm 0.27$ | $10.97 \pm 0.43$ | $6.45 \pm 0.30$ | 4.30 |
| Yes Quantile - Yes Dueling (baseline) | $\mathbf{11.19 \pm 0.35}$ | $\mathbf{5.87 \pm 0.14}$ | $\mathbf{10.45 \pm 0.39}$ | $\mathbf{6.05 \pm 0.10}$ | 0.00 |
| Policy Update Frequency $\eta$ | | | | | |
| $\eta = 1$ | $12.27 \pm 0.49$ | $6.30 \pm 0.14$ | $12.46 \pm 1.12$ | $6.69 \pm 0.27$ | 11.70 |
| $\eta = 2$ | $12.17 \pm 0.61$ | $6.30 \pm 0.34$ | $11.10 \pm 0.52$ | $6.39 \pm 0.23$ | 6.98 |
| $\eta = 3$ | $11.67 \pm 0.39$ | $6.05 \pm 0.31$ | $10.69 \pm 0.24$ | $6.39 \pm 0.13$ | 3.82 |
| $\eta = 4$ (baseline) | $\mathbf{11.19 \pm 0.40}$ | $\mathbf{5.87 \pm 0.14}$ | $\mathbf{10.45 \pm 0.39}$ | $\mathbf{6.05 \pm 0.10}$ | 0.00 |
| CQL regression $\alpha_{\text{CQL}}$ | | | | | |
| $\alpha_{\text{CQL}} = 0$ | $11.49 \pm 0.26$ | $5.98 \pm 0.17$ | $10.59 \pm 0.27$ | $6.16 \pm 0.14$ | 1.93 |
| $\alpha_{\text{CQL}} = 0.1$ | $11.36 \pm 0.51$ | $6.18 \pm 0.42$ | $10.81 \pm 0.21$ | $6.21 \pm 0.30$ | 3.22 |
| $\alpha_{\text{CQL}} = 0.05$ (baseline) | $\mathbf{11.19 \pm 0.40}$ | $\mathbf{5.87 \pm 0.14}$ | $\mathbf{10.45 \pm 0.39}$ | $\mathbf{6.05 \pm 0.10}$ | 0.00 |
| Number of quantiles $N$ | | | | | |
| $N = 4$ | $11.37 \pm 0.34$ | $5.95 \pm 0.19$ | $10.63 \pm 0.30$ | $6.13 \pm 0.13$ | 1.50 |
| $N = 8$ | $11.66 \pm 0.46$ | $5.87 \pm 0.32$ | $10.88 \pm 0.52$ | $6.13 \pm 0.34$ | 2.41 |
| $N = 16$ | $11.33 \pm 0.14$ | $\mathbf{5.77 \pm 0.18}$ | $10.60 \pm 0.31$ | $6.06 \pm 0.16$ | 0.29 |
| $N = 32$ | $11.38 \pm 0.50$ | $5.87 \pm 0.29$ | $10.67 \pm 0.53$ | $6.16 \pm 0.12$ | 1.41 |
| $N = 64$ (baseline) | $\mathbf{11.19 \pm 0.40}$ | $5.87 \pm 0.14$ | $\mathbf{10.45 \pm 0.39}$ | $\mathbf{6.05 \pm 0.10}$ | 0.00 |

- No Dueling, No Quantile - Yes Dueling, Yes Quantile - No Dueling, and Yes Quantile - Yes Dueling, which we used in our main experiments. Table 9 shows that both the quantile approach and our dueling architecture positively impact performance. On generated instances, introducing the dueling architecture to the quantile critic reduced the Greedy gap from $11.59\% \pm 0.53\%$ to $11.19\% \pm 0.35\%$, and for benchmark instances from $10.97\% \pm 0.43\%$ to $10.45\% \pm 0.39\%$. Similar trends were observed with DQN-based critic. These findings confirm the benefit of our novel dueling approach. Furthermore, comparing the dueling non-quantile approach ($11.72\% \pm 0.35\%$) with the dueling quantile critic ($11.19\% \pm 0.35\%$) on generated instances, we observe that the quantile critic results in lower gaps, highlighting the advantage of approximating the full return, with the quantile critic, over estimating only the expected return, with a DQN critic.

**Policy Update Frequency $\eta$.** We also varied the policy update frequency $\eta \in \{1, 2, 3, 4\}$ to study its effect. CDQAC uses $\eta = 4$ by default, which delays policy updates and allows more stable updates for the critic, which in turn, results in more stable updates for the policy. Table 9 shows that larger values for $\eta$ consistently lead to better performance. For example, for $\eta = 1$ the Greedy gap on benchmarks is $12.45\% \pm 1.12\%$, which decreases to $10.45\% \pm 0.39\%$ when $\eta = 4$. A similar pattern is observed for both sampling evaluation and generated instances. In addition to performance gains, higher values of $\eta$ also reduce training time, as the policy is updated less frequently. These results indicate that less frequent policy updates contribute to more stable learning.

**CQL Regression $\alpha_{\text{CQL}}$** To test the importance of CQL regression in CDQAC, we evaluated both CDQAC without CQL regression $\alpha_{\text{CQL}} = 0$ and with a stronger regression $\alpha_{\text{CQL}} = 0.1$. Table 9 indicates that both removing the regression or increasing the regression strength have a negative effect on the performance of CDQAC. Therefore, $\alpha_{\text{CQL}} = 0.05$ achieves the best performance; however, as noted by Kumar et al. (2020), the optimal value of $\alpha_{\text{CQL}}$ might differ between problem settings and types of datasets.

**The Number of Quantiles $N$** Lastly, we examined the sensitivy to the number of quantiles $N$ used by the critic of CDQAC. The results of these experiments (Table 9) indicate that CDQAC is not sensitive to the number of quantiles used. Table 9 shows that after increasing the number of quantiles to $N = 16$, the positive effect on CDQAC performance decreases.

The most essential component of CDQAC is the **policy update frequency**. Table 9 shows that without using a delayed update the performance decreases by 11.7% on average. The critic network has the second most important effect on performance, with CQL third and the number of quantiles last.

## H.2 RESULTS OFFLINE RL

Table 10: Results of FJSP offline RL comparison $10 \times 5$, for all training datasets (PDR, GA, PDR-GA, and Random). The columns show the evaluation benchmarks sets and the rows the methods. The mean and standard deviation of the gap (%) are reported from four different seeds. **Bold** indicates best result (lowest gap) for either the Greedy and Sampling (100 solutions) evaluation, for a given training dataset.

| | Generated $10 \times 5$ | | Brandimarte (mk) | | Hurink edata | | Hurink rdata | | Hurink vdata | |
|---|---|---|---|---|---|---|---|---|---|---|
| | Greedy | Sampling | Greedy | Sampling | Greedy | Sampling | Greedy | Sampling | Greedy | Sampling |
| **PDR** | | | | | | | | | | |
| BC | 31.79±1.96 | 10.45±0.84 | 72.5±5.36 | 33.7±1.42 | 31.03±2.02 | 13.93±0.89 | 30.04±3.18 | 12.58±1.12 | 14.97±2.2 | 4.16±0.84 |
| Offline-LD (mQRDQN) | 15.4±1.2 | 14.39±0.12 | 22.81±3.76 | 25.07±0.27 | 25.54±2.4 | 12.38±0.06 | 18.74±2.55 | 10.24±0.09 | 11.77±1.11 | 3.37±0.05 |
| Offline-LD (d-mSAC) | 15.26±0.85 | 8.16±0.11 | 43.74±5.43 | 23.18±3.39 | 22.17±2.1 | 10.18±0.8 | 21.93±3.5 | 9.34±2.36 | 7.55±0.76 | 1.3±0.2 |
| IQL | 15.58±0.47 | 8.13±0.17 | 41.75±3.87 | 21.89±1.15 | 22.87±1.84 | 11.25±0.98 | 21.36±3.56 | 8.12±1.0 | 7.32±0.89 | 1.43±0.4 |
| **CDQAC** | **11.49±0.38** | **5.64±0.08** | **12.43±1.45** | **8.3±0.14** | **15.11±1.06** | **9.68±0.57** | **10.81±0.22** | **5.54±0.12** | **3.69±0.25** | **0.78±0.02** |
| **GA** | | | | | | | | | | |
| BC | 14.63±0.7 | 8.91±0.3 | 16.03±1.81 | 15.03±0.46 | 15.27±0.58 | 8.79±0.31 | 11.36±0.59 | 6.69±0.17 | 4.48±0.23 | 1.43±0.07 |
| Offline-LD (mQRDQN) | 17.28±3.88 | 14.52±0.08 | 33.45±8.26 | 26.62±0.62 | 29.64±3.0 | 12.55±0.07 | 22.84±1.78 | 10.47±0.2 | 14.13±1.99 | 3.51±0.06 |
| Offline-LD (d-mSAC) | **11.38±0.64** | **5.29±0.1** | 23.47±3.33 | 12.05±1.37 | 21.55±3.24 | **9.23±1.23** | 16.32±2.16 | 5.99±0.47 | 11.37±1.92 | 2.89±1.02 |
| IQL | 13.02±0.86 | 7.32±0.18 | 26.71±1.01 | 14.57±0.71 | 25.67±2.1 | 10.72±0.58 | 17.37±1.81 | 6.75±0.31 | 10.69±1.76 | 2.14±0.62 |
| **CDQAC** | 11.62±0.35 | 6.09±0.22 | **15.51±1.0** | **9.58±0.76** | 14.87±0.63 | 9.45±0.54 | **10.44±0.4** | **5.39±0.2** | **3.24±0.3** | **0.65±0.01** |
| **PDR-GA** | | | | | | | | | | |
| BC | 16.79±1.13 | 8.86±0.08 | 57.26±4.68 | 27.0±1.77 | 24.37±1.15 | 11.17±0.3 | 28.38±1.03 | 10.99±1.29 | 15.72±1.5 | 3.18±1.17 |
| Offline-LD (mQRDQN) | 14.7±0.99 | 14.33±0.04 | 21.77±1.22 | 25.27±0.45 | 25.53±2.79 | 12.25±0.11 | 19.34±2.61 | 10.33±0.06 | 11.94±2.17 | 3.45±0.05 |
| Offline-LD (d-mSAC) | 12.1±0.65 | 5.9±0.48 | 19.49±2.67 | 11.17±0.68 | 19.04±1.61 | **8.82±0.61** | 13.27±0.84 | 5.58±0.31 | 7.59±1.86 | 1.32±0.32 |
| IQL | 12.4±0.24 | 7.22±0.09 | 33.13±4.61 | 19.43±2.27 | 26.44±3.42 | 11.69±1.26 | 21.42±3.37 | 7.83±0.67 | 11.24±1.83 | 2.06±0.3 |
| **CDQAC** | **11.16±0.43** | **5.88±0.37** | **14.24±1.23** | **8.79±0.74** | **15.3±0.57** | 9.84±0.38 | **10.96±0.56** | **5.51±0.16** | **3.59±0.31** | **0.72±0.03** |
| **Random** | | | | | | | | | | |
| BC | 25.59±2.86 | 14.91±0.05 | 31.74±2.78 | 26.95±0.2 | 22.12±1.46 | 12.26±0.06 | 17.16±2.35 | 10.48±0.17 | 7.98±2.22 | 3.46±0.08 |
| Offline-LD (mQRDQN) | 14.41±0.87 | 14.17±0.14 | 21.42±1.44 | 25.0±1.03 | 19.05±1.5 | 11.93±0.11 | 14.85±1.64 | 9.98±0.15 | 7.91±1.68 | 3.22±0.15 |
| Offline-LD (d-mSAC) | 13.29±0.45 | 6.26±0.27 | 16.62±0.6 | 9.49±0.37 | 16.12±1.43 | **8.24±0.32** | 12.13±0.99 | 5.67±0.23 | 4.14±0.74 | 0.87±0.08 |
| IQL | 15.64±1.2 | 8.98±0.15 | 33.11±5.9 | 18.73±1.42 | 26.91±4.2 | 11.5±1.29 | 17.65±3.0 | 7.5±0.87 | 12.85±5.68 | 2.75±1.45 |
| **CDQAC** | **11.19±0.35** | **5.87±0.14** | **13.78±0.78** | **8.67±0.21** | **14.53±0.41** | 9.54±0.39 | **10.4±0.36** | **5.3±0.22** | **3.1±0.22** | **0.68±0.03** |

In this section, we provide a comprehensive overview of the results discussed in Sect.5.1 and Table1, where we compare our proposed method, CDQAC, to Offline-LD (Remmerden et al., 2025). Table 1 presents the average performance across all evaluation instance sets—both generated and benchmark—for each training size ($10 \times 5$, $15 \times 10$, and $20 \times 10$). The detailed results for each evaluation set are reported in Table 10 (training size $10 \times 5$), Table 11 ($15 \times 10$), and Table 12 ($20 \times 10$).

As shown in Tables 10, 11, and 12, CDQAC consistently outperforms both versions of Offline-LD in nearly all evaluations. There are only a few exceptions: in Table 10, Offline-LD (d-mSAC) marginally exceeds CDQAC in the generated instances and Hurink edata using the sampling evaluation when trained on the GA dataset, as well as on Hurink edata with the sampling evaluation when both methods are trained on the Random dataset. Nevertheless, CDQAC shows better performance on the remaining evaluation sets for both the GA and Random training sets. Furthermore, with larger training sizes, $15 \times 10$ (Table 11) and $20 \times 10$ (Table 12), CDQAC consistently outperforms Offline-LD, and the performance margins widen as the instance size increases. These findings indicate that CDQAC scales more efficiently to larger instance sizes, and is generally an improvement over the offline RL baseline, Offline-LD.

Analyzing CDQAC's performance across different instance sizes and training datasets, we observe that for both $10 \times 5$ (Table 10) and $15 \times 10$ (Table 11), CDQAC achieves the worst performance when trained on the GA dataset across all evaluation sets. In contrast, for $20 \times 10$, CDQAC trained on the GA dataset achieves the best performance on generated instances (Greedy: 5.01%±0.28%), while training on PDR yields the worst results (Greedy: 9.38%±6.1%), accompanied by a high standard deviation. This higher standard deviation with PDR suggests instability during training, as one of the four runs did not train effectively. Additionally, we find that, when trained on GA,

Table 11: Results of FJSP offline RL comparison $15 \times 10$, for all training datasets (PDR, GA, PDR-GA, and Random). The columns show the evaluation benchmarks sets and the rows the methods. The mean and standard deviation of the gap (%) are reported from four different seeds. **Bold** indicates best result (lowest gap) for either the Greedy and Sampling (100 solutions) evaluation, for a given training dataset.

| | Generated $15 \times 10$ | | Brandimarte (mk) | | Hurink edata | | Hurink rdata | | Hurink vdata | |
|---|---|---|---|---|---|---|---|---|---|---|
| | Greedy | Sampling | Greedy | Sampling | Greedy | Sampling | Greedy | Sampling | Greedy | Sampling |
| PDR | | | | | | | | | | |
| BC | 36.31±3.88 | 13.58±0.88 | 58.23±13.17 | 30.47±3.21 | 28.66±6.05 | 12.13±1.48 | 23.93±2.1 | 8.57±1.13 | 10.38±1.4 | 2.04±0.49 |
| Offline-LD (mQRDQN) | 17.36±1.17 | 20.28±0.09 | 22.89±1.89 | 24.88±0.22 | 30.32±1.54 | 12.51±0.17 | 19.93±1.61 | 10.2±0.15 | 10.01±2.47 | 3.33±0.07 |
| Offline-LD (d-mSAC) | 16.37±0.5 | 10.54±0.16 | 39.55±6.46 | 23.6±2.54 | 23.63±6.52 | 11.85±3.21 | 14.93±2.03 | 6.43±0.16 | 5.82±0.95 | 1.26±0.22 |
| IQL | 16.35±0.53 | 10.51±0.22 | 30.95±2.93 | 19.75±0.77 | 20.5±0.29 | **9.98±0.19** | 14.08±1.57 | 6.38±0.08 | 5.54±0.34 | 1.04±0.06 |
| **CDQAC** | **12.21±0.37** | **6.48±0.15** | **14.6±0.78** | **9.6±0.1** | **17.67±1.49** | 10.77±0.35 | **11.67±0.6** | **5.76±0.08** | **3.94±0.43** | **0.87±0.16** |
| GA | | | | | | | | | | |
| BC | 17.21±0.31 | 13.41±0.09 | 28.88±1.67 | 18.13±0.21 | 17.73±0.78 | 9.22±0.12 | 14.43±0.44 | 6.97±0.05 | 11.03±0.61 | 1.8±0.07 |
| Offline-LD (mQRDQN) | 24.67±2.98 | 20.47±0.07 | 45.24±4.87 | 27.03±0.38 | 34.83±1.61 | 12.9±0.11 | 28.1±1.6 | 10.72±0.07 | 19.63±1.82 | 3.78±0.06 |
| Offline-LD (d-mSAC) | 16.11±0.71 | 8.74±0.1 | 29.23±1.9 | 14.89±0.51 | 31.93±2.12 | 13.69±0.86 | 22.88±1.09 | 8.39±0.13 | 16.12±2.14 | 4.71±0.56 |
| IQL | 15.54±0.84 | 11.08±0.2 | 26.69±3.37 | 16.28±0.79 | 26.84±3.04 | **11.78±0.84** | 20.41±2.1 | 7.72±0.47 | 14.15±2.5 | 3.26±1.04 |
| **CDQAC** | **12.3±0.45** | **6.19±0.24** | **19.6±4.61** | **10.22±1.76** | **23.53±6.23** | 11.8±2.82 | **14.37±3.46** | **6.13±0.83** | **7.46±3.69** | **1.63±0.96** |
| PDR-GA | | | | | | | | | | |
| BC | 23.94±4.08 | 13.35±0.76 | 57.21±3.94 | 26.53±0.61 | 27.61±1.06 | 11.73±0.68 | 22.35±2.5 | 8.37±0.79 | 12.78±1.97 | 2.67±0.31 |
| Offline-LD (mQRDQN) | 18.15±1.12 | 20.34±0.04 | 23.98±3.91 | 25.53±0.44 | 27.62±2.08 | 12.52±0.23 | 21.92±1.47 | 10.42±0.14 | 12.19±2.4 | 3.5±0.1 |
| Offline-LD (d-mSAC) | 17.42±0.65 | 9.36±0.36 | 35.9±4.16 | 17.54±1.75 | 34.09±3.15 | 14.81±1.36 | 21.91±1.3 | 8.75±0.29 | 14.99±1.35 | 4.62±0.22 |
| IQL | 15.33±0.52 | 10.5±0.13 | 28.15±1.59 | 19.1±0.66 | 25.06±2.39 | 11.43±0.43 | 16.4±2.51 | 6.69±0.24 | 6.56±2.62 | 1.22±0.26 |
| **CDQAC** | **12.28±0.26** | **6.15±0.47** | **14.75±1.53** | **8.72±0.59** | **18.02±4.44** | **9.55±1.42** | **11.44±0.88** | **5.44±0.28** | **3.51±0.91** | **0.78±0.15** |
| Random | | | | | | | | | | |
| BC | 30.41±3.73 | 20.87±0.09 | 36.61±4.36 | 26.66±0.33 | 25.66±2.26 | 12.33±0.09 | 22.56±3.89 | 10.5±0.12 | 10.92±3.12 | 3.58±0.03 |
| Offline-LD (mQRDQN) | 16.95±0.54 | 20.21±0.07 | 29.14±4.62 | 25.6±0.39 | 29.07±3.02 | 12.58±0.24 | 20.17±2.17 | 10.24±0.12 | 12.83±1.86 | 3.41±0.07 |
| Offline-LD (d-mSAC) | 15.02±0.43 | 8.17±0.31 | 20.44±1.58 | 11.27±0.49 | 30.92±3.15 | 14.52±1.5 | 18.06±1.22 | 7.46±0.33 | 9.97±1.41 | 2.32±0.45 |
| IQL | 15.58±1.64 | 13.75±0.28 | 24.5±2.93 | 18.12±0.42 | 24.63±4.43 | 11.54±0.78 | 19.69±2.85 | 8.81±0.53 | 12.43±3.42 | 3.75±0.86 |
| **CDQAC** | **12.04±0.59** | **6.7±0.62** | **13.58±0.66** | **8.73±0.73** | **14.56±0.55** | **8.51±0.52** | **10.77±0.36** | **5.22±0.12** | **3.16±0.1** | **0.67±0.02** |

Table 12: Results of FJSP offline RL comparison $20 \times 10$, for all training datasets (PDR, GA, PDR-GA, and Random). The columns show the evaluation benchmarks sets and the rows the methods. The mean and standard deviation of the gap (%) are reported from four different seeds. **Bold** indicates best result (lowest gap) for either the Greedy and Sampling (100 solutions) evaluation, for a given training dataset.

| | Generated $20 \times 10$ | | Brandimarte (mk) | | Hurink edata | | Hurink rdata | | Hurink vdata | |
|---|---|---|---|---|---|---|---|---|---|---|
| | Greedy | Sampling | Greedy | Sampling | Greedy | Sampling | Greedy | Sampling | Greedy | Sampling |
| PDR | | | | | | | | | | |
| BC | 33.37±2.71 | 9.29±0.78 | 65.13±6.3 | 34.94±2.12 | 27.47±3.56 | 12.91±2.18 | 24.42±4.57 | 9.11±0.96 | 9.03±2.55 | 1.27±0.04 |
| Offline-LD (mQRDQN) | 27.6±5.91 | 14.82±0.12 | 33.83±2.4 | 26.54±1.25 | 31.03±2.1 | 12.61±0.23 | 28.02±3.74 | 10.55±0.11 | 18.73±2.71 | 3.56±0.09 |
| Offline-LD (d-mSAC) | 15.43±3.82 | 8.38±1.08 | 55.97±4.05 | 33.3±1.67 | 33.17±4.2 | 15.66±2.26 | 23.86±1.87 | 8.91±0.87 | 9.9±2.93 | 2.11±0.9 |
| IQL | 10.43±1.11 | 6.77±0.33 | 45.31±3.96 | 24.95±1.83 | 25.31±4.42 | 11.8±1.26 | 16.59±1.26 | 7.19±0.26 | **5.06±0.41** | **1.06±0.06** |
| **CDQAC** | **9.38±6.1** | **4.38±3.47** | **16.65±0.5** | **9.7±0.7** | **21.5±5.18** | **11.23±1.97** | **15.53±3.05** | **6.98±1.29** | 8.47±4.0 | 2.94±2.31 |
| GA | | | | | | | | | | |
| BC | 11.73±0.59 | 8.4±0.09 | 24.69±1.69 | 18.36±0.44 | 17.76±0.06 | 9.69±0.16 | 13.51±0.4 | 7.12±0.11 | 8.08±1.44 | 1.85±0.04 |
| Offline-LD (mQRDQN) | 41.47±6.36 | 15.55±0.46 | 59.54±3.52 | 27.8±0.81 | 35.95±2.51 | 13.18±0.42 | 32.75±4.96 | 10.9±0.3 | 23.3±4.49 | 3.93±0.2 |
| Offline-LD (d-mSAC) | 20.78±5.21 | 6.75±1.63 | 28.37±0.97 | 14.84±0.77 | 29.33±1.33 | 12.93±0.52 | 21.76±2.85 | 7.76±0.64 | 14.73±2.45 | 4.35±0.52 |
| IQL | 21.12±4.65 | 7.59±0.57 | 28.71±2.85 | 16.24±0.43 | 26.89±2.31 | 11.63±0.56 | 22.18±2.55 | 7.64±0.4 | 13.96±1.01 | 3.14±0.82 |
| **CDQAC** | **5.22±0.63** | **2.19±0.62** | **16.76±2.09** | **9.3±0.36** | **22.62±6.05** | **11.05±3.2** | **13.48±0.97** | **5.92±0.37** | **4.91±1.07** | **0.97±0.2** |
| PDR-GA | | | | | | | | | | |
| BC | 26.02±2.15 | 8.21±0.17 | 53.55±6.26 | 28.02±2.45 | 23.61±1.62 | 10.91±0.87 | 16.26±3.61 | 7.15±0.4 | 5.89±2.1 | 1.08±0.09 |
| Offline-LD (mQRDQN) | 27.62±9.83 | 15.0±0.21 | 29.47±8.62 | 25.59±0.29 | 30.82±5.5 | 12.62±0.29 | 24.68±4.86 | 10.51±0.12 | 17.36±5.16 | 3.6±0.14 |
| Offline-LD (d-mSAC) | 43.5±3.7 | 21.72±5.92 | 55.46±4.38 | 24.48±1.64 | 38.71±1.75 | 19.46±1.27 | 32.65±3.36 | 13.12±1.35 | 22.98±2.97 | 8.78±2.03 |
| IQL | 11.42±2.36 | 6.89±0.25 | 33.97±4.27 | 19.32±1.4 | 24.65±2.62 | 10.79±0.56 | 16.03±1.22 | 6.91±0.25 | 6.44±1.63 | 1.19±0.17 |
| **CDQAC** | **5.01±0.28** | **2.31±0.36** | **15.34±1.11** | **8.9±0.59** | **17.79±5.04** | **9.17±1.49** | **12.3±1.61** | **5.57±0.43** | **4.07±0.91** | **0.83±0.24** |
| Random | | | | | | | | | | |
| BC | 19.65±1.43 | 15.33±0.08 | 35.81±9.72 | 27.02±0.37 | 23.84±2.46 | 12.29±0.1 | 20.23±0.37 | 10.47±0.1 | 11.38±0.95 | 3.51±0.12 |
| Offline-LD (mQRDQN) | 21.73±9.18 | 14.9±0.19 | 40.78±4.11 | 26.36±0.46 | 33.87±1.93 | 12.81±0.25 | 24.68±1.19 | 10.52±0.1 | 15.62±3.59 | 3.59±0.08 |
| Offline-LD (d-mSAC) | 11.59±3.69 | 4.79±1.46 | 22.09±3.25 | 11.7±1.28 | 28.51±2.45 | 13.37±1.85 | 21.7±3.27 | 9.18±1.85 | 13.07±3.76 | 3.7±2.14 |
| IQL | 14.0±4.05 | 10.08±0.93 | 32.8±4.58 | 20.08±0.72 | 31.66±2.55 | 13.17±0.61 | 24.87±3.32 | 9.63±0.39 | 13.81±3.39 | 3.53±0.5 |
| **CDQAC** | **5.2±0.66** | **2.87±0.73** | **16.52±0.3** | **9.73±0.43** | **16.53±1.59** | **9.02±0.28** | **11.63±0.52** | **5.66±0.17** | **3.25±0.2** | **0.76±0.05** |

CDQAC struggles to generalize to unseen evaluation instances compared to when trained on more diverse datasets, such as Random and PDR-GA. This further supports the conclusion that training on a diverse set of examples is critical for strong generalization performance in offline RL for FJSP.

## H.3 ADDITIONAL RESULTS JSP

Table 13: Results on JSP benchmarks for CDQAC $10 \times 5$, for all training datasets (PDR, GA, PDR-GA and Random). The mean and standard deviation of the gap (%) are reported from four different seeds. **Bold** indicates best result (lowest gap) for either the Greedy and Sampling (100 solutions) evaluation.

| | Instance Size | Greedy | | | | Sampling | | | |
|---|---|---|---|---|---|---|---|---|---|
| | | PDR | GA | PDR-GA | Random | PDR | GA | PDR-GA | Random |
| Taillard | 15 × 15 | 16.26 ± 0.67 | 16.12 ± 0.69 | 16.33 ± 0.95 | **15.9 ± 0.7** | 11.5 ± 0.51 | 11.27 ± 0.86 | 11.23 ± 0.48 | **10.8 ± 0.55** |
| | 20 × 15 | 20.55 ± 0.95 | 19.7 ± 1.05 | **19.6 ± 1.91** | 19.98 ± 1.91 | 14.8 ± 0.37 | 14.23 ± 0.75 | 14.64 ± 0.58 | **14.12 ± 0.77** |
| | 20 × 20 | 18.65 ± 0.73 | 18.89 ± 1.27 | 17.45 ± 0.7 | **17.19 ± 1.38** | 13.29 ± 0.68 | 14.1 ± 0.72 | 13.88 ± 0.35 | 13.39 ± 0.84 |
| | 30 × 15 | 20.4 ± 0.65 | 21.32 ± 2.78 | 20.44 ± 1.13 | **19.56 ± 0.49** | 15.83 ± 0.34 | 16.04 ± 0.91 | 16.0 ± 0.31 | **15.3 ± 1.13** |
| | 30 × 20 | 22.05 ± 1.64 | 22.58 ± 2.72 | **21.6 ± 2.04** | 22.28 ± 1.01 | **17.89 ± 0.92** | 18.6 ± 1.3 | 18.6 ± 0.4 | 18.27 ± 0.82 |
| | 50 × 15 | 14.26 ± 1.1 | 14.48 ± 1.63 | 13.53 ± 1.41 | **13.06 ± 1.47** | 10.86 ± 0.66 | **10.21 ± 0.75** | 10.47 ± 1.18 | 10.46 ± 1.22 |
| | 50 × 20 | 14.46 ± 0.95 | 15.21 ± 3.36 | **13.83 ± 1.18** | 13.9 ± 1.3 | 11.6 ± 0.35 | 12.07 ± 1.21 | 11.62 ± 0.47 | **11.37 ± 0.52** |
| | 100 × 20 | 6.43 ± 0.12 | 8.1 ± 4.73 | 6.18 ± 0.82 | **5.53 ± 1.12** | 4.66 ± 0.14 | 4.46 ± 1.33 | 4.56 ± 0.49 | **4.25 ± 0.59** |
| | Mean | 16.63 ± 0.85 | 17.05 ± 2.28 | 16.12 ± 1.27 | **15.93 ± 1.17** | 12.55 ± 0.5 | 12.62 ± 0.98 | 12.62 ± 0.53 | **12.24 ± 0.8** |
| Demirkol | 20 × 15 | 24.87 ± 1.51 | **24.03 ± 0.94** | 24.47 ± 2.11 | 24.49 ± 1.83 | 19.4 ± 0.63 | 19.29 ± 0.94 | 19.63 ± 0.81 | **18.82 ± 0.86** |
| | 20 × 20 | 23.3 ± 0.36 | **21.29 ± 1.19** | 22.01 ± 1.12 | 21.71 ± 1.47 | 17.66 ± 0.45 | 17.62 ± 1.15 | 18.03 ± 0.54 | **17.13 ± 0.71** |
| | 30 × 15 | 29.63 ± 0.69 | **28.22 ± 1.8** | 28.71 ± 2.63 | 28.76 ± 1.72 | 24.21 ± 0.61 | **23.22 ± 1.1** | 24.2 ± 1.21 | 23.67 ± 1.7 |
| | 30 × 20 | 28.72 ± 1.13 | **28.33 ± 1.0** | 28.53 ± 2.57 | 28.6 ± 2.39 | 23.72 ± 0.61 | 23.71 ± 0.5 | 24.15 ± 1.55 | **23.56 ± 1.29** |
| | 40 × 15 | 26.98 ± 1.0 | **25.1 ± 1.35** | 25.76 ± 2.78 | 25.51 ± 2.85 | 22.62 ± 0.98 | **20.31 ± 0.84** | 21.73 ± 1.63 | 21.15 ± 1.66 |
| | 40 × 20 | 29.42 ± 1.18 | **27.49 ± 1.45** | 28.5 ± 2.67 | 28.77 ± 1.74 | 24.88 ± 0.18 | **24.06 ± 1.03** | 25.1 ± 1.7 | 24.58 ± 1.49 |
| | 50 × 15 | 27.82 ± 0.94 | **25.03 ± 2.61** | 26.49 ± 3.84 | 25.06 ± 5.42 | 23.8 ± 0.85 | **20.83 ± 0.97** | 22.53 ± 2.5 | 22.5 ± 2.74 |
| | 50 × 20 | 30.43 ± 0.96 | **27.5 ± 1.63** | 28.71 ± 2.98 | 28.65 ± 2.58 | 26.35 ± 0.69 | **24.65 ± 1.28** | 26.1 ± 1.52 | 25.67 ± 1.06 |
| | Mean | 27.65 ± 0.97 | **25.87 ± 1.5** | 26.65 ± 2.59 | 26.44 ± 2.5 | 22.83 ± 0.62 | **21.71 ± 0.98** | 22.68 ± 1.43 | 22.13 ± 1.44 |

Table 14: Results on JSP benchmarks for CDQAC $15 \times 10$, for all training datasets (PDR, GA, PDR-GA and Random). The mean and standard deviation of the gap (%) are reported from four different seeds. **Bold** indicates best result (lowest gap) for either the Greedy and Sampling (100 solutions) evaluation.

| | Instance Size | Greedy | | | | Sampling | | | |
|---|---|---|---|---|---|---|---|---|---|
| | | PDR | GA | PDR-GA | Random | PDR | GA | PDR-GA | Random |
| Taillard | 15 × 15 | 16.73 ± 0.6 | 17.23 ± 1.28 | 17.35 ± 1.89 | **16.7 ± 0.97** | 11.6 ± 0.48 | 11.26 ± 0.84 | 11.39 ± 0.86 | **11.24 ± 0.83** |
| | 20 × 15 | 21.63 ± 0.33 | 21.4 ± 1.92 | 21.57 ± 2.04 | **20.69 ± 1.09** | 15.22 ± 0.23 | 14.77 ± 1.13 | 14.87 ± 0.92 | **14.71 ± 0.28** |
| | 20 × 20 | 18.73 ± 0.71 | 18.51 ± 1.41 | 19.0 ± 1.11 | **18.14 ± 0.81** | 13.61 ± 0.59 | **13.49 ± 0.57** | 13.76 ± 0.52 | 13.53 ± 0.5 |
| | 30 × 15 | **20.6 ± 0.65** | 21.27 ± 1.27 | 21.33 ± 1.4 | 20.86 ± 0.88 | 16.01 ± 0.14 | 16.21 ± 0.88 | 16.07 ± 0.51 | **15.92 ± 0.3** |
| | 30 × 20 | 23.52 ± 1.14 | **23.17 ± 0.34** | 23.94 ± 0.69 | 23.55 ± 1.14 | 18.43 ± 0.6 | **18.15 ± 0.47** | 18.43 ± 0.62 | 18.29 ± 0.5 |
| | 50 × 15 | 14.9 ± 0.28 | 14.6 ± 1.09 | **14.05 ± 1.31** | 15.47 ± 2.47 | 11.45 ± 0.54 | 10.71 ± 1.06 | 10.8 ± 1.4 | **10.48 ± 0.43** |
| | 50 × 20 | **14.82 ± 0.77** | 16.46 ± 0.99 | 15.41 ± 1.03 | 16.47 ± 4.19 | 12.05 ± 0.54 | 11.93 ± 0.82 | 12.17 ± 1.13 | **11.57 ± 0.24** |
| | 100 × 20 | 6.44 ± 0.34 | 8.24 ± 2.43 | **6.01 ± 0.97** | 8.0 ± 3.19 | 4.88 ± 0.25 | 4.73 ± 0.34 | **4.52 ± 0.49** | 4.96 ± 0.75 |
| | Mean | **17.17 ± 0.6** | 17.61 ± 1.34 | 17.33 ± 1.31 | 17.48 ± 1.84 | 12.91 ± 0.42 | 12.66 ± 0.77 | 12.75 ± 0.81 | **12.59 ± 0.48** |
| Demirkol | 20 × 15 | 27.13 ± 0.74 | 26.03 ± 1.0 | **24.94 ± 1.91** | 26.05 ± 1.37 | 20.23 ± 0.8 | **19.4 ± 1.05** | 19.5 ± 1.3 | 19.59 ± 0.72 |
| | 20 × 20 | 24.01 ± 0.5 | 22.86 ± 1.19 | 22.73 ± 2.13 | **22.67 ± 1.4** | 17.59 ± 0.62 | 17.4 ± 0.62 | 17.6 ± 0.66 | **17.23 ± 0.68** |
| | 30 × 15 | 30.3 ± 1.13 | 29.66 ± 1.54 | 29.19 ± 1.49 | **29.15 ± 1.19** | 25.93 ± 1.37 | **24.04 ± 1.21** | 24.05 ± 1.9 | 24.25 ± 0.87 |
| | 30 × 20 | 30.43 ± 1.04 | 28.65 ± 1.32 | 28.5 ± 2.35 | **28.24 ± 1.57** | 24.92 ± 0.64 | **23.0 ± 0.83** | 23.72 ± 1.55 | 23.46 ± 1.07 |
| | 40 × 15 | 27.81 ± 0.97 | 25.68 ± 0.97 | **24.77 ± 2.6** | 25.65 ± 1.67 | 25.61 ± 1.71 | 23.51 ± 1.04 | **21.03 ± 1.25** | 21.38 ± 1.49 |
| | 40 × 20 | 30.54 ± 1.26 | **27.64 ± 2.05** | 28.47 ± 2.71 | 28.99 ± 0.81 | 25.86 ± 1.0 | **23.63 ± 0.98** | 24.48 ± 1.73 | 24.21 ± 1.6 |
| | 50 × 15 | 29.14 ± 0.94 | **23.84 ± 4.59** | 24.28 ± 5.27 | 26.16 ± 2.21 | 25.09 ± 1.04 | 20.78 ± 2.54 | 21.87 ± 3.35 | **20.65 ± 3.01** |
| | 50 × 20 | 31.56 ± 1.3 | 28.85 ± 1.36 | **28.43 ± 1.44** | 30.53 ± 1.94 | 27.19 ± 1.34 | **24.4 ± 0.79** | 24.97 ± 0.87 | 25.47 ± 1.48 |
| | Mean | 28.87 ± 0.99 | 26.65 ± 1.75 | **26.42 ± 2.49** | 27.18 ± 1.52 | 23.79 ± 0.98 | **21.71 ± 1.16** | 22.16 ± 1.69 | 22.03 ± 1.36 |

In Sect. 5.3, we compared CDQAC on the Taillard and Demirkol instances. The results in Table 6 included only CDQAC trained on the Random dataset for $10 \times 5$ instances. In this section, we show the results for the other training sets for both $10 \times 5$ (Table 13) and $15 \times 10$ (Table 14) instances.

Tables 13 and 14 show only minor performance differences between the training datasets. Table 14 contains the largest difference between the mean Greedy results of Demirkol between PDR (28.87%±0.99%) and PDR-GA (26.42%±2.49%). We also notice that PDR and Random perform better with the Taillard instances compared to GA, but GA performs better on the Demirkol instances. We hypothesize that this difference comes from the differing distributions of processing times: Demirkol instances have processing times ranging from 1 to 200 and those of Taillard only from 1 to 100, whereby CDQAC was trained on instances similar to Taillard instances. These re-

sults contrast with those of FJSP in App. H.2, where GA was unable to generalize well to benchmark instances that have a different distribution to the training instances. These results suggest that the choice of training data has a fundamentally different impact in JSP compared to FJSP.

## H.4 ADDITIONAL RESULTS DATASET SIZE

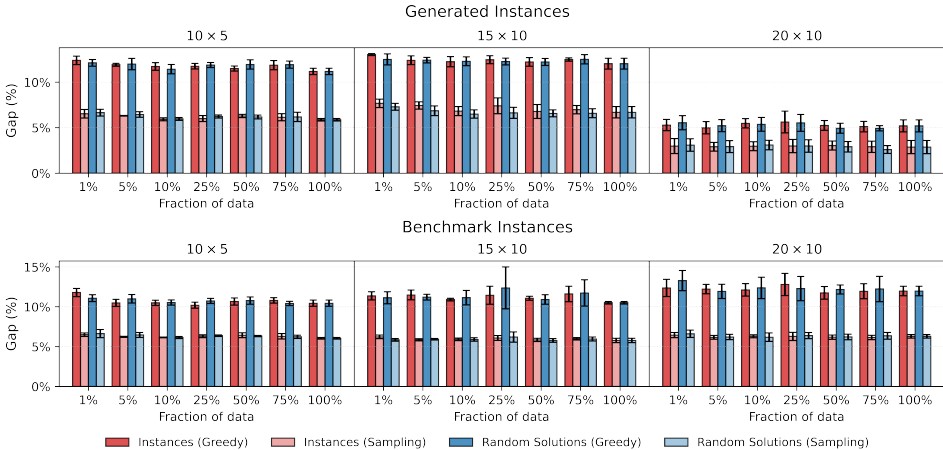

Figure 7: Effect of different dataset sizes. We evaluate the sample efficiency of CDQAC by reducing the Random training dataset in two ways. **Red**: the number of *instances* (1%: 5 instances, 5%: 25 instances, 10%: 50 instances, 25%: 125 instances, 50%: 250 instances, 75%: 375 instances, 100%: 500 instances, with each instance having 100 random solutions). **Blue**: the number of *random solutions* per instance (1%: 1 solution, 5%: 5 solutions, 10%: 10 solutions, 25%: 25 solutions, 50%: 50 solutions, 75%: 75 solutions, 100%: 100 solutions, for each instance, with 500 instances in total). Performance is reported as the mean gap across four seeds, with error bars indicating standard deviation.

In Sect. 5.4, we demonstrated that reducing the number of training in the Random training dataset had little impact on overall performance on the FJSP benchmark sets, Brandimarte and Hurink. In this section, we provide a more comprehensive analysis by including results on generated evaluation instances. Additionally, we introduce a second evaluation for the reduction of the dataset, in which we decrease the number of solutions generated per instance by the random policy. For both evaluations, we considered subsets containing 1%, 5%, 10%, 25%, 50%, 75%, and 100% of the original dataset size. Specifically, when reducing the number of instances, we used either 5, 25, 50, 125, 250, 375, or 500 instances, each with 100 random solutions. When reducing the number of random solutions per instance, we used 500 instances, each with either 1, 5, 10, 25, 50, 75, or 100 random solutions.

As shown in Fig. 7, decreasing the dataset, either by limiting the number of instances or by reducing the number of random solutions per instance, does not lead to a significant loss in performance. The results remain relatively stable, with the standard deviation mostly below 1.5%. The sole exception occurs for $15 \times 10$ on the benchmark instances at 25%, when reducing the number of random solutions, where the greedy evaluation shows a standard deviation of 2.63%. Notably, this increased standard deviation is only observed for benchmark instances and not for generated instances at 25% random solutions, as evidenced in Fig. 7. This suggests that larger datasets may improve generalization to previously unseen instances. Another benefit is training stability, with larger dataset producing a smaller standard deviation. In general, these findings reinforce our conclusion from Sect. 5.4: CDQAC maintains competitive performance even when trained on substantially reduced datasets, underscoring its sample efficiency.

## H.5 ADDITIONAL JSP BASELINES

In Table 15, we have included an additional comparison for JSP, where we compare CDQAC to other constructive learning-based approaches. The main distinction between the results in Table 6

Table 15: Results JSP benchmarks. Average gap (%) is reported. In this additional comparison, we compare CDQAC to constructive learning-based approaches that only function for JSP and to approaches that function for both JSP and FJSP. The approaches that only function for JSP are: L2D (Zhang et al., 2020), CL (Iklassov et al., 2023), Sched (Park et al., 2021), SL (Corsini et al., 2024), GD (Pirnay & Grimm, 2024), OD (Remmerden et al., 2025), and IL (Lee & Kim, 2025). Approaches that can do both JSP and FJSP are: DAN (Wang et al., 2023), Res (Ho et al., 2024), and CDQAC (ours). We note the best performing overall approach with $^*$, and the best approach that can handle both JSP and FJSP in **bold**.

| | Instance Size | Greedy | | | | | | | | | | Sampling | | | | | |
|---|---|---|---|---|---|---|---|---|---|---|---|---|---|---|---|---|---|
| | | L2D | CL | Sched | SL | GD | OD | IL | DAN | Res | CDQAC | CL[a] | SL[a] | GD[b] | DAN[c] | Res[c] | CDQAC[c] |
| Taillard | 15 × 15 | 28.1 | 14.3 | 15.3 | 13.8 | 9.6 | 25.8 | 8.8* | 19.0 | 17.6 | **15.0** | 9.0 | 7.2* | 10.1 | 13.2 | 13.3 | **10.4** |
| | 20 × 15 | 32.7 | 16.5 | 19.4 | 15.0 | 9.9* | 30.2 | 11.7 | 22.1 | 21.2 | **17.7** | 10.6 | 9.3* | 9.8 | 17.4 | 16.1 | **13.2** |
| | 20 × 20 | 31.8 | 17.3 | 17.2 | 15.2 | 11.1* | 28.9 | 13.2 | 18.0 | 18.0 | **17.6** | 10.9 | 10.0* | 10.4 | 13.3 | 15.8 | **12.9** |
| | 30 × 15 | 30.2 | 18.5 | 18.0 | 17.1 | 9.5* | 29.2 | 10.3 | 21.7 | 20.1 | **19.1** | 14.0 | 11.0 | 8.5* | 17.2 | 18.0 | **14.9** |
| | 30 × 20 | 35.2 | 21.5 | 18.7 | 18.5 | 13.8* | 33.1 | 14.7 | 23.2 | 22.3 | **21.2** | 16.1 | 13.4 | 12.3* | 19.0 | 19.7 | **17.9** |
| | 50 × 15 | 21.0 | 12.2 | 13.8 | 10.1 | 2.7* | 20.6 | 4.3 | 14.8 | 15.6 | **13.0** | 9.3 | 5.5 | 2.6* | 12.7 | 13.2 | **9.9** |
| | 50 × 20 | 26.1 | 13.2 | 13.5 | 11.6 | 6.7* | 24.3 | 9.0 | 16.0 | 14.4 | **12.8** | 9.9 | 8.4 | 7.7* | 13.1 | 14.1 | **11.0** |
| | 100 × 20 | 13.3 | 5.9 | 6.6 | 5.8 | 1.7* | 12.7 | 2.5 | 7.3 | 6.5 | **5.3** | 4.0 | 2.3 | 1.3* | 5.9 | 6.5 | **3.6** |
| | Mean | 27.3 | 14.9 | 15.4 | 13.4 | 8.1* | 25.6 | 9.3 | 18.2 | 17.0 | **15.2** | 10.5 | 8.4 | 7.8* | 14.4 | 14.6 | **11.7** |
| Demirkol | 20 × 15 | 36.3 | – | – | 18.0* | – | 35.8 | – | – | 26.1 | **22.9** | – | 12.0* | – | – | 22.6 | **18.4** |
| | 20 × 20 | 34.4 | – | – | 19.4* | – | 32.8 | – | – | 21.5 | **20.3** | – | 13.5* | – | – | 21.5 | **16.5** |
| | 30 × 15 | 37.8 | – | – | 21.8* | – | 38.8 | – | – | 27.6 | **27.1** | – | 14.4* | – | – | 29.4 | **23.1** |
| | 30 × 20 | 38.0 | – | – | 25.7* | – | 36.0 | – | – | 29.9 | **27.9** | – | 17.1* | – | – | 28.3 | **23.4** |
| | 40 × 15 | 34.6 | – | – | 17.5* | – | 35.5 | – | – | 26.2 | **25.5** | – | 11.7* | – | – | 28.4 | **20.2** |
| | 40 × 20 | 39.2 | – | – | 22.2* | – | 38.5 | – | – | 27.7 | **27.9** | – | 16.0* | – | – | 30.9 | **24.1** |
| | 50 × 15 | 33.2 | – | – | 15.7* | – | 34.1 | – | – | 27.4 | **25.0** | – | 11.2* | – | – | 29.5 | **21.7** |
| | 50 × 20 | 37.7 | – | – | 22.4* | – | 38.9 | – | – | 30.0 | **28.6** | – | 15.8* | – | – | 32.8 | **25.1** |
| | Mean | 36.4 | – | – | 20.3* | – | 36.3 | – | – | 27.0 | **25.7** | – | 14.0* | – | – | 27.6 | **21.6** |

[a] Used 128 samples for each instance during the sampling evaluation.
[b] Used beam search with a width of 16.
[c] Used 100 samples for each instance during the sampling evaluation.

and these results is that none of the additional baselines function on FJSP and only on JSP. These results show that CDQAC performs roughly equally to other RL baselines, such as CL (Iklassov et al., 2023) and Sched (Park et al., 2021), whereby CL slightly outperforms CDQAC. However, both CL and Sched require a training environment and do not work for FJSP. Similarly, we see that SL (Corsini et al., 2024) and GD (Pirnay & Grimm, 2024) , both self-labeling approaches, both outperform CDQAC. We need to note that both GD and SL are costly to train, with both requiring up to seven days of training on a GPU. In comparison, CDQAC can be trained in one or two hours, or even less, depending on the size of the training. Lastly, we note that IL (Lee & Kim, 2025), an Imitation Learning approach for JSP, achieves a performance similar to that of self-labeling approaches. Lee & Kim (2025) state that they used 4000 optimal solutions, found through constraint programming, to train IL. Their results do note that performance diminishes whenever it is trained on fewer solutions, whereby it achieves performance similar to CDQAC, if IL is trained on only 40 solutions. Moreover, IL requires optimal or near-optimal solutions, whereas CDQAC can be trained on any solution quality and does not require optimal solutions as training data.

### H.6 SIGNIFICANCE TEST

Our comparison for FJSP (Sect. 5.2) and JSP (Sect. 5.3) showed that CDQAC outperformed DANIEL (Wang et al., 2023) in most evaluations. To assess whether these results are significant, we conducted a one-sided Wilcoxon signed-rank test for both JSP and FJSP.

**FJSP.** Although CDQAC consistently outperformed DANIEL in most FJSP evaluations, the margins were smaller than in other results. To this end, we paired all results from Tables 3, 4, and 5, in both greedy and sampling evaluations. Furthermore, we paired the results of both $10 \times 5$ and $15 \times 10$ in Table 3, resulting in a sample size of 26 pairs. The statistical test yielded a $p \approx 0.018$ rejecting the null hypothesis of $p > 0.05$, indicating that CDQAC, trained solely on random data, significantly outperforms the online RL baseline DANIEL (Wang et al., 2023) in our FJSP evaluation.

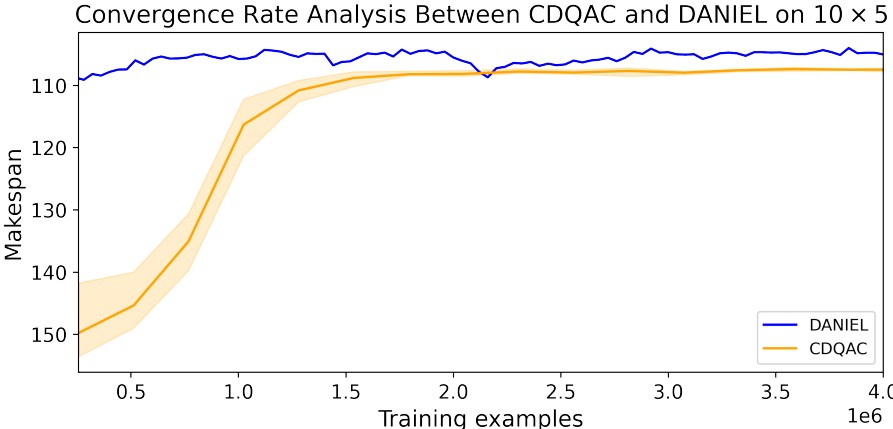

Figure 8: Convergence comparison between CDQAC and DANIEL for $10 \times 5$. The x-axis shows the number of training examples each has seen until this point. Major distinction is that CDQAC is able to reuse all examples during training, whereas DANIEL cannot. CDQAC is the average of four seeds, with the shaded area, being the maximal and minimal evaluations. The DANIEL results are provided by Wang et al. (2023).

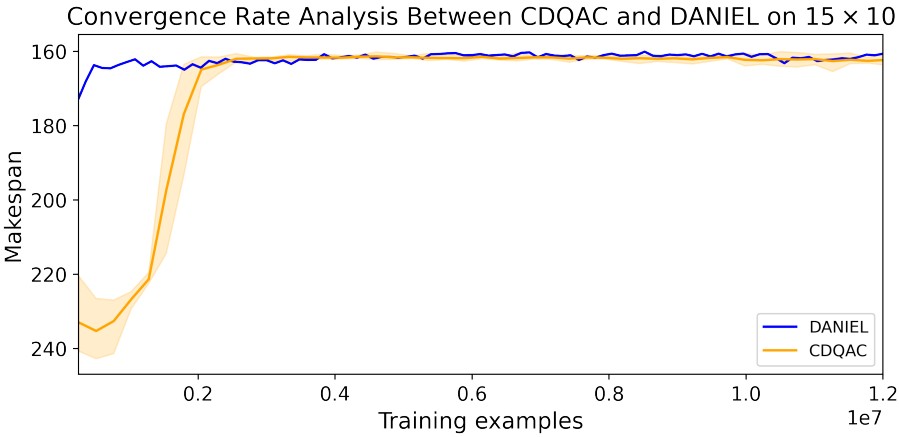

Figure 9: Convergence comparison between CDQAC and DANIEL for $15 \times 10$. The x-axis shows the number of training examples each has seen until this point. Major distinction is that CDQAC is able to reuse all examples during training, whereas DANIEL cannot. CDQAC is the average of four seeds, with the shaded area, being the maximal and minimal evaluations. The DANIEL results are provided by Wang et al. (2023).

**JSP.** To evaluate the significance of the JSP results, we again paired the results of CDQAC and DANIEL in Table 6, whereby we paired each Taillard result, both for greedy and sampling. This results in a sample size of 16 pairs. The Wilcoxon test resulted in $p \approx 0.00022$, indicating that CDQAC also significantly outperforms DANIEL on JSP.

## I   CONVERGENCE ANALYSIS

For further analysis on how CDQAC was able to outperform DANIEL (Wang et al., 2023), the best performing online RL method, we conducted a convergence analysis between CDQAC and DANIEL. For each evaluation step, we calculated the number of transitions each approach has seen. For CDQAC, this is the number of training steps multiplied by the batch size. For DANIEL, this is the number of episodes between each evaluation step, multiplied by the number of concurrent runs for each episode, which is multiplied by the number of PPO epochs and average episode length of

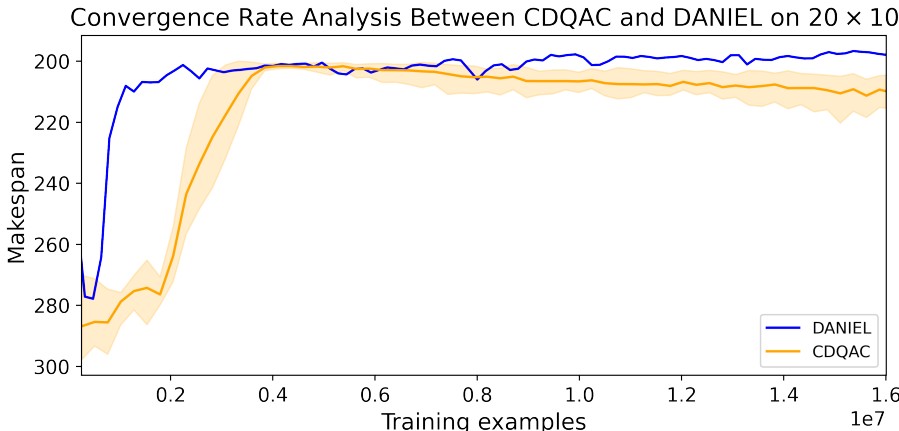

Figure 10: Convergence comparison between CDQAC and DANIEL for $10 \times 5$. The x-axis shows the number of training examples each has seen until this point. Major distinction is that CDQAC is able to reuse all examples during training, whereas DANIEL cannot. CDQAC is the average of four seeds, with the shaded area, being the maximal and minimal evaluations. The DANIEL results are provided by Wang et al. (2023).

each episode. DANIEL Wang et al. (2023) evaluates every 10 episodes, performs 20 concurrently runs each episode, and performs 4 PPO epochs, and the average episode length is $|J| \times |M|$, where $|J|$ and $|M|$ are the number of jobs and machines, respectively. An essential detail is that CDQAC will reuse transitions found in the training dataset, whereas DANIEL does not, and only trains on the transitions found in a single episode for 4 epochs. Therefore, the x-axis does not signify the number of *different* transitions trained on.

Figs. 8 and 9 show that CDQAC and DANIEL converge to a stable policy for both $10 \times 5$ and $15 \times 10$. Moreover, Fig. 8, in combination with the results in Table 4 indicate a stronger performance of DANIEL on the in-distribution instance set $10 \times 5$, while CDQAC was able to outperform DANIEL on the out-of-distrbution instance, namely the benchmark instances (Table 1) and larger generated instances (Table 5). This indicates that DANIEL is overtraining on $10 \times 5$, reducing its ability to generalize to instances that have a different distribution. For $15 \times 10$, we see in Fig 9 that CDQAC and DANIEL converge to similar performance. These results match those found in Table 4, where CDQAC was able to outperform DANIEL on $15 \times 10$. When comparing the results for the FJSP benchmarks (Table 3) when both CDQAC are trained on $15 \times 10$, we again notice that CDQAC outperforms DANIEL in most evaluations. This indicates that DANIEL is also overtraining for $15 \times 10$. Lastly, Fig. 10 shows that for $20 \times 10$ CDQAC is not able to converge to a stable policy, while DANIEL does, which matches the results in Table 4.

