# OpenReview forum: "Generalizing Beyond Suboptimality: Offline Reinforcement Learning Learns Effective Scheduling through Random Solutions"
_ICLR.cc/2026/Conference — Submitted to ICLR 2026_

### Official Review · Reviewer_PJtk · 2025-10-30

**Soundness:** 2
**Presentation:** 2
**Contribution:** 2
**Rating:** 4
**Confidence:** 5

**Summary:**

This paper introduces Conservative Discrete Quantile Actor-Critic (CDQAC), an offline reinforcement learning algorithm for job shop and flexible job shop scheduling. CDQAC aims to learn high-quality scheduling policies directly from static datasets generated by suboptimal heuristics such as random, genetic algorithm (GA), or dispatching rules, without relying on simulated environment interaction. The method combines a quantile-based critic, a delayed policy update, and a dueling network structure to model the return distribution conservatively and stably. Experiments on JSP and FJSP benchmarks show that CDQAC outperforms both existing offline and online RL baselines while maintaining strong generalization to larger instances.

**Strengths:**

The empirical results are comprehensive across JSP and FJSP benchmarks, demonstrating scalability and transferability. The finding that random datasets yield superior generalization provides a novel perspective on data diversity and pessimism in offline RL. Overall, the paper is clearly written, methodologically solid, and offers meaningful insights for applying offline RL to combinatorial optimization.

**Weaknesses:**

Despite the strong presentation, the empirical evaluation leaves several open concerns.
- The reliance on datasets generated from suboptimal heuristics raises doubts about the true generality of the learned policy, as performance improvements may stem from overly simple environments rather than genuine learning from poor data.
- The comparison set for both offline and online RL baselines is narrow—especially for online methods—and lacks sufficient training details, making fairness unclear.
- The claim that CDQAC trained on random data outperforms fully trained online PPO-based methods is difficult to accept without deeper evidence of equal training conditions or convergence verification.
- Similarly, the demonstration that five training trajectories suffice for good performance may reflect limited state diversity rather than strong sample efficiency.

**Questions:**

- The justification for using datasets generated by suboptimal heuristics is not sufficiently supported—how does training on random or poor-quality data yield policies that outperform those derived from expert or online learning methods?
- The experimental comparison includes very few offline RL baselines.
- Section 5.3 lacks details on the training configuration for online RL methods; it is unclear whether the models were fully trained to convergence under comparable computational budgets.
- The result that well-trained online RL performs worse than offline RL trained on static random data seems counterintuitive—additional diagnostic analysis could strengthen the argument.
- The conclusion that five trajectories suffice for strong generalization appears overly optimistic; more analysis is needed to show whether those trajectories capture meaningful dynamic diversity.
- The composition of the state vector is underexplained—what specific scheduling features (machine utilization, remaining processing time, job priority, etc.) are encoded, and how do they influence performance?
- The paper does not compare against classical dispatching rules or metaheuristics widely used in scheduling; including such baselines would improve the relevance and credibility of the results.

---

> ### Author Response · Authors · 2025-11-21
> **Response to Reviewer PJtk (Part 1/3)**
>
> Dear reviewer,
>
> We are pleased to read that you found our work methodologically sound and clearly written. We would like to thank you for your comments, which have helped us significantly improve the paper. In the following, we address your questions and concerns.
>
> ---
> ### W1: Reliance on suboptimal datasets
>
> We would like to emphasize that our datasets are not “poor” in quality; rather, they contain suboptimal but valid solutions. Our environment follows the setup proposed by **DANIEL** [1], which uses 26 state features (see Table 7 in App. B). Moreover, both JSP and especially FJSP are highly complex NP-Hard combinatorial optimization problems. Therefore, our performance does not stem from over-simplified environments, since our state space has either comparable or more features than standard offline RL benchmarks [2].
>
> Our findings, that a dataset with broad coverage (Random) can outperform a narrow, near-optimal dataset (GA), are supported by prior theoretical offline RL work [3–5], which establishes conditions under which this can occur, namely if the training dataset has sufficient coverage. We show that Random has a significantly higher coverage in Table 2 in the main paper and explain these results on lines 365 to 407.
>
>
> ### W2: Additional Offline and Online RL baselines
> In the revised version, we have added additional comparisons with baseline methods. For offline RL, we compare with Behavioral Cloning (BC) and Implicit Q-Learning (IQL) [6]. For online RL, we compare with Residual [7]. In the following, we first discuss the results of offline baselines, followed by online methods.
>
> **Table 1:** Improved Offline Comparison, with Implicit Q-learning (IQL) and Behavioral Cloning (BC). It reports the average gap (%) on all FJSP evaluation sets.
> |Eval| Method | PDR | GA | PDR‑GA | Random |
> |---|---|---|---|---|---|
> | Greedy | BC | 29.1 ± 3.2 | 13.9 ± 0.6 | 22.3 ± 2.2 | 21.8 ± 2.5 |
> || Offline‑LD (mQRDQN) | 22.2 ± 2.4 | 30.8 ± 3.5 | 21.8 ± 3.6 | 21.4 ± 2.6 |
> || Offline‑LD (d‑mSAC) | 23.2 ± 3.0 | 21.0 ± 2.1 | 25.9 ± 2.2 | 16.9 ± 1.8 |
> || IQL | 19.9 ± 1.8 | 20.6 ± 2.1 | 19.2 ± 2.3 | 21.3 ± 3.5 |
> || CDQAC (Ours) | **12.3 ± 1.7** | **13.0 ± 2.1** | **11.3 ± 1.3** | **10.6 ± 0.5** |
> | Sampling | BC | 10.7 ± 0.9 | 8.3 ± 0.1 | 9.4 ± 0.5 | 13.1 ± 0.0 |
> || Offline‑LD (mQRDQN) | 13.6 ± 0.2 | 14.2 ± 0.2 | 13.6 ± 0.1 | 13.6 ± 0.2 |
> || Offline‑LD (d‑mSAC) | 11.6 ± 1.3 | 8.8 ± 0.6 | 11.6 ± 1.2 | 7.7 ± 0.8 |
> || IQL | 10.0 ± 0.5 | 9.1 ± 0.5 | 9.4 ± 0.5| 10.7 ± 0.7 |
> || CDQAC (Ours) | **6.5 ± 0.7** | **6.4 ± 0.8** | **5.8 ± 0.5** | **5.8 ± 0.3** |
>
> Table 1 presents the updated comparison. CDQAC consistently outperforms IQL and BC by a substantial margin. IQL is only slightly better compared to Offline-LD, however, significantly worse compared to CDQAC on all datasets. BC trained on GA achieves gaps of 13.9% (greedy) and 8.3% (sampling), which remain worse than CDQAC trained on GA (13.0% and 6.4%). This difference aligns with prior observations [3]: BC degrades substantially when the initial state distribution changes. This explains the gap between CDQAC and BC, since the initial distributions differ across instances.
>
> CDQAC’s advantage is further underscored by the fact that it trains effectively on all datasets, while BC performs reasonably well only on GA. As discussed in the original submission, most offline RL methods are designed for continuous action spaces or are incompatible with the instance-dependent action spaces of FJSP/JSP.
>
> For online RL, we add **Residual** [6], which functions on both JSP and FJSP. We retrained Residual following the same protocol used for CDQAC, DANIEL, and FJSP-DRL. Table 2 shows that CDQAC significantly outperforms Residual across all evaluations.
>
> **Table 2:** Average Gap on all evaluations between Residual and CDQAC on Greedy (G) and Sampling (S). The Table is the table in the main paper.
>
> | Evaluation | Residual (G) | CDQAC (G) | Residual (S) | CDQAC (S) |
> |------------|--------------|-----------|--------------|-----------|
> | **FJSP Benchmarks (Table 3)** | 11.61 | **9.46** | 8.45 | **5.14** |
> | **FJSP Generated (Table 4)** | 13.72 | **9.00** | 10.68 | **4.54** |
> | **FJSP Generalization (Table 5)** – (30×10 & 40×10) | 12.99 | **3.80** | 12.90 | **2.66** |
> | **JSP (Table 6)** – (Taillard + Demirkol) | 22.00 | **20.45** | 21.10 | **16.65** |

---

> ### Author Response · Authors · 2025-11-21
> **Response to Reviewer PJtk (Part 2/3)**
>
> ### W3: Equal training conditions and convergence verification
> We fully agree that a fair comparison is essential. Our experimental setup (Section 5) already specified that CDQAC and all online RL baselines, DANIEL, FJSP-RL, and the additional Residual baseline, are trained on instances generated from the same distribution. The instance-generation process is described on lines 267 and 272, and we explicitly note on lines 303 and 316 that all baselines are trained on data sampled from this same distribution. Consequently, CDQAC does not benefit from exposure to any different or broader set of training instances. Moreover, CDQAC and all baselines utilize the same validation set, ensuring a fair comparison.
>
> We have also provided training curves in App. G and a convergence analysis in App. I, where we directly compare the learning convergence of DANIEL and CDQAC.
>
>
> ### W4: Using 5 Trajectories.
> We want to clarify that we never train on only 5 trajectories. When we reduce the training set to only 1%, we use 5 instances, with 100 solutions each, as shown in Fig. 3, Section 5.4, and App H.4. For the smallest instance size, 10x5, this still will be around **25,000** transitions. The convergence rate, in App. I, shows that CDQAC converges after **250,000** (not necessarily unique) samples, we see that this is 10% of the required data. This means CDQAC will see an example only 10 times on average before converging. Furthermore, as an off-policy bootstrapped method, CDQAC continually refines the target for each transition through the dueling quantile critic. Thus, each replayed transition provides a progressively more informative learning signal, enabling CDQAC to extract significantly more value from limited data
>
> Moreover, [8] showed that they can train effectively for JSP with only **10** expert trajectories. Any JSP instance has around $(n!)^m$ possible solutions [10], with $n$ the number of jobs and $m$ the machines. FJSP itself is way more complex, because we not only decide the sequence of operations, but also on which machine the operation will run on. The state space is even larger than the number of solutions, as it encompasses all possible partial solutions for each instance, given that CDQAC is a **constructive** RL approach.
>
>
> ### Q1: Datasets generated with suboptimal heuristics
> We thank you for raising this question, which we can explain based on our results. CDQAC, an offline RL approach, prefers a suboptimal training dataset with a wide coverage, meaning diverse state and action pairs, over narrow optimal datasets. Prior work explains this with two reasons. (1) **Intristic Uncertainty** [4], which means how uncertain an offline RL method is, depending on the absence of state–action pairs from the optimal policy in the training set. A wider coverage reduces this uncertainty. (2) However, a dataset generated with optimal examples is not enough [3,4], since a wider dataset confirms **pessimism**, the CQL regression, allowing for more accurate learning. Moreover, [8] also showed this previously, whereby they achieved better performance with a sub-optimal dataset than with an optimal one for JSP.
>
> ### Q2: Few offline baselines
> As previously mentioned, we have included IQL and BC as additional offline baselines. The results can be found in Table 1. We cannot include additional offline baselines, since they either only function for a continuous action space, ours is discrete, or cannot learn for JSP/FJSP instance-dependent state and action space size.
>
> ### Q3: Training configuration for online RL methods
> We agree that this is important and therefore included them in the original submission in Section 5, when we discuss our experimental setup. In our current version, we have expanded on it. We also previously stated this in our rebuttal. Moreover, our convergence analysis, in App. I show that CDQAC and DANIEL, the best-performing online counterparts, converge to a stable policy under similar computational budgets.

---

> ### Author Response · Authors · 2025-11-21
> **Response to Reviewer PJtk (Part 3/3)**
>
> ### Q4: CDQAC outperforming online RL.
>
> To assess why CDQAC outperformed DANIEL, we provide a convergence analysis in App. I. The figures show that CDQAC and DANIEL both converge to a stable policy for both 10x5 and 15x10, with the same validation instance set. For 10×5, we observe that DANIEL converges to a slightly better policy than CDQAC, whereas for 15×10, CDQAC and DANIEL converge to a similar performance.
>
> These observations are consistent with Table 3, which summarizes the results from Tables 3, 4, and 5 in the main paper. Table 3 shows DANIEL outperforms CDQAC on **in-distribution** 10×5 instances, and CDQAC surpasses DANIEL on 15×10. On the **out-of-distribution** evaluation sets, CDQAC consistently exceeds DANIEL’s performance, with the exception of the sampling-based evaluation on the 10×5 benchmark. Overall, CDQAC demonstrates stronger generalization to **out-of-distribution** instances.
>
> **Table 3:** Comparison between CDQAC and DANIEL on in-distribution and out-of-distribution instances.
> | Setting | Size | Method | Greedy (%) | Sampling (%) |
> | --- | --- | --- | --- | --- |
> | **Generated (In-Distribution)** | **10×5** | DANIEL | **10.87** | **5.57** |
> |  |  | CDQAC | 11.56 | 5.98 |
> |  | **15×10** | DANIEL | 12.42 | 6.79 |
> |  |  | CDQAC | **11.10** | **5.85** |
> | **Benchmarks (Out-of-Distribution)** | **10×5 (trained)** | DANIEL | 10.59 | **5.26** |
> |  |  | CDQAC | **9.22** | 5.50 |
> |  | **15×10 (trained)** | DANIEL | 10.30 | 5.28 |
> |  |  | CDQAC | **9.69** | **4.78** |
> | **Large Generated (Out-of-Distribution) Trained 10x5** | **30×10** | DANIEL | 5.10 | 4.43 |
> |  |  | CDQAC | **4.43** | **3.11** |
> |  | **40×10** | DANIEL | 3.65 | 3.77 |
> |  |  | CDQAC | **3.17** | **2.21** |
>
> ### Q5: Meaningful information in 5 trajectories
> We want to clearly repeat that we never claim to train on only 5 trajectories, but rather 5 instances with 100 random solutions each, resulting in at least **25,000** transitions with the smallest training size, 10x5. These contain enough information, since our results in Section 5.4 show no significant performance drop. App. I shows that CDQAC requires approximately 250,000 training examples to converge, which means that CDQAC will observe each transition only 10 times on average if trained on 1% of the data before converging.
>
> ### Q6: State Vector Composition
> We included this in our original version in Table 7 in App. B, where we show that we used 26 features, based on DANIEL [1]. The primary contribution of this work lies in the offline RL methodology rather than in proposing a new state representation. We therefore adopt the established state formulation to enable a fair comparison between CDQAC and DANIEL, ensuring that observed performance gains can be attributed to the offline RL approach rather than the state design. This choice is consistent with prior work, such as Offline-LD [8], which evaluates its method using the state representation introduced in L2D [9].
>
> ### Q7: Comparison against classical approaches.
> We agree that this is important and therefore included them in our original paper, where we compare them to classical dispatching rules and metaheuristics. For example, we compared against Genetic Algorithm (GA), and MOR-SPT and MOR-EST in Tables 3, 4, and 5. In addition, we compared to Constraint Programming in Table 3. In Table 6, the JSP results, we compared against classical dispatching rules MOR and MWR. However, we agree that additional baselines could be included for JSP. Therefore, we included CP and Mixed-Integer Programming (MIP), both exact methods, for JSP with a time limit of 30 minutes. Table 4 shows that CDQAC outperforms MIP on $50 \times 15$ and $50 \times 20$, and both MIP and CP on the largest instance size of Taillard ($100 \times 20$). This indicates that CDQAC can scale well to larger instance sizes compared to exact methods.
>
> **Table 4:** JSP results on the largest Taillard set
> |Size|CDQAC Greedy|CDQAC Sampling|MIP|CP|
> |---|---|---|---|---|
> |50x15|13.0|9.9|12.2|3.0|
> |50x20|12.8|11.0|13.6|2.8|
> |100x20|5.3|3.6|11.0|3.9|
>
> ---
> ## References
> [1] Wang et al. (2023). Flexible job shop scheduling via dual attention network-based reinforcement learning. IEEE.
>
> [2] Fu et al. (2020). D4RL: Datasets for Deep Data-Driven RL. Arvix.
>
> [3] Kumar et al. (2022). When Should Offline RL Be Preferred Over Behavioral Cloning? ICLR.
>
> [4] Jin et al. (2021). Is Pessimism Provably Efficient for Offline RL? ICML.
>
> [5] Li et al. (2023). Survival Instinct in Offline RL. NeurIPS.
>
> [6] Kostrikov et al. (2021). Offline RL with Implicit Q-Learning. ICLR.
>
> [7] Ho et al. (2024). Residual Scheduling for Job Shop Scheduling. IEEE Access.
>
> [8] van Remmerden et al. (2025). Offline Reinforcement Learning for Learning to Dispatch for Job Shop Scheduling. Machine Learning.
>
> [9] Zhang et al. (2020). Learning to Dispatch for Job Shop Scheduling via Deep Reinforcement Learning. NeurIPS.
>
> [10] Pfitzer et al. (2018). Event-Driven Production Rescheduling in Job Shops. IEEE CASE.

---

> > ### Author Response · Authors · 2025-11-27
> >
> > Dear Reviewer PJtk,
> >
> > We would like to thank you again for your detailed review, which helped significantly improve our paper.
> >
> > As the December 3rd deadline approaches, we would like to kindly invite you to take a moment to review our rebuttal. We have addressed your comments in detail, clarified the points that may have been misunderstood, and updated our submission with additional results and improved clarifications.
> >
> > If our responses resolve your concerns, we would greatly appreciate your consideration in updating the score. Please don't hesitate to contact us if you require any additional clarification.
> >
> > Best regards,
> >
> > The Authors

---

### Official Review · Reviewer_Quv3 · 2025-10-31

**Soundness:** 2
**Presentation:** 3
**Contribution:** 3
**Rating:** 4
**Confidence:** 4

**Summary:**

This paper proposes Conservative Discrete Quantile Actor-Critic (CDQAC), a conservative distributional offline RL method for learning dispatching policies in JSP/FJSP. CDQAC combines a quantile critic with conservative value regularization and delayed policy updates to stabilize the learning process. The offline approach reports strong performance on FJSP/JSP benchmarks and the notable observation that policies trained on random datasets can sometimes outperform training on near-optimal trajectories.

**Strengths:**

This paper addresses exploration-free policy learning (offline RL) for large FJSP/JSP instances by combining distributional Q-learning with conservative regularization, dueling Quantile network, and delayed policy update.
It demonstrates sample efficiency and generalization across instance sizes in FJSP, one of the most challenging COPs.
The finding that random datasets can be effective for training is practically meaningful for dataset construction in scheduling.

**Weaknesses:**

JSP comparisons are incomplete and relatively weak. While FJSP results include several recent methods, JSP needs stronger baselines. In particular, compare against imitation learning approaches that learn machine–operation pairing policies without a simulator; the following work achieves SOTA on JSP:
- Lee, J. H., & Kim, H. J. (2024). Graph-based imitation learning for real-time job shop dispatcher. IEEE T-ASE.

Modern online RL baselines should be included as JSP baselines (not only in Related Work). For example:
– Park, J., Bakhtiyar, S., & Park, J. (2021). ScheduleNet. arXiv:2106.03051.
– Chen, R., Li, W., & Yang, H. (2022). TII 19(2):1322–1331.
– Iklassov, Z. et al. (2023). IJCAI, pp. 5350–5358.
These papers report the performance on the Taillard (1993) datasets.

The integration of quantile critic + dueling + conservative regularization + delayed policy updates is presented, but its novelty and motivation are under-emphasized. If this combined offline scheduling is first in this domain, authors need to state it explicitly, explain what had to be modified to make the components work together, and add ablations isolating each component’s effect, not only about quantile critic and dueling (Table 9).

The encoder adopts the DAN/DANIEL structure (Wang et al., 2023) and operation/machine features largely as is.

**Questions:**

1. As you note that CDQAC can outperform the heuristic that generated its training data, do policies trained on optimal schedules achieve near-optimal quality on the same instances?

2. Why can policies trained on random datasets outperform near-optimal data? Imitation learning is sensitive to dataset quality, yet CDQAC sometimes shows the opposite trend.
3. Against modern online RL and imitation learning, how close is CDQAC in solution quality?
4. Could you provide learning curves of CDQAC across training-set size and composition (random/PDR/GA/near-optimal mixtures)?

Typo
“Learning-based methods for Scheduling Problems” → capitalization (L80)
“Demirkol 80x20” → it may be 50x20 (L865)

---

> ### Author Response · Authors · 2025-11-21
> **Response to Reviewer Quv3 (Part 1/2)**
>
> Dear reviewer,
>
> We thank you for your detailed review, which helped improve the paper. We have corrected the typos you identified and addressed all your questions and concerns below.
>
> ---
> ### W1: Additional JSP baselines
>
> To strengthen our comparison, we have included **Residual**[1] for both online RL comparison FJSP (Tables 3, 4, 5) and JSP (Tables 6), since Residual also functions on both JSP and FJSP. Table 1 shows that CDQAC consistently achieves lower gaps and stronger generalization than Residual, outperforming it across all FJSP and JSP benchmarks.
>
> **Table 1:** Average Gap on all evaluations between Residual and CDQAC on Greedy (G) and Sampling (S). The Table is the table in the main paper.
>
> | Evaluation | Residual (G) | CDQAC (G) | Residual (S) | CDQAC (S) |
> |------------|--------------|-----------|--------------|-----------|
> | **FJSP Benchmarks (Table 3)** | 11.61 | **9.46** | 8.45 | **5.14** |
> | **FJSP Generated (Table 4)** | 13.72 | **9.00** | 10.68 | **4.54** |
> | **FJSP Generalization (Table 5)** – (30×10 & 40×10) | 12.99 | **3.80** | 12.90 | **2.66** |
> | **JSP (Table 6)** – (Taillard + Demirkol) | 22.00 | **20.45** | 21.10 | **16.65** |
>
>
> In your review, you also suggest additional JSP baselines. While we agree that this could further strengthen the empirical evaluation, such a comparison would not be fair, since CDQAC is able to do both FJSP and JSP, whereas the proposed baselines only do JSP. We described this focus on methods that can do both JSP and FJSP in our related work (lines 302 to 316). We state this in our experimental setup on lines 80 to 93.
>
> However, we agree that this comparison could strengthen our paper. We included additional baselines in Table 15 in App H.5, where we compare CDQAC to baselines that function for JSP, but not FJSP, since we still believe that it is important to include them. We included [2-4] in this comparison. We could not include [5], since the authors only report the results of 16 out of 80 Taillard instances and have their code not publicly available.
>
> ### W2: Ablation Study and novel components
> Thank you for highlighting the need to further emphasize the novelty of CDQAC and to expand the ablation study. Our original submission has ablations on the Dueling Quantile Critic and Delayed Policy Update. In the revised version, we have further added CQL Regression to this analysis.
>
> Table 2 shows that the Delayed Policy Update has the greatest impact on performance, increasing the average gap by 11.7% when removed. The full ablation results appear in Table 9 in Appendix H.1. We also strengthened our explanation of the novel aspects of CDQAC in Section 4 (lines 141–146).
>
> **Table 2**: Ablation study of the components of CDQAC. The results show how much, on average, the gap increased without the component.
> | Setup | **Baseline** | **No Dueling** | **No Quantile** | **No CQL** | **No Policy Delay** |
> |:---:|:---:|:---:|:---:|:---:|:---:|
> | Avg $\Delta$ | 0% | +4.3% | +2.86% | +1.93% | +11.7% |
>
> ### W3: Using DAN encoder
>
> Our main contribution lies in the offline RL method CDQAC, not in proposing a new neural encoder. Nevertheless, CDQAC includes novel architectural components, such as the Dueling Quantile Network, but our aim was to isolate the benefits of the RL method itself. Offline-LD [6] similarly adopts the encoder from L2D [7]. Adopting the DANIEL encoder [8] allows for a fair and controlled comparison between CDQAC and DANIEL, ensuring that improvements arise from the offline RL methodology rather than architectural differences.
>
> ### Q1: Training on optimal solutions
> Thank you for your interesting question. In offline RL, training on optimal solutions will likely **not** result in better performance, compared to other, more diverse, training datasets. Prior work [9] explains that this stems from the wider dataset confirming **pessimism**, the CQL regression. Therefore, a wider coverage allows for more accurate learning of the actual return distribution. This distribution is used by CDQAC to learn a new policy. Offline-LD [6] also confirms this empirically for JSP, where they used two types of datasets, both generated with optimal solutions through constraint programming, whereby they trained on the original datasets, and one where they added noise to the training dataset by switching scheduling operations in the solution. They found that when trained on this noisy dataset, Offline-LD achieves higher performance than when trained on the original training set. Our results show a similar trend, with CDQAC achieving significantly higher performance with PDR-GA than with GA. We explain these results on lines 365-407 in the paper.

---

> ### Author Response · Authors · 2025-11-21
> **Response to Reviewer Quv3 (Part 2/2)**
>
> ### Q2: Why can policies trained on random datasets outperform near-optimal data
> Thank you for the insightful question. In offline RL, training solely on optimal solutions does not necessarily improve performance compared with using more diverse datasets. We explain this on lines 365-407 in the paper. As explained in [9], broader dataset coverage strengthens pessimism confirmation (i.e., the CQL regression term), enabling more accurate estimation of the return distribution. CDQAC relies on this return distribution to learn a new policy. Another reason is **Intristic Uncertainty** [10], which means how uncertain an offline RL method is depending on the absence of state–action pairs from the optimal policy in the training set. Therefore, a wider coverage reduces **Intristic Uncertainty** and improves **Pessimism** confirmation.
>
>
> ### Q3: Solution quality difference between CDQAC, IL, and Online RL
> We agree that the comparison between offline RL, IL, and online RL is of great interest. Our original submission already compared CDQAC to online RL methods. For FJSP, Tables 3–5, in the main paper, include DANIEL, FJSP-DRL, and Residual; for JSP, Table 6 compares DANIEL and Residual, and also the offline RL approach Offline-LD, and its online counterpart L2D. Following your helpful suggestion, we also added additional JSP online RL baselines in Table 15 in App H.5.
>
> We further expanded the offline comparison by adding Behavioral Cloning (BC), also known as Imitation Learning (IL), and Implicit Q-Learning (IQL). Table 3 (below) shows that BC outperforms several offline RL baselines when trained on GA data, but CDQAC remains the best-performing method across all datasets. As discussed in [9], BC suffers from sensitivity to shifts in initial state distributions. This explains the performance gap between CDQAC and BC, since different instances have different initial states.
>
> **Table 3:** Improved Offline Comparison, with Implicit Q-learning (IQL) and Behavioral Cloning (BC). It reports the average gap (%) on all FJSP evaluation sets.
> |Eval| Method | PDR | GA | PDR‑GA | Random |
> |---|---|---|---|---|---|
> | Greedy | BC | 29.1 ± 3.2 | 13.9 ± 0.6 | 22.3 ± 2.2 | 21.8 ± 2.5 |
> || Offline‑LD (mQRDQN) | 22.2 ± 2.4 | 30.8 ± 3.5 | 21.8 ± 3.6 | 21.4 ± 2.6 |
> || Offline‑LD (d‑mSAC) | 23.2 ± 3.0 | 21.0 ± 2.1 | 25.9 ± 2.2 | 16.9 ± 1.8 |
> || IQL | 19.9 ± 1.8 | 20.6 ± 2.1 | 19.2 ± 2.3 | 21.3 ± 3.5 |
> || CDQAC (Ours) | **12.3 ± 1.7** | **13.0 ± 2.1** | **11.3 ± 1.3** | **10.6 ± 0.5** |
> | Sampling | BC | 10.7 ± 0.9 | 8.3 ± 0.1 | 9.4 ± 0.5 | 13.1 ± 0.0 |
> || Offline‑LD (mQRDQN) | 13.6 ± 0.2 | 14.2 ± 0.2 | 13.6 ± 0.1 | 13.6 ± 0.2 |
> || Offline‑LD (d‑mSAC) | 11.6 ± 1.3 | 8.8 ± 0.6 | 11.6 ± 1.2 | 7.7 ± 0.8 |
> || IQL | 10.0 ± 0.5 | 9.1 ± 0.5 | 9.4 ± 0.5| 10.7 ± 0.7 |
> || CDQAC (Ours) | **6.5 ± 0.7** | **6.4 ± 0.8** | **5.8 ± 0.5** | **5.8 ± 0.3** |
>
>
> ### Q4: Training Plots
> We appreciate your suggestion to include training plots, which indeed helped clarify why CDQAC performs less stable on $20 \times 10$. We have added these plots in Appendix G, covering CDQAC, Offline-LD, BC, and IQL, following the helpful suggestion by you and reviewer **PJtk**. The plots show that CDQAC converges quickly and stable on the $10 \times 5$ and $15 \times 10$ training sets. They also confirm that CDQAC is less stable on $20 \times 10\$; however, all other offline RL baselines show significantly more instability under the same conditions.
>
> ---
> We hope that we answered all your questions and addressed the concerns you raised. If not, please let us know. We would also like to thank you for your valuable advice, which significantly contributed to the improvement of the paper.
>
> ---
> ### References
> [1] Ho et al. (2024). Residual Scheduling for Job Shop Scheduling. IEEE Access.
>
> [2] Lee, J. H., \& Kim, H. J. (2024). Graph-based imitation learning for real-time job shop dispatcher. IEEE T-ASE.
>
> [3] Park, J., Bakhtiyar, S., & Park, J. (2021). ScheduleNet. arXiv:2106.03051
>
> [4] Iklassov, Z. et al. On the Study of Curriculum Learning for Inferring Dispatching Policies on the Job Shop Scheduling (2023). IJCAI, pp. 5350–5358
>
> [5] Chen, R., Li, W., \& Yang, H. (2022). A Deep Reinforcement Learning Framework Based on an Attention Mechanism and Disjunctive Graph Embedding for the Job-Shop Scheduling Problem TII 19(2):1322–1331
>
> [6] van Remmerden et al. (2025). Offline Reinforcement Learning for Learning to Dispatch for Job Shop Scheduling. Machine Learning, 114.
>
> [7] Zhang et al. (2020). Learning to Dispatch for Job Shop Scheduling via Deep Reinforcement Learning. NeurIPS.
>
> [8] Wang et al. (2023). Flexible job shop scheduling via dual attention network-based reinforcement learning. IEEE Transactions on Neural Networks and Learning Systems.
>
> [9] Kumar et al. (2022). When Should Offline RL Be Preferred Over Behavioral Cloning? ICLR.
>
> [10] Jin et al. (2021). Is Pessimism Provably Efficient for Offline RL? ICML.

---

> > ### Author Response · Authors · 2025-11-27
> >
> > Dear Reviewer Quv3,
> >
> > We would like to thank you again for your detailed review, which helped significantly improve our paper.
> >
> > As the December 3rd deadline approaches, we would like to kindly invite you to take a moment to review our rebuttal. We have addressed your comments in detail, clarified the points that may have been misunderstood, and updated our submission with additional results and improved clarifications.
> >
> > If our responses resolve your concerns, we would greatly appreciate your consideration in updating the score. Please don't hesitate to contact us if you require any additional clarification.
> >
> > Best regards,
> >
> > The Authors

---

### Official Review · Reviewer_fDH7 · 2025-10-31

**Soundness:** 3
**Presentation:** 3
**Contribution:** 3
**Rating:** 6
**Confidence:** 3

**Summary:**

This paper introduces Conservative Discrete Quantile Actor-Critic (CDQAC), a novel offline reinforcement learning algorithm designed for job shop and flexible job shop scheduling problems. CDQAC learns effective scheduling policies directly from datasets generated by suboptimal heuristics, eliminating the need for simulated environment training. The method leverages a quantile-based critic and delayed policy updates to estimate return distributions and improve stability. Extensive experiments show that CDQAC consistently outperforms both the heuristics that generated the training data and state-of-the-art offline and online RL baselines, achieving high sample efficiency and strong generalization—even when trained on data from random heuristics.

**Strengths:**

- CDQAC can learn highly effective scheduling policies even when trained only on data generated by random or suboptimal heuristics, demonstrating strong generalization capabilities
- The method eliminates the need for simulated environment interactions, making it practical for real-world scheduling problems where simulators may not be available
- The algorithm consistently outperforms both the heuristics used to generate the training data and state-of-the-art offline and online RL baselines across various scheduling benchmarks
- CDQAC achieves strong results with limited data, highlighting its efficiency and practicality for offline RL in industrial scheduling contexts

**Weaknesses:**

I am not sure whether the below points are actual weaknesses of the paper or whether they arise due to the lack of my knowledge of scheduling problems and related literature - I am happy to increase the score if these can be addressed & I will give myself a lower confidence to account for this.

1) The main reported metric is the optimality gap - i.e. the gap to some method that is even better. The immediate question is of course why not use the better method - if I am not mistaken then this one would take 30 mins to provide a better solution for any of the considered problems - 30 mins does not sound like a lot, especially when considering RL training time is likely higher (I believe it is not reported).
2) If I assume the better method is for some reason not applicable, i.e. because 30 mins is too long, then probably the training time of the RL algorithm does not matter as much - the time needed to solve is reported for some problems within seconds, which is likely only the inference time. I wonder then why training time does not seem to matter - maybe the policy can be reused across many new instances, while the "classical" solution needs to start from scratch always, but I am not sure.
3) Somewhere along the same lines: It seems that the main reason heuristics and RL methods are considered in these settings in the first place is because classical approaches do not scale to large problem sizes / complexities. I wonder if sufficiently large problem instances have been considered in the paper then, since any problem is solved better by a classical approach within just 30 minutes, which again does not sound like a lot (also 30 minutes is a relatively weak quantification; I don't know whether this uses the same GPU hardware, better would be to report FLOPS used by both methods). I presume it would make sense to consider a problem size where the best method "switches", i.e. where the optimality gap becomes negative since the classic approach cannot come up with a reasonable solution within 30 minutes, while the RL solution can.

I guess generally the paper could use a bit more clarifications on the constraints and requirements that arise in practice from these scheduling problems - at least for me some things remain unclear.

**Questions:**

see weaknesses

---

> ### Author Response · Authors · 2025-11-21
> **Response to Reviewer fDH7**
>
> Dear reviewer,
>
> We are excited to read that you see the benefit of our paper, especially in relation to real-world scheduling problems. In the following paragraphs, you can find all the answers to the questions and potential unclarities you raised.
>
> ---
> ### W1: Optimality Gap
> Your question about RL relative to CP highlights a key trade-off in scheduling: solution quality versus computational time. Exact methods, such as CP and Mixed-Integer Programming (MIP), can find high-quality solutions but often require minutes or hours, which is impractical for many real-world settings where decisions must be made in seconds. For these settings, Dispatching Rules, such as MOR-SPT and MWR, are more desirable. RL-based methods, such as CDQAC, act similarly to classical dispatching rules, where they select a single operation, in the case of JSP, or a machine-operation pair for FJSP, at each time step. Our results show that CDQAC can find solutions in a few **seconds**.
>
> We use the optimality gap rather than the makespan because the optimal makespan varies substantially across instances of the same size. Two instances of the same size may differ significantly in their optimal makespans (e.g., 1000 vs. 500), making raw makespan values inappropriate for direct comparison. The optimality gap provides a normalized score that is standard in scheduling research.
>
> For the generated instances (Tables 4 and 5 in the main paper), we computed optimal or near-optimal solutions using Constraint Programming (CP), with a time limit of 30 minutes. For benchmark instances (Tables 3 and 6 in the main paper), we used publicly available optimal solutions, or the best-known upper bounds, if no optimal solution is known. For these benchmark instances, finding the upper bound can take significantly longer than 30 minutes, potentially requiring **days** to find a solution. In the revised version, we added footnotes with links containing these reference solutions.
>
> ### W2: Training Time of RL
> You are indeed correct that training is not taken into account for the evaluation time. This follows similar experimental setups [1-5]. CDQAC is trained once on a fixed training set, after which inference is performed on entirely unseen instances (including larger sizes). Consequently, training is a one-time cost and not part of per-instance evaluation. CDQAC can thus be reused for any instance and size, even if trained on smaller instances. Exact solvers, such as CP, attempt to optimize a solution until they either find an optimal solution or reach their time limit.
>
> Following the suggestions of reviewers fDH7 and PJtk, we added learning curves in Appendix G. These curves show that CDQAC converges within 10k–50k training steps, corresponding to under 30 minutes for the largest training configuration (20×10). This demonstrates that training is computationally modest.
>
> ### W3: Exact methods on large instance sizes
>
> Following reviewer PJtk’s recommendation, we added CP and MIP baselines, both with a 30-minute time limit for the JSP benchmarks. Table 1 shows that CDQAC matches or outperforms MIP on 50×15 and 50×20, and surpasses both CP and MIP on the largest Taillard instances (100×20). This demonstrates that CDQAC scales more effectively than exact solvers as instance sizes increase. The optimality gap is not negative for CDQAC and CP, since Taillard is a well-known benchmark, for which it is standard to use the current best-known solution, which took significantly longer than **30 minutes** to find.
>
> FLOP comparison is not standard for this setting [1–5], because exact solvers run CPU-bound tree-search and constraint-propagation procedures that do not map cleanly to floating-point operations, while methods like CDQAC rely on GPU-based computations. Since these workloads differ fundamentally, FLOP counts are not comparable.
>
> **Table 1:** JSP results on the largest Taillard instances (reports Optimality Gap %)
> |Size|CDQAC Greedy|CDQAC Sampling|MIP|CP|
> |---|---|---|---|---|
> |50x15|13.0|9.9|12.2|3.0|
> |50x20|12.8|11.0|13.6|2.8|
> |100x20|5.3|3.6|11.0|3.9|
>
> ---
> Hopefully, our improved version and our explanation in this comment helped clarify these points. If not, please let us know, and we are happy to provide further information.
>
> ---
> ### References
> [1] Wang et al. (2023). Flexible job shop scheduling via dual attention network-based reinforcement learning. IEEE Transactions on Neural Networks and Learning Systems.
>
> [2] Song et al. (2023). Flexible job-shop scheduling via graph neural network and deep reinforcement learning. IEEE Transactions on Industrial Informatics,
>
> [5] van Remmerden et al. (2025). Offline Reinforcement Learning for Learning to Dispatch for Job Shop Scheduling. Machine Learning, 114.
>
> [4] Ho et al. (2024). Residual scheduling: A new reinforcement learning approach to solving job shop scheduling problem. IEEE Access.
>
> [5] Zhang et al. (2020). Learning to Dispatch for Job Shop Scheduling via Deep Reinforcement Learning. NeurIPS.

---

> ### Author Response · Authors · 2025-11-27
>
> Dear Reviewer fDH7,
>
> We would like to thank you again for your detailed review, which helped significantly improve our paper.
>
> As the December 3rd deadline approaches, we would like to kindly invite you to take a moment to review our rebuttal. We have addressed your comments in detail, clarified the points that may have been misunderstood, and updated our submission with additional results and improved clarifications.
>
> If our responses resolve your concerns, we would greatly appreciate your consideration in updating the score. Please don't hesitate to contact us if you require any additional clarification.
>
> Best regards,
>
> The Authors

---

### Official Review · Reviewer_JoUx · 2025-11-01

**Soundness:** 4
**Presentation:** 3
**Contribution:** 3
**Rating:** 8
**Confidence:** 4

**Summary:**

This paper presents Conservative Discrete Quantile Actor-Critic (CDQAC), an offline reinforcement learning algorithm for Job Shop Scheduling Problems (JSP) and Flexible Job Shop Scheduling Problems (FJSP). The method learns scheduling policies from static datasets without requiring environment interaction, using a quantile critic with dueling architecture and conservative Q-learning to prevent overestimation of out-of-distribution actions. CDQAC employs delayed policy updates and a Dual Attention Network to encode operations and machines. The algorithm demonstrates superior performance compared to offline and online RL baselines across standard benchmarks, achieving sample efficiency with only 10-20 training instances. The research reveals that CDQAC performs best when trained on random solutions rather than expert heuristics, attributed to higher state-action coverage (7.7× more than priority dispatching rules) enabling better solution stitching. The method generalizes to larger problem instances and maintains stability even with 1% of original training data, challenging conventional offline RL assumptions about data quality requirements for combinatorial optimization problems.

**Strengths:**

1. A good summary of related work in the area, and setting the context of the relevance of the proposed work clearly
2. Learning from offline trajectories, especially lower quality ones like random or genetic algorithms, is an important problem tackled in the paper.
3. The problem setting is explained clearly, the evaluations use state-of-the-art algorithms as baseline
4. Expansion from job scheduling to flexible job scheduling is significant. Expansion to varied number of state-action spaces, although not new, is both a good characteristic of the solution, and its generalizability has been evaluated, limitations have been noted.

**Weaknesses:**

1. An ablation study is missing. The proposed solution has many different components, and it is unclear which of these components are critical to the overall performance.
2. Data used is synthetic, the experiment results will be stronger if real-world data were used.

**Questions:**

1. What is the computational requirements, training time, memory usage, and inference latency?
2. How were the hyper-parameters determined?
3. Why do you need to use to asymmetric Huber loss? In what way is it asymmetric? Have you compared it with the standard L2 loss?
4. What is the purpose of the dueling quantile network? Would the algorithm still be performant if we remove it?
5. What is the sensitivity to the number of quantiles you use? What happens if you reduce the quantiles to as few as 3 or 4?

---

> ### Author Response · Authors · 2025-11-21
> **Response to Reviewer JoUx (Part 1/2)**
>
> We thank you for your review and that you see the benefit of our work. Your review helped to improve our paper significantly. In the following, we answer the questions and potential weaknesses that you raised.
>
> ---
> ### W1: Missing Ablation Study
>
> The original submission included an ablation study for both the critic design and the delayed policy update in Appendix H.1 (Table 9). Following your suggestion and in line with reviewer Quv3’s request, we extended this study to also ablate the CQL coefficient $\alpha_{\text{CQL}}$ and report the percentage increase in average Gap when removing each component. The main results are summarized in Table 1. The Delayed Policy Update is the most essential component, since without it the average Gap increases by **11.7%**.
>
> **Table 1**: Ablation results (\% increase in average Gap).
> | Setup | **Baseline** | **No Dueling** | **No Quantile** | **No CQL** | **No Policy Delay** |
> |:---:|:---:|:---:|:---:|:---:|:---:|
> | Avg $\Delta$ | 0% | +4.3% | +2.86% | +1.93% | +11.7% |
>
> ### W2: Synthetic Data
> The purpose of the paper is to test whether competitive scheduling policies can be learned purely from trajectories generated by suboptimal heuristics, thereby removing the need for a training environment. We used synthetic training data to ensure parity with the training setup of the online RL baselines FJSP-DRL, DANIEL, and the newly included Residual. This allows for a clean comparison where all methods learn from the same instance distribution.
>
> We then evaluate CDQAC on both in-distribution instances, similar to the training dataset, and out-of-distribution instances, which are significantly different from the training dataset. Table 3 presents in-distribution results, while Tables 2 and 4 report out-of-distribution performance, with Table 4 focusing on large instances. Table 5 provides additional results on standard JSP benchmarks. These benchmark sets (Tables 2 and 5) are based on real-world scheduling problems and are widely used in prior work [1–5]. We hope that future work can use this work for scheduling problems where using a training environment is either infeasible or where training steps are computationally expensive.

---

> ### Author Response · Authors · 2025-11-21
> **Response to Reviewer JoUx (Part 2/2)**
>
> ### Q1: Training setup and latency
> We discussed the training setup in Section 5 between lines 317 and 339. The training time for each CDQAC was around 2.5 hours for $10 \times 5$, 3 hours for $15 \times 10$, and 4 hours for $20 \times 10$. However, we also added training plots in Appendix G.5, which showed that CDQAC converged in significantly fewer training steps than the original 200,000, which means that we required around 5 to 10 minutes for $10 \times 5$, 10 to 15 minutes for $15 \times 10$, and 20 to 30 minutes for $20\times 10$. The RAM usage depended on the training size, where we used most of the 360GB RAM during training for $20 \times 10$, since we loaded the whole dataset into it; however, this is not required and would only likely increase the training time slightly. The evaluation latency for CDQAC is noted in Tables 3, 4, and 5. These show that there is no significant difference between online RL approaches, such as FJSP-DRL and DANIEL. For example, in Table 4, for size $15 \times 10$, CDQAC inference time is 1.16 seconds compared to 1.43 seconds for FJSP-DRL, and 1.35 seconds for DANIEL.
>
> ### Q2: Hyperparameters determined
> Most hyperparameters where determined by **manual hyperparameter tuning**. The exceptions are the DAN encoder, actor and critic network architecture, which we used the same setting as [1], who introduced the DAN encoder. Our ablation study in Table 2 also included different settings of the Policy Update Frequency $\eta$ and CQL alpha $\alpha_\text{CQL}$, which shows that performance increases with higher values of $\eta$ and either increasing or setting $\alpha_\text{CQL}=0$ decreases performance.
>
> **Table 2**: Ablation results (\% increase in average Gap).
> | Setup | **Baseline** | **No Dueling** | **No Quantile** | **No CQL** | **No Policy Delay** |
> |:---:|:---:|:---:|:---:|:---:|:---:|
> | Avg $\Delta$ | 0% | +4.3% | +2.86% | +1.93% | +11.7% |
>
> ### Q3: Asymmetric Huber Loss
> The critic learns $N$ quantiles of the return distribution and must ensure monotonic ordering across quantiles. The Asymmetric Huber loss [6] enforces this by penalizing over- and under-estimation differently for each quantile (e.g., stronger penalties for overestimation of the first quantile and underestimation of the last). The Huber loss itself is a piecewise combination of L1 and L2 loss, using the L1 loss for large losses, and L2 for small losses. This improves stability compared to standard L1 or L2 loss [6]. We did compare to L2 loss, which is the **No Quantile** setting in Table 1.
>
> ### Q4: Purpose of Dueling Quantile Critic
> The Dueling Quantile Critic is required for CDQAC, since the critic learns the value of each scheduling action, through which the policy learns a new scheduling policy that maximizes the return of the critic as explained in Section 4. Table 1 shows that specific components of the critic, the Dueling and Quantile, are not strictly required; however, without them, the performance decreases by **4.3%** without Dueling, and **2.86%** without the Quantile setup. The combination of both the Dueling and Quantile allows for more accurate learning, and thus CDQAC's ability to find a new scheduling policy.
>
> ### Q5: Sensitivity to number of quantiles.
> Thank you for asking this interesting question; based on it, we conducted additional experiments. Table 3 shows that performance can differ slightly with the number of quantiles. Interestingly, we found that when $N=16$, performance is better than with $N=32$. This occurs because the critic learns the return distribution of each state–action pair; therefore, the appropriate number of quantiles is task-dependent [6], and the “optimal” choice is not necessarily the largest one.
>
>
> **Table 3:** Effect of decreasing the number of quantiles $N$ in CDQAC
> |Number of Quantiles|**64 (Baseline)**|**4**|**8**|**16**|**32**|
> |:---:|:---:|:---:|:---:|:---:|:---:|
> |Avg Δ|0%|+1.50%|+2.41%|+0.29%|+1.41%|
>
> ---
> We hope this clarified the questions you raised. If not, we are happy to provide additional clarification.
>
> ---
> ### References
> [1] Wang et al. (2023). Flexible job shop scheduling via dual attention network-based reinforcement learning. IEEE Transactions on Neural Networks and Learning Systems.
>
> [2] Song et al. (2023). Flexible job-shop scheduling via graph neural network and deep reinforcement learning. IEEE Transactions on Industrial Informatics,
>
> [5] van Remmerden et al. (2025). Offline Reinforcement Learning for Learning to Dispatch for Job Shop Scheduling. Machine Learning, 114.
>
> [4] Ho et al. (2024). Residual scheduling: A new reinforcement learning approach to solving job shop scheduling problem. IEEE Access.
>
> [5] Zhang et al. (2020). Learning to Dispatch for Job Shop Scheduling via Deep Reinforcement Learning. NeurIPS.
>
> [6] Dabney et al. (2018). Distributional RL with Quantile Regression. AAAI.

---

> > ### Comment · Reviewer_JoUx · 2025-11-25
> > **Thank you for the detailed responses**
> >
> > I'm satisfied with the responses, they adequately answer my questions. I'll keep my current score as it is already high.

---

> ### Author Response · Authors · 2025-11-27
>
> Dear Reviewer JoUx,
>
> We would like to thank you again for your in-depth review, which helped us improve our work. We are pleased to see that we adequately addressed all your questions and that you will maintain your high score. We hope that you can continue to support our work in any future stages. Thank you again for your engagement and helpful suggestions.
>
> Kind regards,
>
> The Authors

---

### Author Response · Authors · 2025-11-21
**Message to all Reviewers**

We want to thank all the reviewers for their valuable feedback. All their reviews significantly helped us improve the paper. Below, we outline the changes made in our revised version.

## Extended Ablation Study
On suggestions by **JoUx** and **Quv3**, we expanded the ablation study by adding CQL regression and quantifying the contribution of each component. Results show the delayed policy update is the most critical. Table 1 provides the summary; complete results are in App. H.1.

 **Table 1**: Ablation results (\% increase in average Gap).

| Setup | **Baseline** | **No Dueling** | **No Quantile** | **No CQL** | **No Policy Delay** |
|:---:|:---:|:---:|:---:|:---:|:---:|
| Avg $\Delta$ | 0% | +4.3% | +2.86% | +1.93% | +11.7% |


## Offline Baselines
Based on input from **Quv3** and **PJtk**, we added **Behavioral Cloning (BC)** and **Implicit Q-Learning (IQL)** as offline baselines. CDQAC outperforms all offline methods. Additional offline baselines are unsuitable because they assume continuous actions or cannot handle the instance-dependent action spaces in JSP/FJSP.

   **Table 2:** Offline comparison (average gap %, all FJSP evaluation sets).

|Eval| Method | PDR | GA | PDR‑GA | Random |
|---|---|---|---|---|---|
| Greedy | BC | 29.1 ± 3.2 | 13.9 ± 0.6 | 22.3 ± 2.2 | 21.8 ± 2.5 |
|| Offline‑LD (mQRDQN) | 22.2 ± 2.4 | 30.8 ± 3.5 | 21.8 ± 3.6 | 21.4 ± 2.6 |
|| Offline‑LD (d‑mSAC) | 23.2 ± 3.0 | 21.0 ± 2.1 | 25.9 ± 2.2 | 16.9 ± 1.8 |
|| IQL | 19.9 ± 1.8 | 20.6 ± 2.1 | 19.2 ± 2.3 | 21.3 ± 3.5 |
|| CDQAC (Ours) | **12.3 ± 1.7** | **13.0 ± 2.1** | **11.3 ± 1.3** | **10.6 ± 0.5** |
| Sampling | BC | 10.7 ± 0.9 | 8.3 ± 0.1 | 9.4 ± 0.5 | 13.1 ± 0.0 |
|| Offline‑LD (mQRDQN) | 13.6 ± 0.2 | 14.2 ± 0.2 | 13.6 ± 0.1 | 13.6 ± 0.2 |
|| Offline‑LD (d‑mSAC) | 11.6 ± 1.3 | 8.8 ± 0.6 | 11.6 ± 1.2 | 7.7 ± 0.8 |
|| IQL | 10.0 ± 0.5 | 9.1 ± 0.5 | 9.4 ± 0.5| 10.7 ± 0.7 |
|| CDQAC (Ours) | **6.5 ± 0.7** | **6.4 ± 0.8** | **5.8 ± 0.5** | **5.8 ± 0.3** |

## Expanded Online RL Baselines
We included an additional online RL baseline, **Residual**, retrained under our setup, to ensure comparison with DANIEL, FJSP-DRL, and CDQAC. Table 3 shows that **Residual** performs worse than both CDQAC. We also extended our explanation on why CDQAC is able to outperform online RL, based on the advice of **PJtk** and **Quv3**.

**Table 3:** Average Gap on all evaluations between Residual and CDQAC on Greedy (G) and Sampling (S).

| Evaluation | Residual (G) | CDQAC (G) | Residual (S) | CDQAC (S) |
|--|---|--|--|--|
| **FJSP Benchmarks (Table 3)** | 11.61 | **9.46** | 8.45 | **5.14** |
| **FJSP Generated (Table 4)** | 13.72 | **9.00** | 10.68 | **4.54** |
| **FJSP Generalization (Table 5)**| 12.99 | **3.80** | 12.90 | **2.66** |
| **JSP (Table 6)**  | 22.00 | **20.45** | 21.10 | **16.65** |

Reviewer **Quv3** suggested including additional learning-based JSP approaches, which we have added to Appendix H.5. However, we opted not to include these comparisons in the main results, as our primary focus is on approaches capable of handling **both** JSP and FJSP, specifically DANIEL, Residual, and CDQAC. Comparing to methods that function on JSP, but not FJSP, would not be a fair comparison. This is stated in the Related Work and is now emphasized more clearly in the Experimental Setup.

## Additional Classical Baselines for JSP
**PJtk** suggested including comparisons with classical dispatching rules and additional heuristic methods. To support this point, we clarify that the paper already reports results for both Genetic Algorithms (GA), the two best performing dispatching rules, MOR-SPT and MOR-EST for FJSP, and MOR and MWR for JSP. In addition, we also compared with CP for FJSP. We have added **CP** and **MIP** (with a 30-minute time limit) for JSP benchmarks. Table 5 shows that CDQAC is able to match or even outperform these exact methods for the largest instance sizes. We included these results in Table 6 in the revised paper.

   **Table 5:** JSP results on the largest Taillard instances
|Size|CDQAC Greedy|CDQAC Sampling|MIP|CP|
|---|---|---|---|---|
|50x15|13.0|9.9|12.2|3.0|
|50x20|12.8|11.0|13.6|2.8|
|100x20|5.3|3.6|11.0|3.9|

## Training Curves and Convergence
Following suggestions from **Quv3** and **PJtk**, we added learning curves for CDQAC, BC, IQL, and Offline-LD (App. G), showing fast convergence (10k–50k steps) and significantly greater stability for CDQAC. App. I compare the convergence between CDQAC and DANIEL.

## Training Setup
We improved our Experimental Setup (Sect. 5) by adding additional information about the metrics used, based on the recommendations of **fDH7**. We also expanded our explanation on the online RL baselines training configuration, based on the advice of **PJtk**. We ensure that CDQAC and all online RL approaches are trained under equal conditions. All methods are trained on instances generated from the same distribution for 10x5, 15x10, and 20x10, and use the same validation set.

---

### Author Response · Authors · 2025-12-02
**Final Remarks (Part 1/3)**

We thank all reviewers for their constructive feedback. All reviewers are positive about our contribution, which shows our proposed method, **CDQAC**, not only outperforms existing offline and online baselines but overturns assumptions about what low-quality data can achieve. Moreover, our contribution shows that we do not require a training environment for scheduling problems, as noted by reviewers **fDH7** and **PJtk**, and that CDQAC is sample efficient, as noted by all reviewers. The reviewers raised minor concerns about additional baselines and a question about equal training conditions. We have conducted extensive additional experiments — including new offline baselines (BC, IQL), a new online baseline (Residual), extended ablations, training-curve evaluations, and expanded JSP comparisons with MIP/CP and additional JSP baselines — to strengthen our submission. We also clarified that CDQAC and the online baselines are trained and evaluated under equal conditions.

---

## **Reviewer JoUx (Rating: 8)**
Reviewer **JoUx** provided a highly positive assessment of our contribution. They highlighted that our approach demonstrates superior performance relative to both offline and online RL baselines, and emphasized the sample efficiency of CDQAC, which achieves strong results even when trained on as few as 10 - 20 instances. JoUx further acknowledged our extension from JSP to FJSP, one of the most challenging combinatorial optimization problems, echoed by **Quv3**. **JoUx** raised concerns about missing ablation studies and real-world applicability, which we addressed as follows.

**1. Missing ablation study for CDQAC components**
- We expanded the original ablation study, found in Appendix H.1 (Table 9), substantially, adding ablations for the CQL regression term, and clarified that we included ablation studies on the Policy Delay and the Dueling Architecture. Table 1 shows that the Policy Delay has the largest impact on performance.
**Table 1**: Ablation results (% increase in average Gap).

| Setup | **Baseline** | **No Dueling** | **No Quantile** | **No CQL** | **No Policy Delay** |
|:---:|:---:|:---:|:---:|:---:|:---:|
| Avg $\Delta$ | 0% | +4.3% | +2.86% | +1.93% | +11.7% |

**2. Generalization and real-world applicability**
- We clarified the role of synthetic vs. benchmark data and emphasized that standard benchmarks (Taillard, Demirkol, Hurink, Brandimarte) represent real-world scheduling characteristics.

**3. Questions on computational requirements, latency, and memory**
- We explained the training costs and stated that we included the latency of CDQAC in Tables 3, 4, and 5 in the original submission.
- CDQAC converges far sooner than the originally allotted 200k steps (typically 10k–50k), and inference latency is on par with FJSP-DRL and DANIEL.

**JoUx** confirmed satisfaction with the revisions and kept their high score.

---

## **Reviewer fDH7 (Rating: 6)**
Reviewer **fDH7** emphasizes that CDQAC learns highly effective scheduling policies even from random or weak heuristics, demonstrating generalization, and highlights that eliminating the need for simulator interaction makes the method practical for real-world settings, which was also noted by **PJtk** and **JoUx**. **fDH7** further notes that CDQAC consistently outperforms both the heuristics that generated the training data and SOTA offline and online RL baselines, while achieving strong results with limited data. These points align with observations made by **JoUx**, who praised the expansion from JSP to FJSP and recognized the CDQAC’s ability to generalize across varied instance sizes, also mentioned by **Quv3**, who acknowledged CDQAC outperforming all offline and online baselines.

**fDH7** raised concerns about the experimental setup and how RL is trained for scheduling problems like JSP and FJSP. We addressed these concerns as follows:

**1. Clarity of evaluation metrics and setup**
- Section 5 was expanded to describe metrics and experimental setup more explicitly. We explained to reviewer **fDH7** the critical trade-off between runtime and performance.
- We explained that training is a one-time cost, and afterwards, CDQAC can find solutions for any instance, even larger sizes. Therefore, training time is not taken into account, similar to prior work.

**2. Scaling to larger instance sizes**
- We also stated that CDQAC is able to scale better to larger instance sizes compared to exact methods such as CP and MIP, as shown in Table 2.

**Table 2:** JSP results on the largest Taillard instances (reports Optimality Gap %)
|Size|CDQAC Greedy|CDQAC Sampling|MIP|CP|
|---|---|---|---|---|
|50x15|13.0|9.9|12.2|3.0|
|50x20|12.8|11.0|13.6|2.8|
|100x20|5.3|3.6|11.0|3.9|

---
Our rebuttal addressed all of fDH7’s concerns, and their questions helped us refine the clarity of our experimental description. **fDH7** stated he would increase his score if we addressed these concerns.

---

> ### Author Response · Authors · 2025-12-02
> **Final Remark (Part 2/3)**
>
> ## **Reviewer Quv3 (Rating: 4)**
> Reviewer **Quv3** states that CDQAC effectively tackles exploration-free policy learning for large-scale FJSP/JSP, noting that CDQAC exhibits strong sample efficiency and generalization across instance sizes. They also highlight the practically meaningful finding that schedules trained on random datasets can outperform those learned from near-optimal data — an insight similarly recognized by **fDH7**, who emphasized CDQAC’s ability to learn from any solution, even sub-optimal solutions, and by **JoUx**, who praised its ability to generalize from diverse suboptimal examples.
>
> **Quv3** raised concerns about additional baselines, both offline and online, reuse of the encoder network, and an incomplete ablation study. We show below how we addressed these concerns.
>
> **1. Need for more offline baseline comparisons**
> - Added **Behavioral Cloning (BC)** and **Implicit Q-Learning (IQL)** across all datasets.
> - We explained the difference between Imitation Learning, which is BC, and Offline RL. We also explained how CDQAC can outperform BC on the expert dataset, GA.
> - CDQAC outperforms all offline baselines on every dataset (PDR, GA, PDR-GA, Random), as seen in Table 3.
>
> **Table 3:** Improved Offline Comparison, with Implicit Q-learning (IQL) and Behavioral Cloning (BC). It reports the average gap (%) on all FJSP evaluation sets.
> |Eval| Method | PDR | GA | PDR-GA | Random |
> |---|---|---|---|---|---|
> | Greedy | BC | 29.1 ± 3.2 | 13.9 ± 0.6 | 22.3 ± 2.2 | 21.8 ± 2.5 |
> || Offline-LD (mQRDQN) | 22.2 ± 2.4 | 30.8 ± 3.5 | 21.8 ± 3.6 | 21.4 ± 2.6 |
> || Offline-LD (d-mSAC) | 23.2 ± 3.0 | 21.0 ± 2.1 | 25.9 ± 2.2 | 16.9 ± 1.8 |
> || IQL | 19.9 ± 1.8 | 20.6 ± 2.1 | 19.2 ± 2.3 | 21.3 ± 3.5 |
> || **CDQAC (Ours)** | **12.3 ± 1.7** | **13.0 ± 2.1** | **11.3 ± 1.3** | **10.6 ± 0.5** |
> | Sampling | BC | 10.7 ± 0.9 | 8.3 ± 0.1 | 9.4 ± 0.5 | 13.1 ± 0.0 |
> || Offline-LD (mQRDQN) | 13.6 ± 0.2 | 14.2 ± 0.2 | 13.6 ± 0.1 | 13.6 ± 0.2 |
> || Offline-LD (d-mSAC) | 11.6 ± 1.3 | 8.8 ± 0.6 | 11.6 ± 1.2 | 7.7 ± 0.8 |
> || IQL | 10.0 ± 0.5 | 9.1 ± 0.5 | 9.4 ± 0.5| 10.7 ± 0.7 |
> || **CDQAC (Ours)** | **6.5 ± 0.7** | **6.4 ± 0.8** | **5.8 ± 0.5** | **5.8 ± 0.3** |
>
> **2. Need for more online RL comparison**
> - We already benchmarked against DANIAL that is considered as a SOTA online DRL for FJSP.
> - Following suggestion from the reviewer, we now added a newly retrained **Residual**.
> - CDQAC consistently outperforms Residual across all FJSP and JSP evaluations as shown in Table 4.
>
> **Table 4:** Average Gap on all evaluations between Residual and CDQAC on Greedy (G) and Sampling (S). The table references the table in the submission.
>
> | Evaluation | Residual (G) | CDQAC (G) | Residual (S) | CDQAC (S) |
> |------------|--------------|-----------|--------------|-----------|
> | **FJSP Benchmarks (Table 3)** | 11.61 | **9.46** | 8.45 | **5.14** |
> | **FJSP Generated (Table 4)** | 13.72 | **9.00** | 10.68 | **4.54** |
> | **FJSP Generalization (Table 5)** – (30×10 & 40×10) | 12.99 | **3.80** | 12.90 | **2.66** |
> | **JSP (Table 6)** – (Taillard + Demirkol) | 22.00 | **20.45** | 21.10 | **16.65** |
>
> **3. Why random datasets outperform expert heuristics**
> - We improved our original explanation, which shows an empirical evaluation of **State-Action Coverage (SACO)** supported by multiple theoretical works.
>
> **4. Additional JSP learning-based baselines**
> - We included additional baselines that only function on **JSP**, in Appendix H.5, and clarified why our main comparisons focus on methods that handle both **JSP and FJSP**.
>
> **5. Improved Ablation Study**
> - We added the CQL coefficient to the ablation study, which already included the policy delay and network architecture.
> - We quantified the performance decrease without each component as shown in Table 1.
>
> **6. Reuse of DAN encoder**
> - We clarified that our main contribution is in the offline RL methodology by introducing CDQAC.
> - This allows for a fair comparison, since it enables us to isolate the benefit of our offline RL method.
> - CDQAC makes significant contributions to the network architecture, which we include in the ablation study (Table 1).

---

> ### Author Response · Authors · 2025-12-02
> **Final Remarks (Part 3/3)**
>
> ## **Reviewer PJtk  (Rating: 4)**
> **PJtk** recognizes that the paper shows comprehensive empirical results across both JSP and FJSP benchmarks, demonstrating scalability and generalization, and highlights the novel insight that policies trained on random datasets can achieve superior generalization, an observation echoed by **Quv3**, who noted the practical importance of this finding for dataset construction, and by **fDH7**, who emphasized CDQAC’s ability to learn strong policies from weak or random trajectories. **PJtk** further remarks that the work is clearly written and provides meaningful insights for applying offline RL to combinatorial optimization, aligning with **JoUx**’s assessment of our writing.
>
> **PJtk**’s main concerns are about fairness in our comparison with online RL baselines, how we can train with limited training data, and that we could include additional baselines — online, offline and classical heuristics. In our rebuttal, we addressed these concerns as follows:
>
> **1. Difficulty of FJSP and JSP**
> - We explained that FJSP and JSP are both NP-Hard problems and are thus highly complex, as noted by all other reviewers **Quv3**, **fDH7**, and **JoUx**.
> - We explained that our environment consists of **26 state features** (see Table 7 in App. B).
>
> **2. Fairness and completeness of online baselines**
> - Added **Residual** as an additional RL baseline for JSP and FJSP, retrained to match the training setup of CDQAC and other baselines.
> - Improved Section 5 to clarify that DANIEL, FJSP-DRL, Residual, and CDQAC were all trained on the same instance distribution and validation set.
>
> **3. Request for more classical baselines**
> - We clarified that our comparison already contained metaheuristics, such as Genetic Algorithms (GA) and dispatching rules.
> - Added **CP and MIP** for JSP benchmarks, as shown in Table 2, complementing GA and dispatching rules.
>
> **4. Training on limited data**
> - This concern is due to a misunderstanding. We explained that we train on at least **25,000** transitions, obtained through 5 instances.
> - CDQAC is an off-policy method, meaning that it can reuse examples and that its target becomes more refined.
>
> **4. Training curves and convergence analysis**
> - Added CDQAC, BC, IQL, and Offline-LD training curves to Appendix G, showing superior stability and fast convergence.
> - Included a convergence analysis in Appendix I, showing that CDQAC and DANIEL, the best-performing online baseline, both converge to a stable policy.
> - Explained that CDQAC outperforms DANIEL on **out-of-distribution** instances, as shown in Table 5.
>
> **Table 5:** Comparison between CDQAC and DANIEL on in-distribution instances and out-of-distribution instances.
> | Setting | Size | Method | Greedy (%) | Sampling (%) |
> | --- | --- | --- | --- | --- |
> | **Generated (In-Distribution)** | **10×5** | DANIEL | **10.87** | **5.57** |
> |  |  | CDQAC | 11.56 | 5.98 |
> |  | **15×10** | DANIEL | 12.42 | 6.79 |
> |  |  | CDQAC | **11.10** | **5.85** |
> | **Benchmarks (Out-of-Distribution)** | **10×5 (trained)** | DANIEL | 10.59 | **5.26** |
> |  |  | CDQAC | **9.22** | 5.50 |
> |  | **15×10 (trained)** | DANIEL | 10.30 | 5.28 |
> |  |  | CDQAC | **9.69** | **4.78** |
> | **Large Generated (Out-of-Distribution) Trained 10x5** | **30×10** | DANIEL | 5.10 | 4.43 |
> |  |  | CDQAC | **4.43** | **3.11** |
> |  | **40×10** | DANIEL | 3.65 | 3.77 |
> |  |  | CDQAC | **3.17** | **2.21** |
>
> **5. Clarifications on metrics and experiment design**
> - Section 5 was revised to integrate clearer definitions and experimental details, ensuring that CDQAC and the online baselines are trained with equal setups.
>
> ---
>
> # **Closing Statement**
>
> We conducted extensive additional experiments — including new offline baselines (BC, IQL), a new online baseline (Residual), extended ablations, training-curve evaluations, and expanded JSP comparisons with MIP/CP and additional JSP baselines. Moreover, we improved clarity on the experimental setup to ensure that CDQAC is trained and evaluated under equal conditions compared to online baselines, by stating that we use the validation set for all of them and that the training instances are generated from the same distribution.
>
> We strongly believe our additional experiments and improved explanation address all major concerns and significantly strengthen the submission.

---

### Meta-Review · Area_Chair_teKU · 2025-12-28

**Summary:**

1.Lack of ablation study by Reviewer JoUx and Reviewer Quv3.

2.Questions about the evaluation metric(the optimality gap), training time of RL, and larger problem sizes by Reviewer fDH7.

3.Lack of  comparisons with stronger imitation learning baselines by Reviewer Quv3.

4.The reason why policies trained on random datasets outperform near-optimal data by Reviewer Quv3 and Reviewer PJtk.

5.The comparison set for both offline and online RL baselines is narrow by Reviewer PJtk.

6.The claim that CDQAC trained on random data outperforms fully trained online PPO-based methods is difficult to accept by Reviewer PJtk.

**Reviewer Concerns:**

1.Lack of ablation study by Reviewer JoUx and Reviewer Quv3: during the rebuttal phase, the authors provide ablation studies, which show the effect of each components: dueling networks, quantile, CQL and policy delay. Thus I think this point is addressed.

2.Questions about the evaluation metric(the optimality gap), the training time of RL, and larger problem sizes by Reviewer fDH7: during the rebuttal phase, the author provides clarifications about the optimality gap and the training time of RL. The author also provides results on larger problem sizes. Thus I think this point is addressed.

3.Lack of  comparisons with stronger imitation learning baselines by Reviewer Quv3: during the rebuttal phase, the author includes additional baselines for JSP. Thus I think this point is addressed.

4.The reason why policies trained on random datasets outperform near-optimal data by Reviewer Quv3 and Reviewer PJtk: during the rebuttal phase, the author explains that the state-action coverage of random datasets is much larger than that of near-optimal datasets. I do not agree. The quality of random datasets is quite low though the state-action coverage is higher. CQL trained from random datasets achieves the worst performance compared with trained from medium, mixed and expert datasets in CQL paper. Thus I think this point is still outstanding.

5.The comparison set for both offline and online RL baselines is narrow by Reviewer PJtk: during the rebuttal phase, the author compares with Behavioral Cloning (BC) and Implicit Q-Learning (IQL) for offline algorithms, and compares with Residual for online RL. I think it is acceptable to compare with BC and IQL for offline algorithms. Thus I think this point is addressed.

6.The claim that CDQAC trained on random data outperforms fully trained online PPO-based methods is difficult to accept by Reviewer PJtk: during the rebuttal phase, the author claims that offline RL algorithms converges to a stable policy. However, the claim that CDQAC trained on random data outperforms online RL algorithms is wrong. The reason why CDQAC outperforms DANIEL is that DANIEL is not a sota online RL baselines. Online RL algorithms can explore new correct data by interacting with environments, and improve the policy. If you train your policy online until enough time, you will definitely get better policies than CDQAC. Thus I think this point is still outstanding.

**Reviewer Scores:**

Reviewer JoUx would keep his or her score as 8 if he or she has been able to participate fully in the discussion.

Reviewer fDH7 would keep his or her score as 6 if he or she has been able to participate fully in the discussion.

Reviewer Quv3 would keep his or her score as 4 if he or she has been able to participate fully in the discussion.

Reviewer PJtk would keep his or her score as 4 if he or she has been able to participate fully in the discussion.

---

### Decision · Program_Chairs · 2026-01-26

Reject